# Biomimetic supramolecular protein matrix restores structure and properties of human dental enamel

Abshar Hasan[1,2,3,4], Andrey Chuvilin [5,6], Alexander Van Teijlingen [7], Helena Rouco[1,2], Christopher Parmenter [1,8], Federico Venturi[9], Michael Fay [8,9], Gabriele Greco [10,11], Nicola M. Pugno [11,12], Jan Ruben[13], Charlotte J. C. Edwards-Gayle[14], Benjamin Myers[15], Ingrid Dreveny [1], Nathan Cowieson[14], Adam Winter[16], Sara Gamea [17,18], X. Frank Walboomers[13], Tanvir Hussain [9], José Carlos Rodríguez-Cabello [19], Frankie Rawson [2,15], Tell Tuttle [7], Sherif Elsharkawy [17], Avijit Banerjee [17], Stefan Habelitz [20] & Alvaro Mata [1,2,3,4] ✉

Tooth enamel is characterised by an intricate hierarchical organization of apatite nanocrystals that bestows high stiffness, hardness, and fracture toughness. However, enamel does not possess the ability to regenerate, and achieving the artificial restoration of its microstructure and mechanical properties in clinical settings has proven challenging. To tackle this issue, we engineer a tuneable and resilient supramolecular matrix based on elastin-like recombinamers (ELRs) that imitates the structure and function of the enamel-developing matrix. When applied as a coating on the surface of teeth exhibiting different levels of erosion, the matrix is stable and can trigger epitaxial growth of apatite nanocrystals, recreating the microarchitecture of the different anatomical regions of enamel and restoring the mechanical properties. The study demonstrates the translational potential of our mineralising technology for treating loss of enamel in clinical settings such as the treatment of enamel erosion and dental hypersensitivity.

Formation of dental enamel, the hardest and most mineralised tissue of vertebrates[1], relies on the 3D assembly and organisation of the protein amelogenin[2]. During development, this protein transforms from a disordered conformation to ordered β-rich fibrillar structures[3], nucleating and epitaxially growing hierarchically organised apatite nanocrystals[4,5]. This process results in a highly structured enamel tissue that protects teeth throughout life by providing strength and anti-abrasive properties against physical, chemical, and thermal insults[6]. However, irreversible enamel loss due to mechanical wear, trauma, or erosion from acidic foods or fermentation by-products from bacteria can lead to carious lesions and ultimately tooth loss[7]. These oral problems affect nearly half of the global population at an annual cost of ~ US$544 billion[8].

The properties of enamel result from its composition and structure[1]. Carbonated hydroxyapatite (CHAp) nanocrystals of - 50 nm in diameter and micrometres in length are the building blocks of dental enamel. They are aligned, densely packed, and organised in specific configurations in different anatomical regions. The outermost aprismatic enamel layer in permanent teeth is ~ 16–45 μm thick and composed of aligned apatite nanocrystals with their c-axis perpendicular to the tooth surface[9]. Subjacent to this, prismatic enamel comprises nanocrystals organised into - 4–8 μm thick prisms[10], which are delineated by inter-prismatic crystallites aligned at a - 60° angle to the prism axis[1]. Here, undulating trajectories of prisms create decussation patterns known as Hunter-Schreger bands, forming longitudinal (parazone) and transverse (diazone) bundles[11,12]. The stiffness of

enamel, expressed as Young's modulus (E), allows small deformations under high forces (e.g., during mastication) while its hardness (H) enables it to resist localised plastic deformation due to indentation or scratching (e.g., during teeth grinding)[13]. The wear strength of enamel protects teeth from chemical erosion (e.g., from extrinsic or intrinsic acids), abrasion (e.g., toothbrushing), or attrition (e.g., tooth-on-tooth contact), while its toughness allows it to tolerate fracture without falling apart. The combination of all these properties allows enamel to withstand thousands of chewing cycles per hour[14], biting forces of up to 770 N, and decades of exposure to harsh oral environments[1].

The recreation of dental enamel has been a major goal in materials science[15]. Methods based on acidic[16], hydrothermal[17], and high-power laser[18] treatments have enabled the ambient deposition of mineral, but without the capacity to recreate the architecture and function of healthy enamel. Recently, several enamel analogues comprising organic-inorganic nanocomposites such as polyacrylic acid-zinc oxide nanowires[19] and polyvinyl alcohol-HAp nanowires[20] have been reported to generate functional hierarchical architectures. However, these technologies rely on non-physiological fabrication conditions, which restrict their applicability in clinical settings[21].

To address the clinical need, bioinspired approaches offer promising alternatives to regrow different anatomical regions of enamel[22]. For instance, Moradian-Oldak and colleagues have developed amelogenin-based scaffolds capable of growing layers of aprismatic enamel-like structures ex vivo[9]. Other approaches based on phase-transited lysozyme films with amelogenin-like peptides[23], amelogenin-derived shortened ADP5 (shADP5) oligopeptide[24], modified leucine-rich amelogenin peptide (mLRAP)[25], phosphorylated and phosphonated peptides[26], or triethylamine stabilised calcium phosphate ions[27] have reported remineralisation of prismatic enamel. However, these strategies suffer from drawbacks that restrict their clinical translation such as having toxic and noxious components (e.g., triethylamine[27]), time-consuming application processes (e.g., 30 min[23], 15 min[27], 12 h[26], 10 min/day[28]), partial recovery of architecture and functional properties (e.g., aprismatic enamel[9], prismatic enamel[23,25–27]), and limited control of the mineralisation process. Thus, remineralising technologies that recreate the diverse architecture and spectrum of functional properties of dental enamel in a patient and clinically-friendly manner remain an unmet challenge.

In this study, we report on the design and performance of a clinically-friendly supramolecular protein matrix using disordered elastin-like recombinamer (ELR) molecules to emulate key structural and functional features of the β-rich fibrillar amelogenin matrix driving epitaxial and hierarchical mineralisation in dental enamel. We use both computational and experimental work to describe the matrix design and demonstrate the epitaxial growth of mineralised layers up to 10 μm thick from the surface of teeth exhibiting different levels of erosion down to bare exposed dentine ex vivo. The matrix is highly stable and able to regrow all the anatomical features of enamel, including prismatic, interprismatic, and aprismatic regions, while restoring tissue stiffness, hardness, toughness, coefficient of friction, wear strength, and stability to extensive abrasion from toothbrushing, chewing and grinding, and exposure to acidic solutions.

## Results and discussion

### Rationale for the design of the mineralising ELR matrix to emulate enamel development

Recent studies have shown that intrinsically disordered amelogenin molecules assemble into functional β-rich fibrillar structures via intermolecular interactions in the presence of $Ca^{2+}$ ions[29,30]. These structures nucleate and grow apatite nanocrystals in vitro and during enamel development[4,5]. We have demonstrated the capacity to grow organised mineralised structures by modulating the levels of order and disorder of an intrinsically disordered elastin-like recombinamer (ELR) consisting of hydrophobic (VPGIG), hydrophilic (VPGKG), and acidic

statherin (DDDEEKFLRRIGRFG) motifs[31]. We hypothesised that the disordered nature of the ELR molecules can be harnessed to generate supramolecular ELR ensembles that imitate structural and functional characteristics of the amelogenin matrix[2–4]. By generating these amelogenin-like β-rich supramolecular assemblies over enamel or dentine, it may be possible to facilitate epitaxial, integrated, and hierarchical enamel-like mineralisation from dental tissues. Thus, we combined $Ca^{2+}$ ions with ELR molecules, which are reported to promote β-sheet conformation in proteins[32] and peptides[33] via ion bridge formation. We also used drying as a mechanism to induce molecular order through molecular crowding, as previously reported[34,35]. We reasoned that the incorporation of $Ca^{2+}$ ions and drying of the ELR solution can together be used to promote β-rich fibrillar ELR structures, which we refer to as 'ELR fibrils'. Furthermore, the incorporated $Ca^{2+}$ ions would also serve as nucleating points to facilitate mineralisation. In addition, given the capacity of the ELR matrix to create a confined environment for mineral nucleation and growth[31], we envisage that the thickness of the ELR layer applied to dental tissue can spatially regulate the resulting mineral layer thickness in situ. To further support clinical translation, the choice of components in the formulation has been guided by considerations of rapid application, ease of use, and overall practicality in a clinical setting.

### Supramolecular assembly of the mineralising ELR matrix

We placed a drop of 1% w/v ELR solution without or with 1.5 mM $Ca^{2+}$ ions on either scanning electron microscope (SEM) stubs or transmission electron microscope (TEM) grids and allowed them to dry at room temperature (25 °C). SEM and TEM imaging revealed that the ELR molecules assembled into ELR fibrils only in the presence of $Ca^{2+}$ ions (Fig. 1a, Supplementary Fig. 1 and Supplementary Discussion 1). These fibrils ranged between 15–40 nm in width and several micrometres in length, which corresponds to the classical fibrillar geometry[36]. Presence of $Ca^{2+}$ ions within these fibrils was confirmed by X-ray photoelectron spectroscopy (XPS) while weak electrostatic interactions between $Ca^{2+}$ ions and ELR molecules were confirmed by isothermal titration calorimetry (ITC) and UV-Vis (Supplementary Fig. 2). Furthermore, we demonstrate the possibility to use $Ca^{2+}$ ions and crosslinking during matrix fabrication to engineer ELR matrices with tuneable levels of ordered β-conformations, fibril structures (Supplementary Discussion 2, 3; Supplementary Figs. 3–6), and number of potential nucleation points for mineralisation (Supplementary Fig. 7). These results demonstrate the synergistic role of $Ca^{2+}$ ions, crosslinking, and solvent evaporation in forming ELR fibrils, while generating a tuneable matrix with Ca-rich nucleating sites for mineralisation.

### Structural characterisation of the ELR fibrils

Wide-angle X-ray scattering (WAXS) analysis on the ELR fibrils revealed a sharp peak at $Q = 1.3$ Å$^{-1}$ corresponding to a periodic β-strand separation $d_\beta = 4.7$ Å perpendicular to the fibril axis and a broader hump at $Q = 0.6$ Å$^{-1}$ corresponding to a characteristic inter-sheet packing distance of 10 Å in a two-dimensional (2D) crystalline lattice (Fig. 1b)[37]. This is a typical cross-β diffraction pattern observed in several peptides and proteins that exhibit fibrillar structures[37]. Furthermore, the 4.7 Å β-strand separation corresponds to stacks of hydrogen-bonded β-sheets in folded protein structures[37,38] and is similar to the structural motifs found in both the protein matrix of developing human enamel[39] and amelogenin ensembles formed in vitro[40]. Peak at $Q = 1.04$ Å$^{-1}$ can be attributed to interactions between α-helices[41] while peaks at $Q = 1.07, 1.49, 1.72$, and $2.0$ Å$^{-1}$ (Fig. 1b) align with the (001), (111), (112), and (211) Miller indices of $CaCl_2$ suggesting peaks arising from $CaCl_2$ crystallisation (Supplementary Fig. 8c). In addition, FTIR analysis revealed the presence of antiparallel β-sheets (Fig. 1c)[42], which are also comparable to those in the enamel-developing amelogenin matrix[3,5]. On the other hand, small-angle X-ray scattering (SAXS) analysis showed a low Q-decay of 4.1,

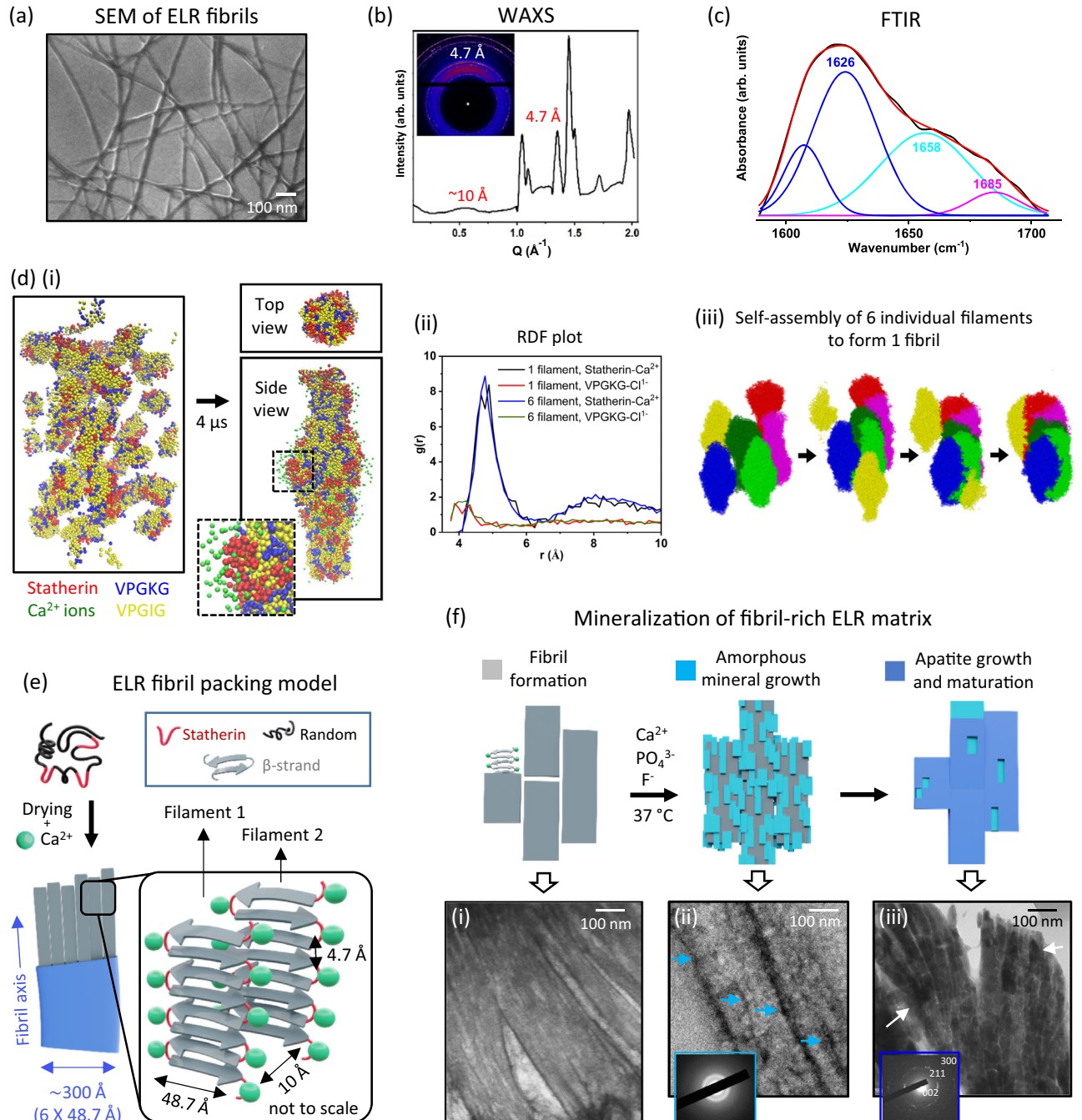

**Fig. 1 | ELR fibril formation and mineralisation capacity. a** SEM micrograph of ELR fibrils formed after drying a drop of 1% w/v ELR solution (in DMF/DMSO at 9/1 ratio) containing 1.5 mM Ca²⁺ ions. Representative micrograph from $n = 5$ independent experiments. **b** WAXS data of ELR fibrils show reflections at Q = 1.34 and 0.6 Å⁻¹ which correspond to 4.7 and 10 Å, respectively. The inset shows a 1030 pixel wide WAXS detector image with the beam position slightly off-centred due to its setup. The inset shows a WAXS detector image of dried ELR fibrils. **c** Deconvoluted FTIR spectra (1600 to 1700 cm⁻¹) showing the antiparallel arrangement of the β-sheet. **d** (i) Computational model depicting self-assembly of ELR molecules in the presence of Ca²⁺ ions, 30 equilibrated ELR molecules (β-sheet secondary structure with coiled secondary structure for the statherin motif) were inserted into a 16 x 16 x 30 nm aqueous rectangular cuboid box and equilibrated with a 0.2 M concentration of Ca²⁺ ions. (ii) RDF plot of statherin – Ca²⁺ and VPGKG – Cl¹⁻ interactions with statherin shown in red, Ca⁺² ions shown in green within and at the surface of the protein aggregate, other protein residues are shown in transparent

yellow. (iii) Aggregation of 6 individually coloured protofilaments, each containing 30 ELR proteins. The system starts at an evenly distributed collection of filaments that, over time, aggregate along a line. **e** Schematic illustration of the proposed model showing the supramolecular organisation of the ELR molecules within ELR fibrils. Each ELR fibril (blue layer) is ~300 Å in thickness and comprises of 6 individual filaments (grey sticks) measuring 48.7 Å thick each. **f** Schematic illustration and their corresponding TEM images showing the formation and mineralisation of the ELR matrix. (i) a 2 µL drop of 5% w/v ELR solution containing 1.5 mM Ca²⁺ ions and hexamethylene diisocyanate (HDI, 0.56% v/v) crosslinker was casted on a TEM grid and dried to form a crosslinked matrix of ELR fibrils. (ii) Formation of amorphous calcium phosphate (ACP, blue arrows) phase at 2 h of incubation in mineralisation solution containing 10 mM Ca²⁺, 6 mM PO₄³⁻, and 2 mM F⁻ ions. (iii) Transformation of ACP into apatite nanocrystals (white arrows) after 24 h of mineralisation. Schematic illustrations in (**e**, **f**) were prepared by scientific illustrator Leonora Martínez Nuñez.

characteristic of a large surface fractal nanostructure with smooth surfaces (Supplementary Fig. 8a), which correlates with SEM observations (Fig. 1a) depicting a network of smooth fibrils. A small broad peak at $Q = 0.129$ Å$^{-1}$ (Supplementary Fig. 8a) corresponding to a d-spacing of 48.7 Å was attributed to the width of an individual filament[43], which undergoes periodic stacking to form ELR fibrils similar to those reported for β-rich fibrillar structures[43]. Interestingly, 48 Å wide filaments arranged parallel to each other have been observed in embryonic bovine enamel matrix[44], which are structurally similar to the 48.7 Å wide filaments of our ELR fibrils. These results demonstrate the capacity of the ELR matrix to form cross-β fibrillar structures exhibiting similar supramolecular organisation and dimensions as those found in the amelogenin-rich enamel matrix.

## Coarse-grained simulation of the formation of ELR fibrils

To better understand the underlying matrix assembly mechanism, we performed simulations of the ELR molecules using the MARTINI (v3) forcefield. Thirty ELR molecules were equilibrated in water with Ca$^{2+}$ ions and self-assembled into an elongated filament with VPGIG forming the core of the structure and VPGKG and statherin largely localised at the surface (Fig. 1d (i)). We only observed filament formation in the presence Ca$^{2+}$ ions (Supplementary Fig. 9), which is consistent with the experimental SEM results (Fig. 1a). Furthermore, measurement of radial distribution function (RDF) confirmed interactions between Ca$^{2+}$ ions and statherin regions of single filaments as well as ELR aggregates of 6 filaments, where Ca$^{2+}$ ions interacted with their surface and absorbed into their internal structure (Fig. 1d (ii)). On the other hand, interactions between the counter ion (Cl$^{1-}$) and VPGKG motifs were not observed (Fig. 1d (ii)). In addition, we investigated the aggregation of these filaments to form the fibril structure by inserting 6 filaments into a box with water between them and equilibrated. These filaments aggregated parallel to each other (Fig. 1d (iii)) and in concordance with experimental observations, displayed a width of 51.6 Å/filament. Interestingly, increasing the number of filaments to 8 did not lead to assembly along a single plane. These results confirmed the role of Ca$^{2+}$ ions in facilitating ELR assembly into filaments, which are in turn assembled into the fibrillar structure.

## ELR fibril formation model

Based on experimental and simulation results, we propose a model for the molecular packing of the ELR fibrils (Fig. 1e). From WAXS data, we assigned the 10 Å to the inter-sheet packing distance and 4.7 Å to the periodic β-strand separation along the fibril axis[37,38], while SAXS data was used to assign the 48.7 Å to the individual filament width[43]. Furthermore, from the simulations, the hydrophobic VPGIG motifs would aggregate at the core of the filament, while charged VPGKG and statherin motifs would localise in the outer layers of each filament. In addition, V and G residues would promote ordered β-conformation[45] and the negatively charged regions (statherin) would interact electrostatically with Ca$^{2+}$ ions during the formation of the ELR fibrils (Fig. 1e). This arrangement of hydrophobic and charged motifs is in agreement with peptide-based fibrillar models[3,46]. Furthermore, given the average fibril width of ~300 Å (Fig. 1a) and individual filament width of 48.7 Å, we conclude that an average ELR fibril consists of ~6 filaments. These results demonstrate the potential to use our mineralising technology to recreate structural features of the natural amelogenin matrix with high precision and reproducibility.

## Mineralisation of the ELR fibrils

Given the key role of β-rich fibrillar structures in enamel development[4], we then investigated the mineralisation capacity of our ELR matrix. First, a drop of 5% w/v ELR solution containing 1.5 mM Ca$^{2+}$ ions and 0.56% v/v hexamethylene diisocyanate (HDI) crosslinker was casted and dried on a TEM grid to form a 50–100 nm thick ELR coating. TEM imaging confirmed the presence of ELR fibrils within the coating

(Fig. 1f, (i)). Upon exposure to a mineralisation solution supersaturated with respect to fluorapatite[31], ELR fibrils templated the formation of ~20–40 nm thick needle-like amorphous mineral platelets were observed within the ELR matrix at 2 h (Fig. 1f, (ii)), which evolved into highly crystalline fluorapatite nanocrystals after 24 h (Fig. 1f, (iii), Supplementary Fig. 10). These nanocrystals were ~50 nm in diameter, ~1 μm in length, and aligned along the direction of the ELR fibrils. These geometrical characteristics are similar to those growing within the amelogenin-rich matrix during enamel development[4,5], as well as in systems involving amelogenin peptide[47] and other protein-templated mineralisation[31].

## ELR matrix-guided preferential crystal growth along the c-axis

During early enamel development, the amelogenin matrix stabilises and fuses prenucleation clusters via a non-classical crystallisation process, guiding the growth of apatite nanocrystals preferentially along the c-axis[4,47]. Thus, we focussed on assessing the capacity of the ELR matrix to preferentially control the direction of crystal growth. To do this, we embedded commercially available HAp nanocrystals within the ELR matrix deposited on the TEM grid. After 24 h in the mineralisation solution, precursor amorphous mineral was observed, which may arise by the fusion of prenucleation clusters along the c-axis of the existing nanocrystals (Fig. 2a, blue square), as previously reported[47,48]. This amorphous mineral undergoes transformation into crystalline structures with identical orientation (Fig. 2a, yellow square), as seen in classical amorphous-to-crystalline transitions during apatite formation[27,49]. These results demonstrate the capacity of our supramolecular ELR matrix to facilitate the oriented, integrated, and anisotropic growth of the existing crystals, reminiscent of the mineralisation mechanism observed during amelogenesis[4,5]. In addition, these results were complemented with umbrella sampling simulations, which revealed that more energy is required to remove ELR fragments bound on the a-axis than the c-axis of the apatite nanocrystal (Supplementary Fig. 11), as reported previously for a small 12-mer amelogenin sequence[50], thus energetically favouring crystal growth along the c-axis. In contrast, nanocrystals mineralised without an ELR matrix exhibited irregular growth due to the fusion of amorphous mineral along all the crystal axes (Supplementary Fig. 12). Overall, these results confirmed the role of the ELR matrix in directing epitaxial crystal growth preferentially along the c-axis.

## Remineralization of enamel and dentine

An ideal enamel regenerative technology would enable remineralisation of the different architectures of the natural tissue (Fig. 2b) and the recovery of its functional properties. Towards this goal, we first assessed the capacity of the ELR matrix to regrow aprismatic enamel. We drop cast 10 μL of 5% w/v ELR solution containing 1.5 mM Ca$^{2+}$ ions and 0.56% v/v HDI on a 2 mm x 4 mm area of acid-etched aprismatic enamel to form a ~10 μm thick ELR coating as observed under SEM (Fig. 2c and Supplementary Fig. 14c) and exposed it to the mineralisation solution supersaturated with respect to fluorapatite[31] for 10 days. The ELR matrix induced epitaxial growth of apatite nanocrystals from the enamel-ELR interface through the ELR matrix, forming an integrated and organised mineral layer (Fig. 2d and Supplementary Discussion 4). This layer comprised ~50 nm diameter nanocrystals that were several microns in length and exhibited the typical hexagonal apatite morphology (Fig. 2d inset) of native enamel. Then, to assess remineralization of prismatic enamel (Fig. 2e), we similarly deposited a ~2 μm thick ELR coating on acid-etched prismatic enamel (Fig. 2f and Supplementary Fig. 14a) and repeated the 10-day mineralisation process, which triggered the epitaxial growth of similar nanocrystals but now maintaining the distinctive prismatic and inter-prismatic architectures (Fig. 2g). Finally, to test the remineralizing potential of the ELR matrix on teeth with completely eroded enamel (Fig. 2h), we deposited a ~5 μm thick ELR coating on acid-etched dentine, repeated the

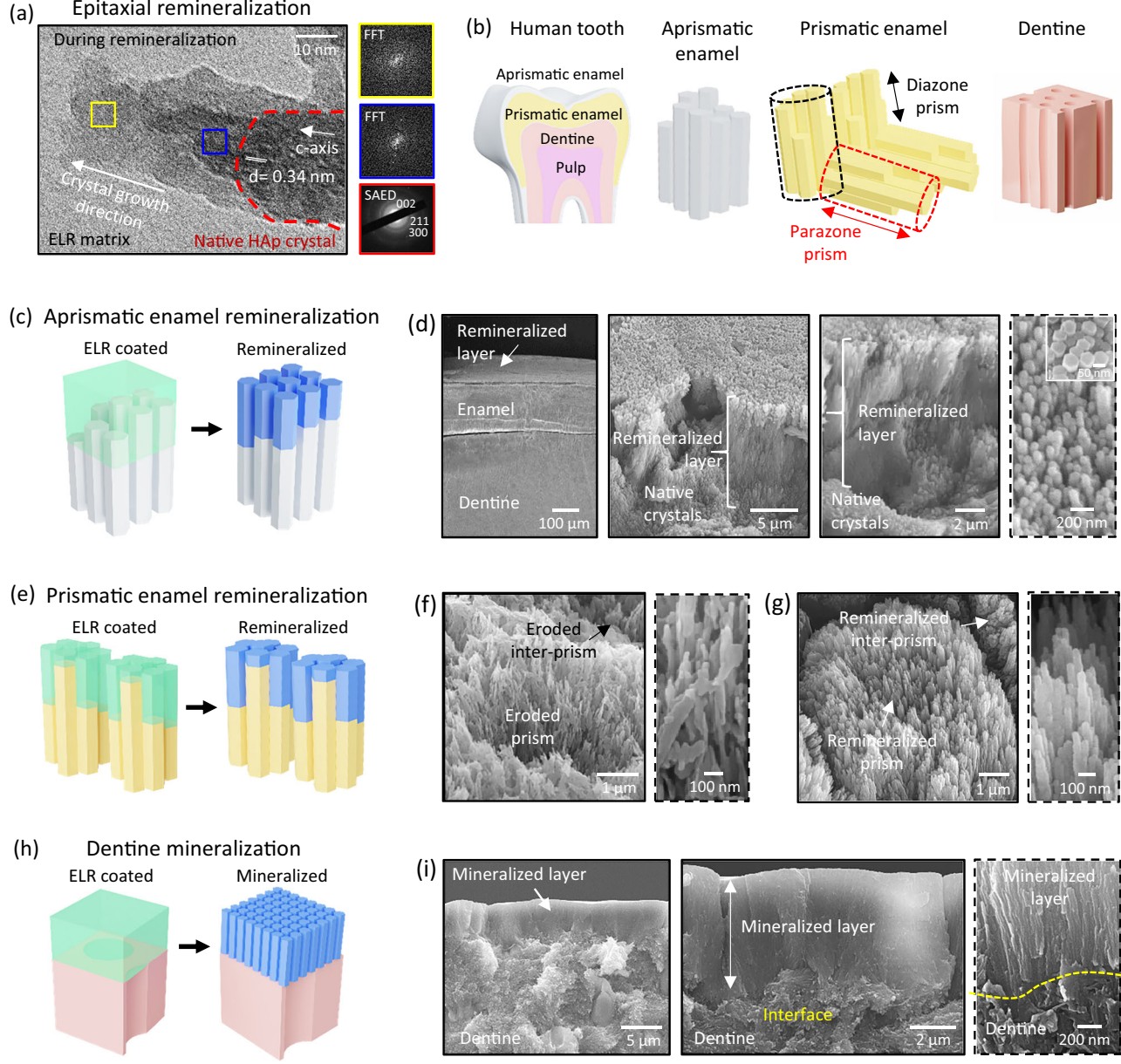

**Fig. 2 | Epitaxial remineralization from different anatomical regions of the tooth. a** TEM image showing the preferential growth of an apatite nanocrystal along the c-axis while embedded within the ELR matrix (representative image from $n = 3$ independent experiments). High-resolution TEM and FFT images are presented in the Supplementary Fig. 13. The lattice lines in the apatite nanocrystal grown within the ELR coating appear blurry due to imaging interference and potential beam-induced damages caused by ionisation and electrostatic charging of the beam-sensitive ELR matrix, a common issue for poorly conducting organic samples. **b** Illustration of different anatomical parts of a tooth. **c** Illustration of ELR coating on a section of aprismatic enamel and (**d**) SEM images of a ~10 μm thick apatite layer grown on aprismatic enamel (representative images from $n = 3$ independent experiments). **e** Illustration of ELR coating on a section of prismatic enamel and SEM images of prismatic enamel (**f**) before and (**g**) after remineralisation (representative images from $n = 8$ independent experiments for each group). **h** Illustration of ELR coating on dentine surface and (**i**) SEM images of a ~5 μm thick apatite layer grown on dentine surface (representative images from $n = 5$ independent experiments). Schematic illustrations in (**b, c, e, h**) were prepared by scientific illustrator Leonora Martínez Nuñez.

mineralisation process, and again observed the epitaxial growth of apatite nanocrystals. In this case, the remineralized layer resembled the architecture of aprismatic enamel but now growing from the dentine surface (Fig. 2i). In all three cases, similar results were obtained using artificial saliva (1.2 mM Ca$^{2+}$, 0.72 mM PO$_4^{3-}$) containing 1 ppm F$^{1-}$ ions to imitate physiological conditions (Supplementary Fig. 15). We used 1 ppm of F$^{1-}$ concentration in artificial saliva, which has been reported to be the average amount present in natural saliva ranging between 0.02 ppm to 1.93 ppm[51]. We have conducted experiments using three different thicknesses of the ELR coating to demonstrate (i) the tuneability of our process to deposit coatings of specific thicknesses (Supplementary Fig. 14) and (ii) that nanocrystal growth is defined by the ELR coating, generating a mineralised layer of equal thickness to the ELR coating (Supplementary Figs. 16, 17). In contrast, tissue sections without ELR coating developed misoriented crystals (Supplementary Fig. 18) commonly found in mineralisation processes without organic matrices[52].

These results suggest that our technology could potentially provide a one-pot solution for the regeneration of dental enamel independently of the level of tooth erosion. To the best of our knowledge, this capacity to regenerate enamel from different anatomical regions of teeth expanding from aprismatic enamel,

prismatic enamel, and exposed dentine has not been achieved before. Here, we define regeneration as the capacity to recreate the architecture of the different anatomical regions of enamel and regain its functional properties. Thus, given the potential clinical implications of such a versatile regenerative technology, we then performed in depth investigations of the remineralisation from each of these anatomical regions.

## Regeneration of prismatic enamel

We fabricated a uniform ~2 μm thick ELR coating on an acid-etched prismatic enamel (Fig. 3a, b and Supplementary Fig. 19). Upon mineralisation, the ELR coating induced the epitaxial growth of apatite nanocrystals from the enamel-ELR interface through the ELR matrix (Fig. 3c and Supplementary Fig. 20), recovering the native architecture of both diazone (Fig. 3d and Supplementary Fig. 21) and parazone (Fig. 3e and Supplementary Figs. 22, 23) prisms as well as inter-prismatic regions (Fig. 3d, e). Similarly, a 5 μm thick ELR coating on prismatic and aprismatic regions of enamel triggered the uniform growth of a ~ 5 μm thick mineral layer over large and uneven areas, recreating the microstructure of both enamel anatomies (Supplementary Fig. 15). While the ELR matrix slowly degrades during the mineralisation process, it remains stable and functional during the enamel remineralization process (Supplementary Fig. 24 and Supplementary Discussion 5). To expedite this ELR degradation process and expose the underlying remineralised layer, we treated the samples with elastase after the remineralisation process was completed (Fig. 3d, e and Supplementary Figs. 21–23). Nanoindentation and microtribological tests were then conducted to assess the capacity of this remineralised enamel to restore key mechanical properties of the native tissue, including Young's modulus (E), hardness (H), coefficient of friction (CoF), and specific wear rate (SWR). As expected, acid-etched enamel displayed dramatically lower E ($36.9 \pm 14.3$ GPa) and H ($1.1 \pm 0.6$ GPa) compared to E ($80.7 \pm 18.3$ GPa) and H ($3.4 \pm 0.9$ GPa) of native enamel (Fig. 3f). This effect results from surface and sub-surface porosities created by the acid treatment that lead to micro-cracking[53] and lower mechanical properties[54]. However, both E ($76.3 \pm 18.7$ GPa) and H ($3.1 \pm 0.8$ GPa) were regained after remineralisation.

Furthermore, we measured CoF and SWR, which together provide an assessment of wear resistance properties[55]. Acid-etched enamel displayed higher CoF ($0.38 \pm 0.04$) and SWR ($4.15 \times 10^{-3} \pm 4.26 \times 10^{-4}$ mm$^3$/Nm) than native enamel (CoF = $0.25 \pm 0.05$; SWR = $1.49 \times 10^{-3} \pm 2.95 \times 10^{-4}$ mm$^3$/Nm) (Fig. 3g and Supplementary Fig. 25a) due to disrupted nanocrystals generating more contact area[56]. However, upon remineralisation, CoF ($0.29 \pm 0.03$) and SWR ($1.68 \times 10^{-3} \pm 1.96 \times 10^{-4}$ mm$^3$/Nm) were again lowered due to recovery of the tissue structure, which is significantly lower than previously reported[23]. We also calculated wear strength (WS), which is the ratio between CoF and SWR and provides a measure of resistance to fatigue failure (Supplementary Fig. 25b). WS also recovered after remineralization from $91.6 \pm 0.23$ GPa to $171.3 \pm 3.8$ GPa (Fig. 3h). This value was significantly higher than that of native enamel ($153.9 \pm 3.2$ GPa), suggesting superior fatigue resistance properties compared to native enamel. This enhanced behaviour may arise from the more densely packed apatite nanocrystals in remineralised enamel (Supplementary Fig. 26). These results demonstrate that our ELR matrix triggers epitaxial, organised, and hierarchical growth of apatite nanocrystals, recreating the architecture of all of the prismatic enamel and restoring key mechanical and microtribological properties.

## Confirmation of the epitaxial remineralization

To assess the crystallographic integration of the newly grown nanocrystals to the underlying native tissue, milled ultrathin sections from remineralised diazone and parazone prisms were cut via a focused ion beam (FIB) and analysed using TEM. Scanning TEM (STEM) and TEM images from remineralised diazone prisms (Fig. 4a, b) depicted new nanocrystals with their c-axis growing and extending from the c-axis of the underlying native nanocrystals. Selected area electron diffraction (SAED) patterns confirmed this crystallographic alignment (Fig. 4a insets). TEM image in Fig. 4b shows that the new nanocrystals extend from the native tissue, while high-resolution TEM (HRTEM) imaging at the interface revealed an indistinguishable boundary between them (Fig. 4c), verifying crystallographic integration via epitaxial remineralization[27]. To further visualise the interface between new and native nanocrystals in diazone prisms, we carefully enzymatically degraded the ELR matrix from the top, leaving only the very bottom to serve as the point of transition between new and native nanocrystals. At this point, high magnification SEM imaging (Fig. 4d) distinctly shows new nanocrystals nucleating and extending from underlying native nanocrystals along the c-axis, complementing the TEM results (Fig. 4a–c). Similarly, FIB sections prepared from remineralised parazone prisms were analysed under TEM (Fig. 4e). EDX mapping revealed a distinct fluorine (F) signal arising from the newly grown fluorapatite crystals (Fig. 4e–g and Supplementary Fig. 27a) while no F signal was detected in the native enamel nanocrystals, thus differentiating the interface between the native and remineralised regions. In addition, high magnification TEM and SAED analyses at the interface also revealed the epitaxial growth of new nanocrystals extending from the end of the c-axis of native nanocrystals (Fig. 4h), exhibiting no distinguishable boundary between them (Fig. 4i). Interestingly, in this parazone region, we also observed bundles of new nanocrystals nucleating on the a-axis of the native nanocrystals while growing along their c-axis (Fig. 4j and Supplementary Fig. 28) following their crystallographic orientation (Fig. 4j insets). Furthermore, energy-dispersive X-ray spectroscopy (EDX), FTIR, and X-ray diffraction (XRD) analyses indicate that the remineralised enamel exhibits a similar chemical composition as native enamel (Supplementary Discussion 3, 7, Supplementary Figs. 29, 30). Overall, these results confirm the role of the ELR matrix in mediating epitaxial growth from all the different anatomical regions of enamel, further supporting the possibility that it may recreate mineralising features of enamel development[4].

## Confirmation of crystallographic integration with collagen fibrils on the dentine surface

We then investigated the potential of the ELR matrix to regrow an enamel-like layer from completely exposed dentine. We infer that the ELR matrix would penetrate the interwoven network of mineralised collagen fibrils (MCFs) and epitaxially grow apatite nanocrystals from it, as in the native dentine-enamel junction (DEJ) (Fig. 5a)[57,58]. To do this, a ~ 2 μm thick ELR coating was deposited on an acid-etched exposed dentine surface and mineralised for 10 days using solution supersaturated with respect to fluorapatite (Fig. 5b). SEM examination revealed the growth of apatite nanocrystals from the dentine-ELR interface through the ELR matrix, forming an organised mineral layer that was structurally similar to aprismatic enamel (Fig. 5c and Supplementary Fig. 31). Furthermore, TEM (Fig. 5d), EDX (Supplementary Fig. 32), and HRTEM (Fig. 5e, (i), (ii), (iii)) images of FIB sections revealed apatite nanocrystals growing from the dentine surface. Nanocrystals (~ 50 nm thick) nucleate and extend from MCFs via lattice continuity (Fig. 5e, (i), (ii)), where each MCF has been reported to be a bundle of 5 - 10 nm thick sub-fibrils[59]. A similar extension of enamel crystals from dentine MCFs has been observed during early amelogenesis[57,58]. While these results are in agreement with reports identifying an epitaxially integrated DEJ[57,58,60], it is important to mention that others have reported a lack of epitaxy between dentine and enamel[61,62]. Nonetheless, at this new mineralised layer-dentine junction, collagen fibrils that are organised orthogonal to the dentine cross-section[63] were observed to change direction and orient parallel to the mineralising nanocrystals (Fig. 5e, (i), (ii)). This reorientation

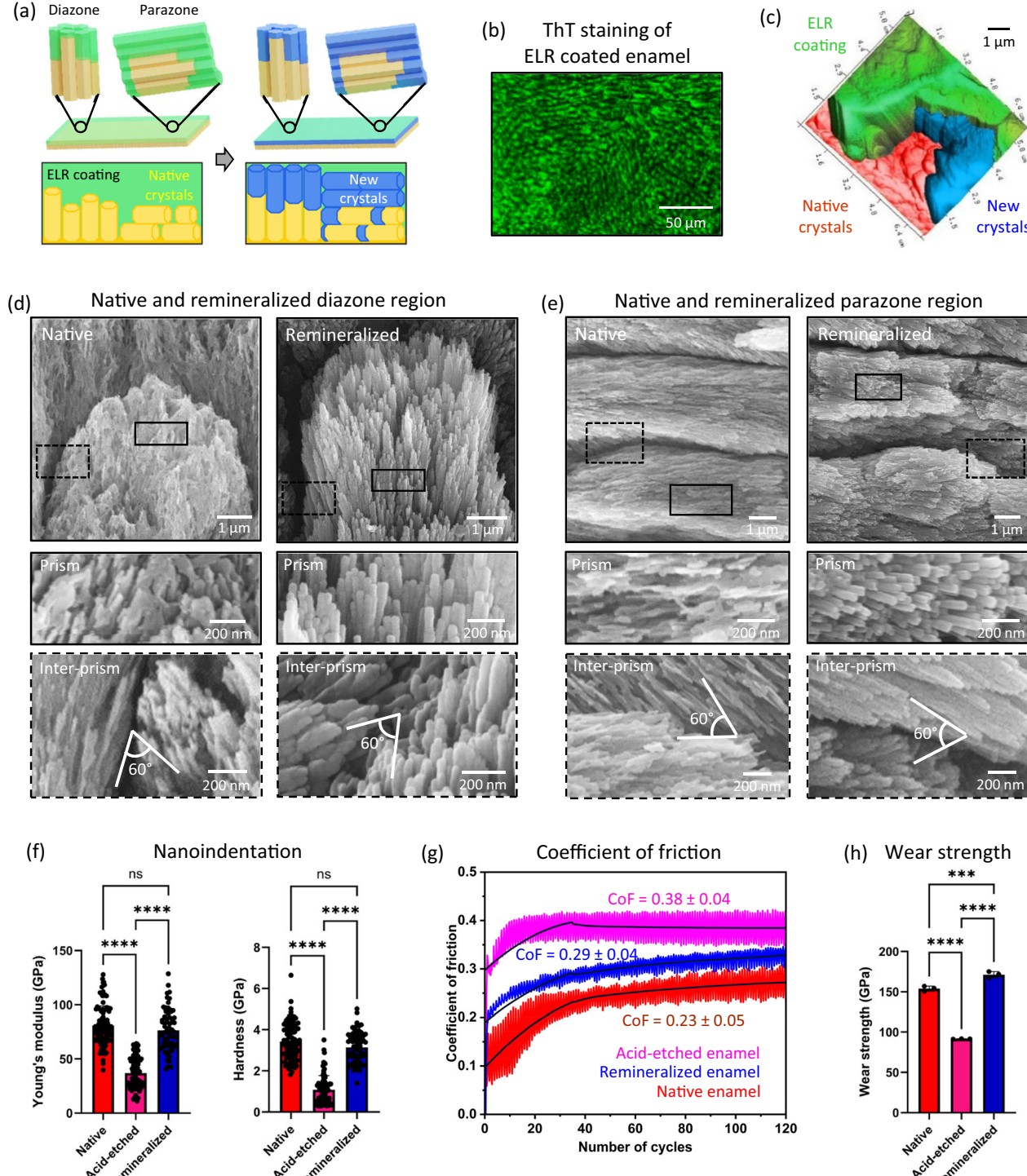

**Fig. 3 | Epitaxial remineralization recreates structure and restores function of prismatic enamel. a** Schematic illustration showing ELR coating and remineralisation of diazone and parazone prisms. **b** Confocal laser scanning microscopy (CLSM) image of ThT-stained ELR-coated prismatic enamel section. **c** Pseudo coloured AFM image of a partly mineralised enamel section showing distinct ELR coating (green), native enamel (brown), and newly grown nanocrystals (blue) at the ELR coating-enamel interface. After the mineralisation process, a part of the ELR coating was protected with nail varnish, while the exposed area was degraded using elastase to partially reveal the remineralised layer on the enamel surface. **d**, **e** SEM images of native and remineralised diazone and parazone prisms, recreating enamel microarchitecture of both prism and inter-prisms (representative images from n = 12 independent experiments). All the left panels in both Fig. 3d, e depict native enamel, while the right panels show remineralised enamel. The new crystals grow epitaxially from the native enamel, covering it and preventing its visualisation once the mineralised layer is formed. **f** Young's modulus (E) and hardness (H) of native, acid-etched, and remineralised enamel. Bars represent the mean values of E and H while dot plot represents indentation measurements compiled from 3 independent experiments for each group. **g** Coefficient of Friction, and (**h**) Wear strength of native, acid-etched, and remineralised enamel samples (n = 3 samples for each group). Data are presented as mean ± SD. Statistical significance was analysed using two-sided one-way ANOVA (Tukey test) in GraphPad Prism ver. 10. In (**f**) **** represents a significant difference p < 0.0001, and 'ns' represents no significant difference. In (**h**) **** represents p < 0.0001 and *** represents p = 0.0008. Schematic illustration in (**a**) was prepared by scientific illustrator Leonora Martínez Nuñez.

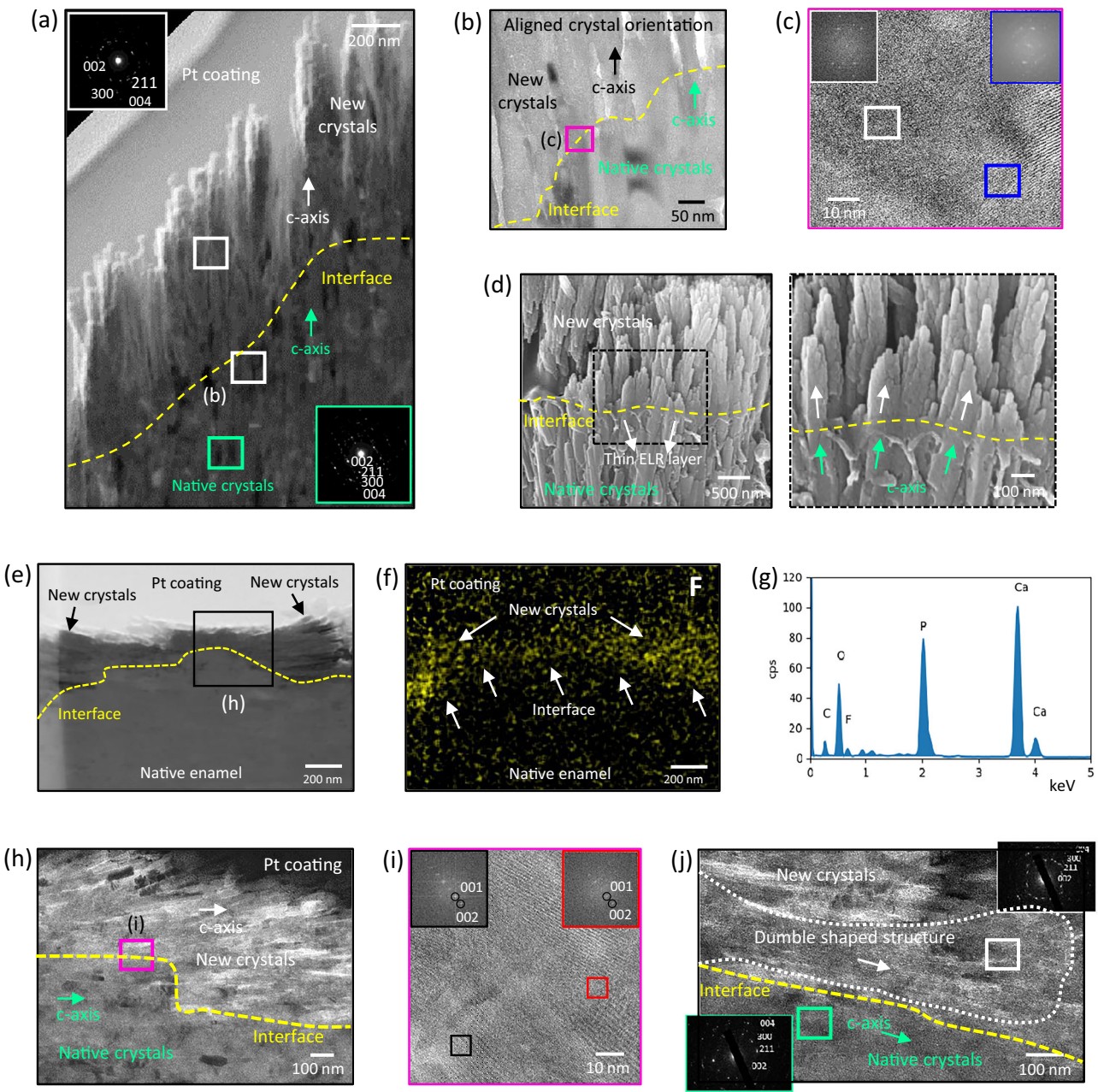

**Fig. 4 | Crystallographic integration of the mineralised layer to the underlying enamel tissue. a** STEM and (**b**) TEM images showing integration between native and newly grown nanocrystals of a remineralised diazone prism. Insets in (**a**) show SAED patterns from native and remineralised regions confirming newly grown nanocrystals co-oriented along the *c*-axis with respect to native nanocrystals. **c** HRTEM image of the interface from (**b**) showing no distinct boundary between native and new nanocrystals, demonstrating crystallographic integration via epitaxial remineralisation. Micrographs in (**a**–**c**) are representative images from *n* = 2 independent experiments. **d** High magnification SEM images showing new nanocrystals nucleating and extending from underlying native nanocrystals along the *c*-axis (representative images from *n* = 4 independent experiments). **e** TEM image, (**f**) EDX map, and (**g**) EDX spectrum confirming and distinctly showing the growth of

remineralised layer on parazone prism. **h** TEM image showing integration and co-alignment between native and newly grown nanocrystals along the *c*-axis in a remineralised parazone prism. **i** HRTEM image of the interface from (**h**) showing no distinct boundary between native and new nanocrystals, confirming crystallographic integration. Insets show FFT patterns from different regions, demonstrating co-orientation. **j** TEM image showing a bundle of newly grown nanocrystals (marked by a white dotted boundary) nucleating on the *a*-axis of the native nanocrystals while growing and extending along the *c*-axis. Insets in (**j**) show SAED patterns recorded from remineralised (white box) and native (green box) regions, confirming the co-orientation between native and newly grown nanocrystals. Micrographs in (**e**–**j**) are representative images from *n* = 2 independent experiments.

imitates the structural orientation of the MCFs at the DEJ[60,64] and shines light on the mineralising mechanisms that may take place during early enamel development. These results demonstrate the capacity of our ELR matrix to grow an aprismatic enamel-like layer from exposed dentine while generating a similar architecture to that of the DEJ observed in early amelogenesis, which to the best of our knowledge has not been achieved before.

## Mechanical and microtribological characterisation of mineralised dentine

The new aprismatic enamel-like layer exhibited comparable E (58.3 ± 16.7 GPa) and H (1.4 ± 0.3 GPa) to native enamel (*E* = 50 - 90 GPa, H = 2.5 - 4 GPa) (Fig. 5f)[9,10]. The lower H value for the remineralised layer compared to native enamel is attributed to the difference in the orientation of the nanocrystals on native enamel and on remineralised

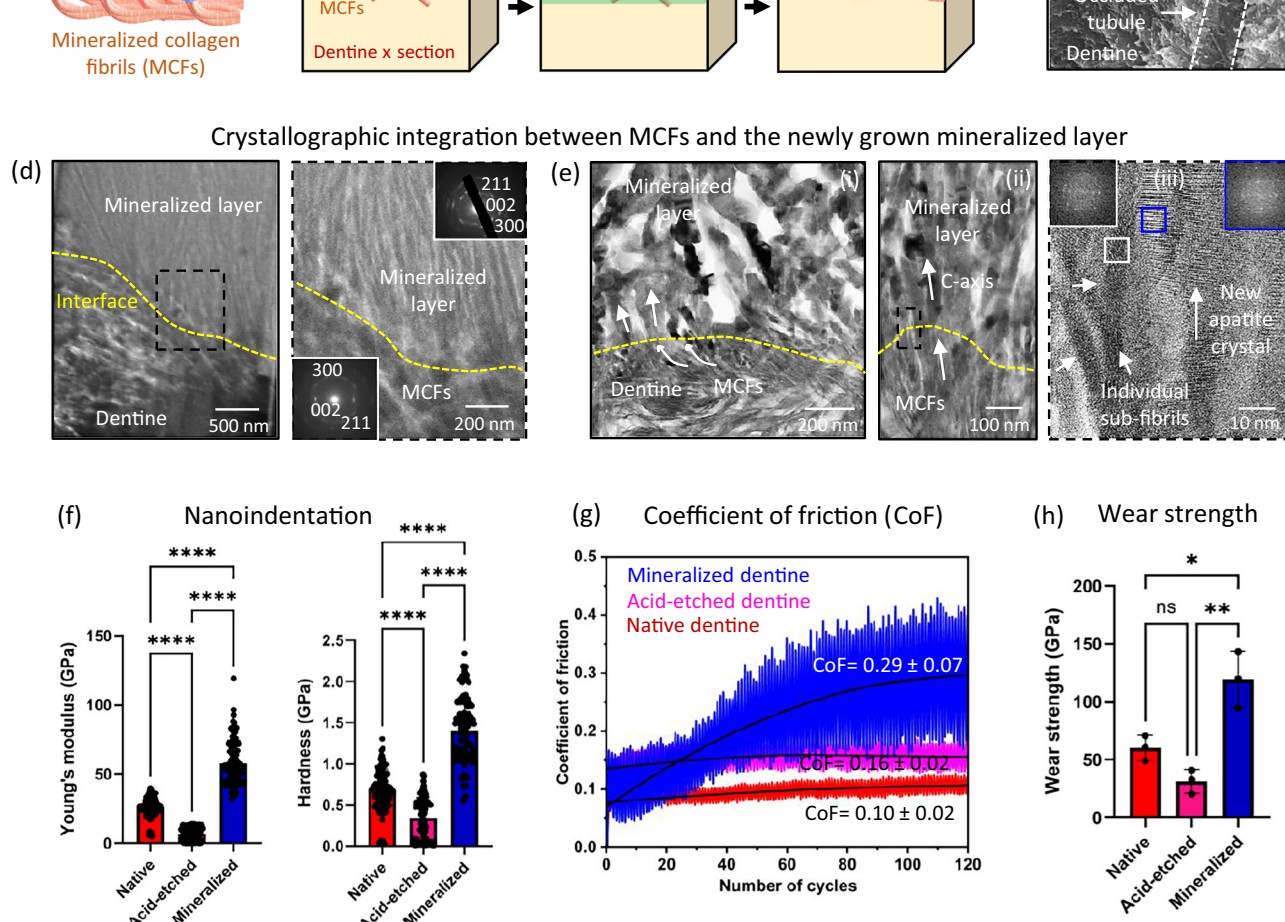

**Fig. 5 | Mineralisation of dentine surface enhances mechanical and micro-tribological properties.** Illustrations of (**a**) the natural dentine-enamel junction (DEJ) with integrated collagen fibrils and enamel crystals and (**b**) the ELR coating on dentine surface and its mineralisation. **c** Cross-sectional SEM image showing a mineralised layer grown on top of exposed dentine, including mineralisation within the dentinal tubules as a result of ELR penetration (results not shown) (representative micrograph from $n = 5$ independent experiments). **d** Low magnification TEM image showing the cross-section of the mineralised layer-dentine interface. Insets show SAED patterns from the remineralised layer and native dentine. **e** TEM images (i) show integration between the mineralised layer and MCFs of dentine. (ii) MCFs undergo re-direction and orient parallel to the mineralising nanocrystals. (iii) HRTEM image showing crystallographic integration between MCFs and newly grown nanocrystals, while insets show FFT patterns demonstrating comparable orientation between them. **f** Young's modulus (E) and hardness (H) of native, acid-etched, and remineralised enamel. Bars represent the mean values of E and H while dot plot represents indentation measurements compiled from 3 independent experiments for each group. **g** Coefficient of Friction, and (**h**) Wear strength of native, acid-etched, and remineralised enamel samples ($n = 3$ samples for each group). Data are presented as mean ± SD. Statistical significance was analysed using two-sided one-way ANOVA (Tukey test) in GraphPad Prism ver. 10. In (**f**) **** represents significant difference $p < 0.0001$. In (**h**) ** represents $p = 0.0015$, * represents $p = 0.0112$, and 'ns' represents no significant difference. Schematic illustrations in (**a**, **b**) were prepared by scientific illustrator Leonora Martínez Nuñez.

dentine, as explained in Supplementary Discussion 8. In addition, CoF (0.29 ± 0.07) and WS (119.1 ± 24.3 GPa) were also comparable to native enamel (CoF = 0.1–0.4, WS = 153.9 ± 3.2 GPa) (Fig. 5g, h)[65], indicating similar microtribological properties. Overall, the results demonstrate the capacity of our ELR matrix to grow an integrated aprismatic enamel-like layer over exposed dentine.

## Translation: toothbrushing, chewing and grinding, fracture toughness, and acid attack

To assess the potential clinical applicability of our mineralising technology, we explored the stability of the remineralised layer under conditions that emulate daily physical and chemical insults to enamel.

First, evaluating the effects of abrasion from routine toothbrushing, we exposed 1 mm thick sections of prismatic enamel to continuous brushing for 60 minutes using an electric toothbrush (Fig. 6a) to simulate toothbrushing for about one year, using a reported protocol[66] (Supplementary Fig. 33a, b). The remineralised enamel did not exhibit visual microstructure loss (Supplementary Fig. 33c) and displayed similar E (68.6 ± 16.4 GPa) and H (2.76 ± 0.8 GPa) after treatment compared to those of native enamel (E = 70.2 ± 15.2 GPa, H = 3.0 ± 0.9 GPa) (Fig. 6b).

Second, we evaluated resistance to fracture from physical insults by applying a 9.8 N indentation force and measuring the apparent fracture toughness ($K_{c(app)}$) (Fig. 6c) following standard protocols[67].

Fracture toughness is considered a critical indicator of enamel quality[68]. Remineralisation of acid-etched enamel increased $K_{c(app)}$ from $0.7 \pm 0.1$ MPa.m$^{0.5}$ to $1.1 \pm 0.1$ MPa.m$^{0.5}$, regaining values similar to those of healthy enamel ($1.2 \pm 0.2$ MPa.m$^{0.5}$) (Fig. 6d and Supplementary Fig. 34)[67].

Third, to evaluate tooth wear due to abrasive challenges such as chewing and grinding, we exposed enamel and dentine samples to 75 N of frictional forces for 2 weeks, simulating mastication for ~3.5 years (Fig. 6e and Supplementary Fig. 35) using a clinically relevant technique[69]. The remineralized enamel exhibited similar wear in terms of volume loss ($V_L = 3.5 \pm 1.4$ mm$^3$) and height loss ($H_L = 30.5 \pm 8.9$ μm) compared to native enamel ($V_L = 2.9 \pm 1.0$ mm$^3$, $H_L = 29.1 \pm 9.3$ μm) (Fig. 6f). Interestingly, surface mineralised dentine exhibited significantly less wear ($V_L = 1.6 \pm 0.6$ mm$^3$, $H_L = 16.8 \pm 6.7$ μm) compared to native dentine ($V_L = 2.5 \pm 0.9$ mm$^3$, $H_L = 28.0 \pm 10.3$ μm) (Fig. 6g), indicating the protective role of the enamel-like mineral layer on dentine. Finally, to investigate stability against acid erosion resulting from dietary habits (Fig. 6h), we exposed tooth samples to 0.1 M acetic acid at pH 4.0[70] for either 15 min or 2 days to emulate short and prolonged erosion, respectively. At 15 minutes, both E and H of native ($E = 16.6 \pm 5.3$ GPa, $H = 1.0 \pm 0.2$ GPa) and remineralized ($E = 35.0 \pm 6.4$ GPa, $H = 1.6 \pm 0.4$ GPa) enamel reduced significantly but the remineralized enamel exhibited higher stability (Fig. 6i). Similarly, at 2 days, E and H were further reduced for both native ($E = 3.2 \pm 1.2$ GPa, $H = 0.14 \pm 0.06$ GPa) and remineralized ($E = 6.2 \pm 0.8$ GPa, $H = 0.15 \pm 0.05$ GPa) enamel, again with the remineralized enamel exhibiting higher mechanical stability (Fig. 6i) and structural integrity (Fig. 6j). This enhanced chemical stability is attributed to the presence of fluorine[71]. We also tested the chemical stability of the mineralised layer grown on dentine against acid erosion. At 15 minutes of acid erosion, both E and H of the remineralised dentine reduced significantly ($E = 12.4 \pm 4.2$ GPa, $H = 0.6 \pm 0.2$ GPa) and were comparable to acid-treated enamel ($E = 16.6 \pm 5.3$ GPa, $H = 1.0 \pm 0.2$ GPa) (Supplementary Fig. 36).

Overall, these results demonstrate the robustness and potential applicability of the ELR matrix and its capacity to grow a remineralised layer exhibiting similar or higher resistance to abrasion, fracture, and acid erosion compared to native enamel. In addition, to further facilitate translation, similar experiments were conducted but now preparing the 5% w/v ELR matrix in ethanol/water mixture at a 9/1 ratio in the presence of 1.5 mM Ca$^{2+}$ ions and using 1.5% v/v glutaraldehyde as crosslinker. Both ethanol and glutaraldehyde are commonly used in Food and Drug Administration (FDA) approved dental products. ELR coatings exhibited high in vitro cytocompatibility when tested against three different cells lines including mouse fibroblast cell line 3T3, human immortalised mesenchymal stem cells (MSCs), and human umbilical vein endothelial cells (HUVECs) (Supplementary Figs. 37–40). Furthermore, the ethanol/water solvent enhances penetration of the ELR solution into the tissue (Supplementary Fig. 41)[72] and allows a rapid (3-4 minutes) ELR matrix assembly of up to 10 μm thick under ambient conditions (Supplementary Fig. 42). Using these reagents, the ELR matrix also triggered similar fibrillar ELR structures (Supplementary Fig. 43a), secondary structure composition (Supplementary Figs. 43b, 44, Supplementary Discussion 9), recreation of enamel microstructure (Supplementary Fig. 43c, d), and restoration of the mechanical properties (Supplementary Fig. 43e, f).

To facilitate translation of the technology, we have conducted the experiments following steps that are commonly used in dental practices including: (i) prophylaxis using abrasive paste to remove the pellicle layer, (ii) acid etching using a 37% phosphoric acid gel for 30 seconds, (iii) washing with water to remove debris, and (iv) air drying the tooth surface for 30 s. Afterwards, the ELR solution was deposited and air dried on the tooth surface in 3-4 min. To imitate the real oral environment, ELR-coated samples were mineralised using modified-artificial saliva (m-AS) under dynamic and abrasive conditions. Our results demonstrated the recreation of both diazone and parazone regions of enamel over large surface areas (Supplementary Fig. 45). Furthermore, to test the capacity to mineralise in natural human saliva, ELR-coated enamel samples were exposed to saliva from three different donors for 2 weeks. Again, the results revealed the capacity of the ELR matrix to recreate prismatic enamel over large surface areas (Fig. 6j and Supplementary Fig. 46a) and restore both the mechanical ($E = 74.6 \pm 15.3$ GPa, $H = 2.7 \pm 1.1$ GPa) (Fig. 6k and Supplementary Fig. 46c) and chemical (Supplementary Fig. 46d, e) properties of natural enamel. These results confirm the robust mineralising capacity of our ELR platform by enabling a similar epitaxial growth despite the inherent compositional complexity of natural saliva. Furthermore, the structure and properties of enamel were recovered on all tested samples, independently of variations in free Ca$^{2+}$ ion concentrations in the saliva from each donor (Supplementary Fig. 46b). In addition, we assessed the stability of the ELR coatings on enamel by exposing them to extreme pH (2, 11), salt solutions (5 M sodium chloride and sodium acetate), extensive tooth brushing (equivalent to ~ 4 months), sonication (37 kHz, 5 minutes), high temperature (55 °C for 2 h), and a standard tape peeling test[23]. These results revealed no apparent effects on the surface morphology (Fig. 6l, m and Supplementary Figs. 47, 48, 49c, 51), mechanical properties (Supplementary Fig. 49a, b), and functionality (i.e., ability to mineralise) (Supplementary Figs. 47, 48) of the ELR coating. This stability and resilience of the ELR coating may result from the ordered β-sheet conformations[34] and strong chemical linkages between amino acids mediated by glutaraldehyde crosslinking[73]. Furthermore, the mechanical properties of the ELR coating on enamel ($E = 6.02 \pm 1.68$ GPa and $H = 0.28 \pm 0.06$ GPa) were comparable to those of the commercially available Duraphat® varnish ($E = 3.88 \pm 0.92$ GPa and $H = 0.30 \pm 0.08$ GPa) (Supplementary Fig. 52), further supporting potential viability for translation. These results confirmed the capacity of the technology to deposit a uniform, resilient, and functional ELR coating over large areas and convoluted tooth structures in 3-4 min.

In this work, we engineered a practical and robust supramolecular fibril-rich ELR matrix that imitates some of the structural and functional characteristics of the enamel-developing matrix to trigger epitaxial and organised growth of apatite nanocrystals into an integrated and functional tissue (Supplementary Discussion 10). We demonstrated the use of this ELR matrix as versatile coatings capable of recreating the microarchitecture and restoring the key functional properties of the different anatomical regions of healthy enamel, including prismatic (i.e., parazone, diazone, inter-prismatic) and aprismatic regions from the surface of teeth, irrespective of the level of erosion down to bare dentine (Supplementary Discussion 11). We confirmed the performance of this remineralised tissue by testing a collection of key functional properties, including integration to the underlying tissue, stiffness, hardness, coefficient of friction, wear strength, and stability to extensive physiological use, including abrasion from toothbrushing and chewing and grinding, fracture due to grinding, and erosion due to exposure to acidic solutions. Our mineralising technology offers a practical and clinically friendly solution to remineralise thin (up to ~ 10 μm) layers of lost enamel in a manner that regains both structure and function of healthy enamel tissue. This capability far surpasses current commercial dental remineralisation alternatives and overcomes key translational obstacles that have so far prevented the capacity to regrow enamel tissue in patients, even as thin functional layers. Moreover, our findings demonstrate a capacity to grow enamel-like structures under conditions that closely imitate various mechanical and chemical challenges found in the mouth. However, these tests do not fully recreate the complexity of the in vivo oral environment and, consequently, to fully confirm the capacity to regenerate natural enamel would require in vivo validation, which we envision to pursue in future work. In conclusion, we envisage that our approach holds great potential for

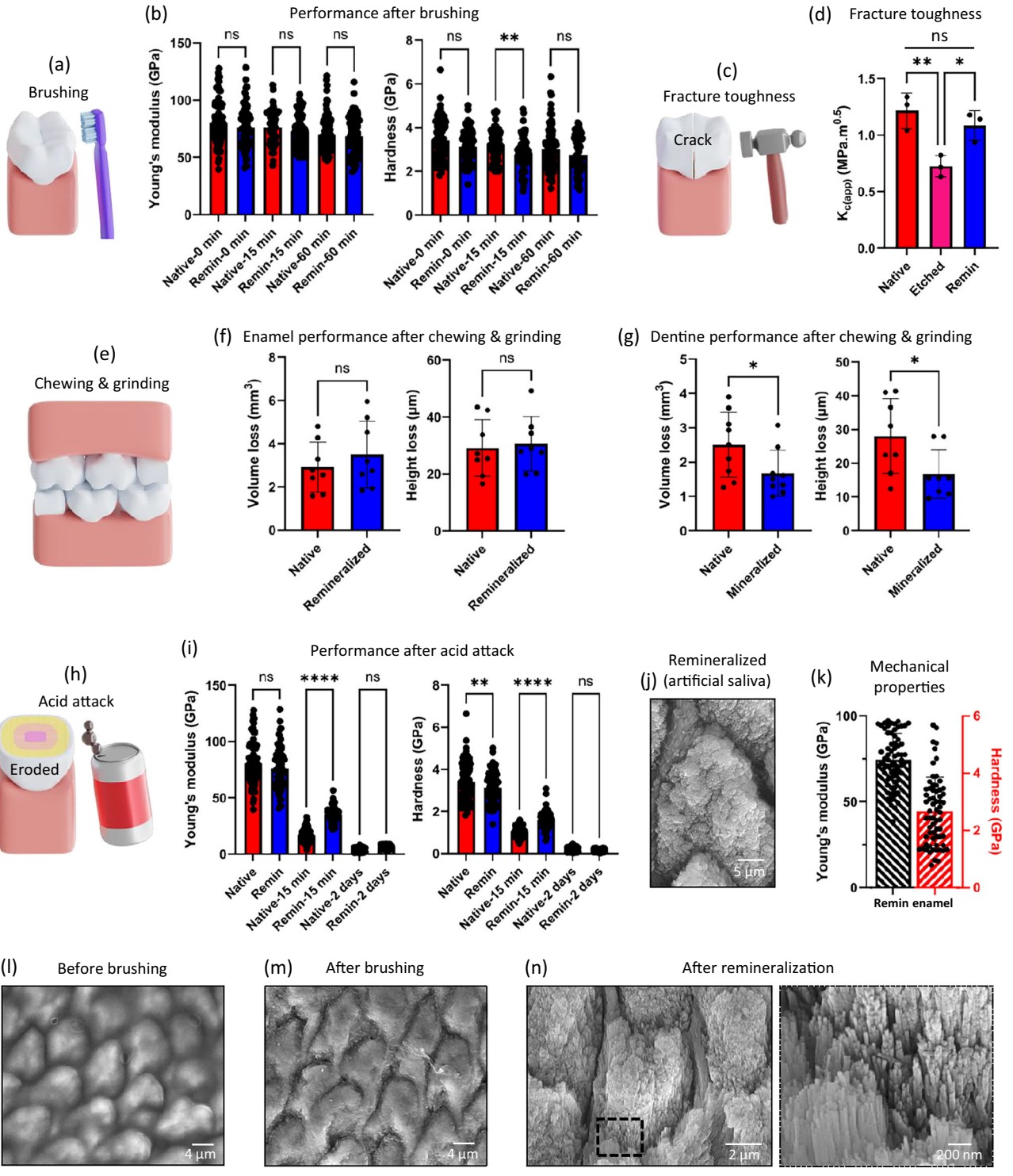

use in clinical settings and opens exciting opportunities for the design of innovative bioinspired biomineralizing materials.

## Methods

### Materials
All materials used in this study were purchased from Sigma-Aldrich, UK, unless specified. Chemicals were used as received without any further processing. Human molar teeth extracted for clinical reasons were used in this study following approval from the Research Ethics Committee, Faculty of Medicine & Health Sciences, University of Nottingham (reference number: FMHS 313-0721). Informed consent was obtained from all donors prior to tooth extraction, with full protection of donor's privacy.

### Fabrication and characterisation of ELR fibrils
ELR molecules (1 or 5%, w/v) (Technical Proteins Nanobiotechnology, Valladolid, Spain) were either dissolved in the mixture of anhydrous dimethyformamide (DMF) and dimethyl sulfoxide (DMSO) solvents at 9/1 ratio or in the ethanol-water mixture at 9/1 ratio followed by the addition of $Ca^{2+}$ ions at a range of concentrations between 0–6.8 mM.

**Fig. 6 | Assessment of performance after common physical and chemical insults. a** Illustration of enamel abrasion due to routine tooth brushing and (**b**) Young's modulus (E) and hardness (H) of native and remineralised (remin) enamel sections after 15 and 60 min of continuous brushing. **c** Illustration of tooth fracture due to physical insults and (**d**) apparent fracture toughness ($K_{c(app)}$) of different enamel samples ($n = 3$ samples). **e** Illustration of tooth wear due to chewing & grinding or other physical insults and volume and height loss evaluated in native and mineralised (**f**) enamel ($n = 8$ samples) and (**g**) dentine samples ($n = 8$ samples) after the chewing & grinding experiment. **h** Illustration of enamel erosion due to acid attack and (**i**) nanoindentation results (E and H) of native and remineralised enamel samples before and after exposure to acid for 15 min and 2 days. **j** SEM image ($n = 3$ samples) and (**k**) nanoindentation results (E and H) of enamel remineralised for 14 days using natural human saliva. SEM images showing ELR coating on prismatic enamel (**l**) before brushing, (**m**) after brushing for 15 min (equivalent

to ~4 months of brushing), and (**n**) after remineralisation for 10 days ($n = 3$ samples). The image presented in (**l**) appears blurry due to a visual effect created by the ELR coating uniformly conforming over the enamel surfaces. In (**b, i, k**) bars represent the mean values of E and H while dot plot represents indentation measurements compiled from 3 independent experiments for each group. Data are presented as mean ± SD. Statistical significance was analysed using two-tailed Student's $t$ test (**f, g**) or two-sided one-way ANOVA (Tukey test) (**b, d, i**) in GraphPad Prism ver. 10. In (**b**) ** represents a significant difference $p = 0.0055$. In (**d**) ** represents significant difference $p = 0.0083$, * represents $p = 0.0328$. In (**g**) * in volume loss represents $p = 0.0480$ while * in height loss represents $p = 0.0309$. In (**i**) **** represents $p < 0.0001$, ** represents $p = 0.0035$. In (**b, d, f, i**), 'ns' represents no significant difference. Schematic illustrations in (**a, c, e, h**) were prepared by scientific illustrator Leonora Martínez Nuñez.

## Attenuated total reflection-Fourier-transform infrared (ATR-FTIR) spectroscopy analysis

A 3 µL of each ELR-Ca solution was dropped on the crystal of ATR-FTIR (Cary 630, Agilent, USA) to record spectra between wavenumber 4000 to 400 $cm^{-1}$ at an average of 64 scans and 2 $cm^{-1}$ resolution at 25 °C. Secondary structure composition (α-helix, β-sheet, β-turn, and random coil) of ELR membranes was analysed by deconvoluting the amide III region (1350–1200 $cm^{-1}$)[31,74] using OriginPro 8.5 software.

To see the effect of drying on ELR's secondary structure composition, a 3 µL drop of ELR-Ca solution was placed on an ATR-FTIR crystal under controlled humidity (<20%) and allowed to dry while recording spectra at different time intervals at 25 °C. To see the effect of simultaneous drying and crosslinking, hexamethylene diisocyanate (HDI) was added to ELR-Ca solution[31]. A 3 µL drop of this ELR-Ca-HDI solution mixture was placed on an ATR-FTIR crystal and allowed to simultaneously dry and crosslink while recording spectra at different time intervals.

To assess the role of $Ca^{2+}$ ions present in the enamel nanocrystals on the secondary structure conformation of the ELR coating, we drop casted 5% w/v ELR solution (DMF/DMSO = 9/1 and 0.56% HDI v/v) without and with 1.5 mM $Ca^{2+}$ ions on the enamel surface and left overnight drying in the glovebox under controlled humidity (<20%) at 25 °C. The secondary structure conformation of the ELR coatings was analysed using ATR-FTIR.

## Scanning electron microscopy (SEM) analysis

For SEM analysis, a drop of 1% w/v ELR solution without and with 1.5 mM $CaCl_2$ solution was placed on cleaned silica surface samples and dried at room temperature (25 °C), sputter coated with 10 nm thick iridium coating (Quorum, Model: 150 T ES, UK), and analysed under SEM at a working distance of 10 mm and operated at an accelerating voltage of 15 kV (JEOL, 7100 F).

## Transmission electron microscopy (TEM) analysis of ELR fibrils

Similarly, for TEM analysis, a drop of 1% w/v ELR solution without and with 1.5 mM $CaCl_2$ solution was placed on amorphous carbon on 200 mesh Cu grid and the excess solution was removed using filter paper. The grid was then dried overnight under controlled humidity (<20%) and analysed without staining under a JEOL JEM-2100F FEGTEM instrument operated at 200 kV. Images were recorded using Gatan Orius camera, and then subsequently by Gatan K3-IS detector and were analysed using Gatan Microscopy Suite® (GMS 3) software.

## TEM analysis of the mineralised ELR fibrils

For TEM analysis of the mineralised ELR fibrils, a drop of 5% w/v ELR solution containing 1.5 mM $CaCl_2$ and 0.56% v/v hexamethylene diisocyanate (HDI) crosslinker was placed on TEM grid and the excess solution was removed using filter paper. The TEM grid was dried overnight at 25 °C before incubating in the mineralisation solution supersaturated with respect to fluorapatite (prepared by dissolving

2 mM hydroxyapatite powder and 2 mM sodium fluoride)[31] for 24 h. Samples were then air-dried and analysed under TEM without any staining. Selected area electron diffraction (SAED) was performed to confirm the crystalline phases in the samples. Obtained TEM images were analysed for fast Fourier-transform (FFT) using ImageJ (National Institute of Health, USA) software.

## Mineralisation of apatite nanocrystals embedded within the ELR matrix

To confirm the ELR matrix mediated crystal extension preferentially along the $c$-axis, we added commercially available HAp nanocrystals to the ELR solution (5% w/v ELR, 1.5 mM $Ca^{2+}$ ions, and 0.56% v/v HDI) and drop casted it on TEM grids while removing any excess solution using filter paper. The TEM grid was dried at 25 °C before incubating in the mineralisation solution supersaturated with respect to fluorapatite[31] for 24 h. For the control sample without the ELR matrix, apatite nanocrystals were added to the mineralisation solution for 24 h before placing on the TEM grids. Samples were then air-dried and analysed under TEM without any staining.

## X-ray photoelectron spectroscopy (XPS) analysis

A 25 µL drop of ELR-Ca (1.5 mM $CaCl_2$) solution was placed and dried on a cleaned silicon wafer and analysed using Phi 5600 XPS (Perkin Elmer, U.S.). Monochromatic Al Kα X-ray source operated at 350 W was used to measure survey and high-resolution C1s and N1s spectra. Atomic percentages of each element from survey spectra were identified using CasaXPS processing software. OriginPro 8.5 software was used to deconvolute the high-resolution Ca 2p3/2 and O1s spectra.

## Isothermal titration calorimetry (ITC) analysis

The interactions between ELR molecules and $Ca^{2+}$ ions were analysed using MicroCal PEAQ-ITC (Malvern Instruments). Solutions were prepared in deionised water to minimise the dilution heat contribution and degassed prior to experiments. ELR concentration was fixed at 100 µM and was titrated with 1 mM $Ca^{2+}$ solution. Experiments were carried out at 25 °C and consisted of 18 successive injections of 2 µL $Ca^{2+}$ solution and were continuously stirred at 750 rpm to ensure proper mixing.

## Small angle X-ray scattering (SAXS) and wide angle X-ray scattering (WAXS) analysis

SAXS was conducted at beamline B21 Diamond Light Source Ltd., Didcot, UK[75]. ELR-Ca (1% ELR w/v; 1.5 mM $CaCl_2$) solution was loaded into a PEI capillary, dried, and put into a multipurpose sample (MPS) cell[76]. 1% w/v ELR (i.e., 10 mg/mL) concentration was used based on a previous study[77] that reported the optimum protein concentration to be in the range 1 to 10 mg/mL for capturing reliable scattering signals. Data was collected for 1 s exposure time, 21 frames at room temperature. Samples were averaged and subtracted to remove frames from the averaging that had radiation damage. For WAXS analysis, ELR-Ca

(1% ELR w/v; 1.5 mM Ca²⁺ ions) samples were dried in PEI capillaries, loaded into a capillary rack, and measured under vacuum at room temperature using Excillium Gallium Metaljet source, at an energy of 9.2 KeV. WAXS data were collected using an offline instrument DL-SAXS (P38) (Xeuss 3.0, Xenocs) using an Eiger 2 R 1 M detector, measuring between $Q = 0.34 – 2.74$ Å$^{-1}$.

## Dynamic light scattering (DLS) analysis
Dynamic light scattering (DLS) experiments were carried out using a Zetasizer nano-ZS90 (Malvern, Inc.) instrument at room temperature (25 °C). ELR solutions (1 mg/mL) without and with 1.5 mM Ca²⁺ ions and 0.1% glutaraldehyde were prepared in ethanol/water (9/1) solvent mixture and analysed using a laser wavelength of 633 nm and a scattering angle of 90°.

## Coarse-grained simulation
AlphaFold[78] was used to generate an initial atomistic protein structure which with coarse-grained using the martinize with vermouth script for the MARTINI 3 forcefield[79]. All simulations were performed using the GROMACS software package[80,81] using a timestep of 20 fs. Velocity rescaling was used to set the temperature to 298.15 K with the protein and non-protein groups coupled separately[82] and a Berendsen barostat used to control the isotropic pressure at 1 bar[83]. Bond lengths within aromatic side chains and backbone side-chain bonds for Ile and Val was constrained using the LINCS algorithm[84]. Short range intermolecular interactions were evaluated using a Lennard-Jones algorithm with a shifted cut off at 1.1 nm and electrostatic interactions evaluated using the reaction-field algorithm with a cut off also at 1.1 nm[85]. All simulation were performed in aqueous conditions using the MARTINI small water (two water molecules to one bead) model with an ionic concentration Ca²⁺ ions of - 0.2 M with Cl¹⁻ serving as the counter ions for the Ca²⁺ ions and protein net charge. All systems were minimised using the steepest descent integrator a tolerance of 100 kJ mol⁻¹ nm⁻¹.

## Quantification of β-rich fibrillar regions in ELR membranes
ELR membranes were fabricated using the procedure described previously [1, 14] after slight modifications. Briefly, ELR solutions were prepared by dissolving 5% w/v ELR molecules in DMF/DMSO (9/1 ratio) solvent mixture, followed by addition of Ca²⁺ ions (0 to 6.8 mM Ca²⁺ ions prepared using CaCl₂.2H₂O) at room temperature inside a humidity controlled (<20%) glovebox (BELLE Technology, UK). HDI crosslinker was added to the above ELR-Ca solutions, drop-casted on top of polydimethylsiloxane (PDMS) substrate, and dried overnight to get crosslinked ELR membranes[31,86]. Further, ELR membranes were embedded in paraffin wax, sectioned to about 3 μm using cryostat microtome (Leica CM3050), and stained with Thioflavin T (ThT, 100 μM in water)[87] and Congo Red[88] dyes, normally used to identify β-rich fibrillar structures.

## Preparation of enamel and dentine sections
Human tooth samples were sectioned (0.5 - 1.0 mm thick) perpendicular to the long axis of the tooth using water cooled diamond saw. Sections were acid-etched using 37 wt% orthophosphoric acid (H₃PO₄) for 30 seconds to imitate enamel erosion in the prismatic (inner) enamel[27]. To partially erode aprismatic (outer) enamel from the tooth surface, the whole tooth samples were subjected to 37 wt% H₃PO₄ and etched for 5 minutes and were later sectioned as described above. Dentine sections were etched for 10 s using 37 wt% H₃PO₄. Acid-etched enamel and dentine sections were washed thrice with de-ionised water, sonicated for 10 min in water bath to remove any residual contaminants, air dried, and stored at 4 °C until used.

## ELR coatings on enamel and dentine sections
To prepare ELR coatings using the DMF/DMSO system, 5% w/v ELR solution (DMF/DMSO = 9/1) containing 1.5 mM Ca²⁺ ions and 0.56% v/v

HDI was drop-casted on acid-etched prismatic, aprismatic enamel, and dentine sections and left to dry and crosslink for 1.5 h in a humidity-controlled environment at room temperature (25 °C). To prepare ELR coating using an ethanol/water system, 5% w/v ELR solution (85% ethanol and 15% water) containing 1.5 mM Ca²⁺ ions and 1.5% v/v glutaraldehyde was drop-casted on acid-etched enamel sections. To prepare membranes, ELR solution was drop-casted on PDMS substrate, and dried overnight to get crosslinked ELR membranes. To differentiate between native and remineralised regions of enamel on the same tooth sample, we used a nail varnish approach[27] where half of the enamel surface was covered with nail varnish and the remaining half was coated with a 5 μm thick ELR coating. Thickness of the ELR coating can be controlled by regulating the volume of ELR solution placed on enamel surface. To demonstrate this, we created 2 mm X 4 mm windows of enamel sections, drop casted different volume of ELR solution varying from 2 μL, 5 μL, and 10 μL (5 μL x2 applications to prevent solution overflow) and measured ELR coating thickness under SEM.

## Mineralisation experiment
ELR coated enamel and dentine sections were subjected to mineralisation using solution prepared by dissolving hydroxyapatite powder (2 mM) and sodium fluoride (2 mM) under acidic pH until complete dissolution[31]. The pH of the solution was later readjusted to 6.0 using 30% NH₄OH solution. To mimic physiological mineralisation conditions, enamel sections were subjected to artificial saliva (AS)[89] prepared by mixing 1.2 mM CaCl₂·2H₂O, 50 mM HEPES buffer, 0.72 mM KH₂PO₄, 16 mM KCl, 4.5 mM NH₄Cl, 0.2 mM MgCl₂·6H₂O, 1 ppm NaF, and pH adjusted to 7.2 using 1 M KOH. Mineralisation experiments were carried out at 37 °C inside a temperature-controlled incubator (LTE Scientific, Oldham, UK). To imitate the oral environment, we prepared modified-artificial saliva (m-AS) by supplementing AS with different salivary enzymes such as α-amylase (100 μ/mL), lysozyme (0.75 μ/mL)[90], and proteinase K (15 ng/mL)[91]. Proteinase K is a serine protease and was used as an alternative to proteinase 3, which is a widely known salivary protease in humans[91]. To imitate the dynamic and abrasive oral environment, we incubated the ELR coated enamel sections in the m-AS along with 5 steel balls of 2 mm diameter inside the 20 mL glass scintillation vials and stirred at 1000 RPM and 37 °C[23]. We used steel balls to imitate external insults that may come food particles or tooth attrition, biting forces, etc. Post mineralisation, enamel sections were treated for enzymatic digestion (elastase, 15 μ/ml for 10 days at 37 °C)[31] to remove ELR coating and to expose apatite crystals. Enamel sections were later washed several times with de-ionised water, sonicated for 5 minutes to remove debris, and air dried.

## Mineralisation using natural saliva
Human saliva was collected from 3 different healthy donors after approval by the University of Nottingham Ethics Committee, reference number FMHS 313-0721. The donors were between 20 and 40 years old and had healthy periodontal conditions with no signs of gingivitis, periodontitis, gingival recession, carious decay, or dental erosion. Donors were instructed to refrain from eating or drinking (except water) for two hours before saliva collection. Unstimulated saliva samples were collected directly into polypropylene tubes. For enamel remineralization, ELR coated enamel sections were separately exposed to 1 mL of whole human saliva collected individually from the different donors for 2 weeks. Enamel sections were incubated at 37 °C, and the saliva was replenished every 3 h during a 12 h daytime period and then left in the same spent saliva for the subsequent 12 h overnight. Post mineralisation, ELR coating was enzymatically degraded using elastase as described in the section 'Mineralisation experiment'. Calcium ion concentrations in the saliva samples were measured using a colorimetric assay kit (Abcam, ab102505), according to the manufacturer's instructions.

## Swelling analysis

The swelling behaviour of the ELR membranes fabricated using ethanol/water (85/15) was investigated using a previously reported methodology[92]. Briefly, dried ELR membranes of known initial weights ($W_i$) were soaked in PBS (pH 6.0) overnight at 37 °C, then carefully removed and immediately weighed ($W_f$) after removing excess PBS using filter paper. The experiment consisted of three samples ($n = 3$). The swelling percentage was calculated using the expression below:

$$\text{Swelling (\%)} = \frac{W_f - W_i}{W_f} \times 100 \tag{1}$$

## Degradation analysis

Degradation studies were performed by immersing the ELR membranes in 1 mL of artificial saliva without and with proteinase K (15 ng/mL) at 37 °C for different times. At each time point, 50 μL of the artificial saliva was removed and analysed for ELR release resulting from the membrane degradation due to mineralisation and enzymatic degradation. The amount of ELR degradation was analysed using the Micro BCA Protein Assay Kit (Thermo Scientific Pierce) by comparing the results to a standard curve generated with varying concentrations of ELR molecules.

## Thermogravimetric analysis (TGA)

TGA analysis was carried out using TGA Q500 V20.13 (TA instruments) using a heating rate of 10 °C/min to 600 °C.

## Profilometer analysis

The ELR coating on the enamel sample was prepared using the ethanol/water solvent mixture as described in section 'Mineralisation experiment'. Briefly, 5% (w/v) ELR solution was prepared in ethanol/water (75/25) solvent mixture followed by the addition of 1.5 mM Ca$^{2+}$ ions and 0.5% (v/v) glutaraldehyde. The ELR solution was deposited on an acid etched enamel sample and allowed to dry for 4 minutes. For profilometer analysis, a long scratch mark was made on the ELR coating at the centre of the sample. A height profile was created along the scratch mark using profilometer Alicona G5 XL optical profilometer (Bruker, Germany) before and after brushing for 5 min.

## SEM and energy-dispersive X-ray (EDX) spectroscopy

Native, acid-etched, and remineralized enamel sections were mounted on aluminium stubs and coated with 10 nm thick Iridium coating to visualise enamel microstructure under SEM (operated at an accelerating voltage of 15 kV). To analyse elemental composition of native and remineralized enamel, samples were coated with 20 nm thick layer of carbon instead of Iridium (Quorum UK, Model: 150 R ES) and analysed under EDX (Oxford Instruments Aztec EDX system) at an accelerating voltage of 15 kV.

## Focused ion beam (FIB) lift out and TEM analysis

To confirm crystallographic integration between newly grown and native enamel nanocrystals, thin lamella from remineralized enamel sections from both diazone and parazone regions were prepared using standard FIB procedure[93] on Zeiss CrossBeam 550 FIB-SEM and Helios 450S FIB-SEM (FEI). Lift out lamellas were analysed using Titan 60-300 TEM (FEI) equipped with a high-brightness field-emission gun (X-FEG), monochromator, image-side Cs-corrector (CEOS) and JEOL JEM-2100F FEGTEM instruments, operated at 300 and 200 kV, respectively.

## Atomic force microscopy (AFM)

Remineralized enamel samples were scanned under Dimension FastScan AFM (Bruker) at 20 °C and 60% relative humidity using AFM probe of 8 nm tip radius at 1024 force curves/line and scan rate of 0.5 Hz. Images were analysed using Gwyddion software obtained from GNU General Public License, as described previously[94].

## X-ray diffraction (XRD)

XRD analysis was performed using D8 ADVANCE with DAVINCI to analyse the phase composition of the native and remineralized enamel sections at room temperature. Instrument was operated with flat plate θ/θ geometry and Ni-filtered Cu-Kα radiation at 45 kV and 40 mA (Kα1 = 1.54059 Å, Kα2 = 1.54442 Å)[31]. The samples were recorded at 2θ values ranging from 5 to 60° with a step size 0.02° and were compared with standard fluorapatite PDF #73−1727.

## Confocal laser scanning microscopy (CLSM)

To confirm the presence of β-rich fibrillar structures in the ELR-Ca coated prismatic enamel, coated and uncoated sections were stained with ThT dye (100 μM) for 1 h, washed with water to remove excess of dye, and scanned under inverted CLSM (Zeiss LSM 700, Germany) using 488 nm laser.

To demonstrate the formation of mineral layer on top of aprismatic enamel, acid-etched sections were coated partly with ELR-Ca (0.34 mM CaCl$_2$) and partly with acid resistant nail varnish (to prevent mineralisation). Coated sections were subjected to mineralisation followed by degradation of ELR coating by elastase digestion, staining with calcein dye (Thermofisher, UK), and scanning under CLSM.

## Acid attack

Both native and remineralized enamel sections were subjected to 0.1 M acetic acid (pH 4.0) and incubated at 37 °C for different time points[70].

## Nanoindentation

Enamel sections were glued to aluminium stub using Loctite Super glue and air dried for 2 h. Young's modulus I and hardness (H) were measured using a reported method[95]. Briefly, samples were indented using a Berkovich tip at NanoTest P3 instrument (Micro Materials company) at a loading rate of 500 μN/s and with the maximum indentation depth of 300 nm.

## Apparent fracture toughness analysis

Apparent fracture toughness ($K_{c(app)}$) was analysed using the indentation crack length approach[67] on the MMT-7 digital microhardness tester (Buehler) equipped with Vickers diamond indenter. At least 5 indents were made on each sample at 0.1 and 9.8 N load and the diagonal lengths and average crack lengths arising from the indentation corners were measured for each indentation to compute the $K_{c(app)}$ using following expression:

$$K_{c(app)} = 0.0084 \left(\frac{E}{H}\right)^{\frac{2}{5}} \left(\frac{2P}{L}\right) \frac{1}{C^{1/2}} \tag{2}$$

Where, E, H, P, L, and C are Young's modulus (GPa), hardness (GPa), load (N), average diagonal length (m), and crack length (m), respectively.

## Coefficient of friction (CoF) and wear strength (WS)

The wear tests were carried out using a rotary ball-on-disc type tribometer (Ducom Instruments, India) which provided a through-tests measurement of the CoF. Briefly, flat enamel samples were fixed on stainless steel plates using superglue. A 6 mm diameter aluminium oxide counter body ball was slid on enamel samples along a circle of diameter 1.65 mm at 60 RPM for covering 200 cycles at a load of 2 N for a total wear distance of 1.04 m.

Specific wear rate (SWR) was calculated as the total removed volume per unit length and unit load. Optical profilometry was carried out using an Alicona G5 XL optical profilometer (Bruker, Germany) based on the focus variation method, with a vertical resolution of

0.1 μm and a lateral resolution of 1.76 μm. The total removed volume was estimated by measuring four radial depth profiles along the wear track, averaging them, and multiplying by the wear track circumference. The associated error is the standard error.

**Toothbrushing abrasion test**
Native and remineralized enamel sections were fixed to the PDMS substrate with their top surface exposed outside. To imitate toothbrushing, a customised setup was built. Briefly, a force of 200 g or ~ 2 N was applied using a soft toothbrush (Oral-B Vitality Plus Electric Toothbrush, UK) on enamel samples immersed covered in a slurry of toothpaste (Oral-B Pro-Expert Toothpaste) prepared by mixing 16 g of toothpaste in 100 mL of deionized water[66]. Samples were subjected to circular strokes for continuous 15 and 60 min that corresponded to 112.5 and 450 days of toothbrushing abrasion, respectively. Brushed samples were later observed under SEM and tested for mechanical properties using nanoindentation.

**Chewing & grinding abrasion test**
Extracted human molars ($n = 32$) were collected and cleaned with a brush with pumice powder. Half of the molars ($n = 16$) were sectioned 1 mm above the cemento–enamel junction, to separate the crowns from the roots to ensure that the dentine is exposed, any exposed pulp chamber is filled with clearfil Majesty ES flow (Kuraray Noritake Dental inc.). Samples were embedded in dental acrylic resin (PMMA Candulor), exposing the enamel and dentine surface. The surfaces of the dentine samples were ground flat and polished with water-cooled 400-grit paper. Reference planes for profilometry were created in the PMMA on both sides of the embedded specimen with a milling machine. 8 samples each for enamel and dentine were first acid etched, coated with ELR coating, and mineralised for 10 days as reported above. Healthy native samples, 8 for each enamel and dentine were loaded in the apparatus as untreated.

Wear simulation was performed using a reported Rub&Roll apparatus and technique[69]. Briefly, enamel and dentine samples were mounted in a rotating cylinder, where the samples protruding 1 mm above the cylinder surface and 4 rods with 4 PVC tubes are mounted where the loading is about 75 N. The specimens are exposed to cyclic loading with water. After 2 weeks of continuous loading in the Rub&Roll with a rotation speed of 10 RPM, a total of 806400 cycles were performed, mimicking about 3.5 years of clinical loading. Samples were scanned using a non-contact profilometer before and after loading, a special sample holder with metal strips was used and the reference area in the specimens as the reference area for profilometric measurements of substance loss. The mean Volume loss ($V_L$, mm³) and height loss ($H_L$, μm) was calculated[96].

**In vitro cytotoxicity assays**
**Cell viability determination by MTS assay.** Biocompatibility of the ELR coatings was quantitatively assessed on three different cell lines, NIH 3T3 fibroblasts (ATCC, USA), human immortalised mesenchymal stem cells (MSCs, Lonza, Switzerland) and human umbilical vein endothelial cells (HUVECs, Lonza, Switzerland), using an MTS assay kit (Abcam, UK). Glass coverslips (15 mm diameter) were coated with the ELR matrix by drop casting a 5% ELR solution prepared in ethanol/water (9/1 ratio) and containing 1.5 mM $Ca^{2+}$ ions and 1.5% glutaraldehyde. Samples were then sterilised by immersing them in 70% ethanol for 30 min. After washing them with PBS to remove the excess of ethanol, they were transferred to 24-well plates (Corning®, USA), UV-irradiated for 1 h and incubated in cell culture media for another hour, to promote protein adsorption and cell attachment[97].

NIH 3T3 fibroblasts and MSCs were cultured in high- and low-glucose Dulbecco´s Modified Eagle´s Medium (DMEM, Fisher Scientific, UK), respectively, containing 10% FBS and 1% antibiotic/antimycotic, while HUVECs were cultured in Endothelial cell growth medium (Lonza, Switzerland) containing 1% antibiotic/antimycotic. All cells were kept in standard cell culture conditions (37ºC and 5% $CO_2$), and when 80% confluency was reached, they were trypsinised and seeded in the sample-containing 24-well plates at a cell density of $5 \times 10^3$ cells/cm² in the case of fibroblasts and MSCs, and $25 \times 10^3$ cells/cm² for HUVECs.

The different cells lines were incubated with the samples ($n = 6$) for 4 and 8 days, respectively. Then, the cell viability was assessed using an MTS assay (MTS Assay Kit, Abcam, UK) following the manufacturer's protocol. After 3 h of incubation, the absorbance at 490 nm was measured using a microplate reader (Infinite M Plex, Tecan, Switzerland), and cell viability relative to positive control (cells cultured in tissue culture plastic) was calculated according to the following equation:

$$\text{Cell viability (\%)} = (\text{Sample Absorbance} / \text{Positive control Absorbance}) \times 100 \tag{3}$$

**Cell viability determination by Live-dead staining.** To qualitatively confirm the biocompatibility of the ELR coatings, a Live/Dead staining (Thermo Fisher Scientific, UK) was carried out following the manufacturer´s instructions. In a similar way as described in the previous section, NIH 3T3 fibroblasts, MSCs and hUVECs were seeded on the top of sterile ELR-coated coverslips, previously placed in 24-well plates, at a cell density of $5 \times 10^4$ cells/cm². After 2 and 4 days, respectively, cells seeded in the samples, along with positive (cells seeded in tissue culture plastic), and negative (cells killed with DMSO) controls, were stained with Calcein AM and ethidium homodimer-1. Calcein AM is hydrolysed into calcein in living cells, leading to a fluorescent green colour, while ethidium homodimer−1 penetrates in cells with compromised membranes staining nucleic acids in red)[98]. In addition, Hoechst 33342 (Thermo Fisher Scientific, UK) was employed to fluorescently stain the cells nuclei in blue. After 1 h of incubation with the staining solution in standard cell culture conditions (37 °C and 5% $CO_2$), cells were washed twice with PBS and imaged under a confocal microscope (Leica TCS SPE, Leica Microsystems, Germany).

**Statistical analysis.** All the data are reported as mean ± SD. GraphPad Prism ver. 10 software was used to perform statistical analysis between the means of different test groups using two-sided one-way analysis of variance (ANOVA) with the Tukey test for comparing 3 or more groups. For comparing 2 groups, a two-tailed Student's $t$ test was performed. $p$-values < 0.05 were considered significant.

### Reporting summary
Further information on research design is available in the Nature Portfolio Reporting Summary linked to this article.

## Data availability
The data that support the findings of this study are provided in the Source Data file, Supplementary Information file, and are also available from the corresponding author upon request. Source data for all main (*i.e.*, Figs. 1–6) and Supplementary Figs. (*i.e.*, Supplementary Fig. 1–52) is available open access and can be found at https://doi.org/10.17639/nott.7600. Source data are provided in this paper.

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

## Acknowledgements

The work was financially supported by the ERC Proof-of-Concept grants ENAMULATE and MINGRAFT, the Engineering and Physical Sciences Research Council [EP/N006615/1], and the Medical Research Council [United Kingdom Regenerative Medicine Platform Hub Acellular Smart Materials 3D Architecture, MR/R015651/1], the NIHR Nottingham Biomedical Research Centre at University of Nottingham, Nottingham, UK, and the project AOCMF-21-04S from the AO Foundation, AO CMF. The AO CMF is a clinical division of the AO Foundation — an independent medically guided not-for-profit organisation. The work was also supported by INNOVATE UK the Smart Grant ENAMEXCEL [10094179]. H.R. acknowledges support from the action "Recualificación de personal universitario: Margarita Salas para la formación de jóvenes doctores", financed by "NextGenerationEU" funds of the European Union. M.F. and C.P. highly acknowledge the support by the Engineering and Physical Sciences Research Council (EPSRC) grant [EP/W006413/1] and [EP/S021434/1] and the University of Nottingham for work done on the K3 and Zeiss CrossBeam 550 FIB-SEM. Authors thank Diamond Light Source Ltd. for the awards of beamtimes sm32387-1,2, and for funding for P38 offline equipment: EPSRC, grant no. [EP/R042683/1]. We thank Dr Sam Burholt for their support with data collection at P38. A.C. acknowledges Spanish MICINN under the Maria de Maeztu Units of Excellence Programme (MDM-2016-0618 and CEX2020-001038-M). J.C.R.-C. is grateful for funding from the Spanish Government (PID2019–110709RB-I00), Junta de Castilla y León (VA317P18), and Centro en Red de Medicina Regenerativa y Terapia Celular de Castilla y León. The authors thank Leonora Martínez Nuñez, a professional scientific illustrator, for preparing the commissioned illustrations used in this work (website: https://www. leonoramartinez.com/).

## Author contributions

A.H. and A.M. conceptualised the work and designed the experiments. A.H., A.C., C.P., and M.F. performed FIB and TEM analysis. A.V.T. carried out simulation experiments, and T.T. performed the analysis. A.H. and H.R. performed in vitro cytotoxicity experiments. A.H., G.G., and N.M.P. analysed the mechanical properties. N.M.P. developed the theoretical mechanical model, and G.G. applied it. A.H., F.V., and T.H. analysed tribological properties. C.J.C.E.-G. and N.C. performed SAXS and WAXS analysis. J.R. performed 'rub-n-roll' experiments and analysed by A.H. and X.F.W. A.H. and B.M. performed AFM experiments and analysed by F.R. A.H. and I.D. performed isothermal calorimetry experiments. A.W. provided tooth and sectioned by S.G. A.H., A.M., S.E., J.C.R.-C., A.B., and S.H. analysed the results. A.H. and A.M. wrote the manuscript. A.M. supervised A.H.

## Competing interests

A.H. and A.M. are two of the co-founders and hold equity in Mintech-Bio Ltd, a company that has been established to translate this technology. The other authors declare no competing interests.

## Additional information

¹School of Pharmacy, University of Nottingham, Nottingham, UK. ²Biodiscovery Institute, University of Nottingham, Nottingham, UK. ³Department of Chemical & Environmental Engineering, University of Nottingham, Nottingham, UK. ⁴Mintech-Bio Ltd., Biodiscovery Institute, University of Nottingham, Nottingham, UK. ⁵CIC nanoGUNE BRTA, Av. de Tolosa 76, San Sebastian, Spain. ⁶IKERBASQUE, Basque Foundation for Science, Plaza Euskadi 5, Bilbao, Spain. ⁷Department of Chemistry, University of Strathclyde, Glasgow, UK. ⁸Nanoscale and Microscale Research Centre, University of Nottingham, Nottingham, UK. ⁹Faculty of Engineering, University of Nottingham, Nottingham, UK. ¹⁰Department of Animal Biosciences, Swedish University of Agricultural Sciences, Uppsala, Sweden. ¹¹Laboratory of Bio-Inspired, Bionic, Nano, Meta, Materials & Mechanics, Department of Civil, Environmental and Mechanical Engineering, University of Trento, Trento, Italy. ¹²School of Engineering and Materials Science, Queen Mary University of London, London, UK. ¹³Department of Dentistry, Regenerative Biomaterials, Radboud University Medical Centre, Nijmegen, The Netherlands. ¹⁴Diamond Light Source Ltd, Harwell Science and Innovation Campus, Didcot, UK. ¹⁵Bioelectronics Laboratory, School of Pharmacy, University of Nottingham, Nottingham, UK. ¹⁶Smilestyle Signature Dental Care, 359 Nuthall Road, Nottingham, UK. ¹⁷Faculty of Dentistry, Oral & Craniofacial Sciences, King's College London, London, UK. ¹⁸Restorative Dentistry Department, Faculty of Dentistry, Tanta University, Tanta, Egypt. ¹⁹BIOFORGE Group, University of Valladolid, CIBER-BBN, Valladolid, Spain. ²⁰Department of Preventative and Restorative Dental Sciences, School of Dentistry, University of California, San Francisco, CA, US. ✉e-mail: a.mata@nottingham.ac.uk

