## [Transparent Peer Review file · Nature Communications]

Biomimetic supramolecular protein matrix restores structure and properties of human dental enamel

Corresponding Author: Professor Alvaro Mata

Version 0:

Reviewer comments:

Reviewer #1

(Remarks to the Author)

I appreciate the authors' comprehensive and detailed response. While they have provided a rationale for not conducting in vivo experiments to validate the clinical potential of their ELR coating, the in vitro experiments using human saliva, though valuable, do not fully capture the complexity of the dynamic oral environment. Specifically, factors such as mechanical stresses (including chewing, friction, and fluid flow) play a crucial role in the real clinical performance of coatings and remineralization processes.

Since the authors emphasize the clinical applicability of their method as a key advantage, I recommend explicitly acknowledging the limitations of the current experimental design within the manuscript. Doing so would enhance the transparency of the study and allow readers to better evaluate the translational potential of the findings.

Reviewer #3

(Remarks to the Author)

The authors have provided a comprehensive and well-structured rebuttal, addressing all concerns raised in the previous review rounds. The additional experiments, particularly those related to the ELR matrix degradation, mineralization in natural saliva, and structural validation, have significantly strengthened the scientific rigor of the study. The new data effectively demonstrate the stability, functionality, and translational potential of the ELR coating.

Based on the substantial revisions and the robust new experimental data, I am satisfied that the authors have adequately addressed all concerns.

Reviewer #4

(Remarks to the Author)

After carefully reading the authors' response letter and the revised manuscript, I believe that, compared with the previous version, this work does not provide substantial evidence to support the authors' claim that ELR can effectively regulate the epitaxial growth of enamel. Moreover, in terms of the repair effect, compared with other enamel repair studies, this work does not offer a strong improvement to justify its publication in a high-impact journal such as Nature Communications. Therefore, I suggest that this work is more suitable for publication in a more specialized journal. The comments are as follows:

1. In introduction part, the motivation behind work to develop an ELR-based strategy to regulate the epitaxial growth of enamel, is still unclear. In most cases, the epitaxial growth of crystals does not require regulation by an organic matrix template; its driving force is the intrinsic dipole field within the crystal. The key challenge is in controlling the kinetics of the process. Most studies focus on stabilizing or destabilizing small-sized precursors (such as ACP) to regulate epitaxial growth. However, this work does not clearly elucidate the role of ELR, lacking both logical depth and a comprehensive analysis.
2. The presentation and description of the data are also puzzling. The paper emphasizes the templating function of ELR, but the data presented, including all the TEM and SEM images, are intended solely to demonstrate epitaxial growth. These data do not prove that epitaxial growth is achieved through the templating effect. In other words, based on the current data, the

claim that the ELR template promotes the epitaxial growth of enamel is unconvincing.

3. There is a lack of animal models or preclinical trial data, and the experiments using natural saliva from different human donors as a substitute for animal experiments are questionable, as factors such as saliva washing and microbial interference have not been considered. In particular, the authors stated in their response letter that almost all animal models are unacceptable, yet they did not specify the reasons, which is puzzling. I speculate that the fundamental reason the authors mentioned for not conducting animal experiments is that the preparation of the ELR membrane requires the use of organic reagents that pose health risks; to date, the authors have not overcome this limitation. Therefore, at present, the use of ELR is quite restricted.

4. The repair efficiency is average: approximately 10 microns grown in 10 days, which is not very fast and lacks clinical immediacy, potentially limiting its practical applications (such as rapid outpatient treatment). In contrast, a recent publication in *Nature Communications* (2025, 16:58) showed a growth of 100–300 nm in 30 minutes, indicating a much faster speed, with the structure of the repair layer closely resembling natural enamel.

5. Although the paper conducted a relatively systematic study on the mechanical properties of the repaired enamel layer, the mineralization period in the study is uniformly 10 days, which is quite long. Moreover, the research on mechanical properties and stability focuses only on the conditions before and after remineralization, without mentioning the stability and mechanical properties during the 10-day mineralization process, when mineralization is not yet complete.

6. There are many technical issues. For example, in Figure 1(b), although the authors indicate that the scattering peaks originate from CaCl₂ in the sample, they still cannot explain these strange signals. In the two-dimensional image, the scattering ring corresponding to 0.47 nm is clearly diffused, yet why is it so sharp in the one-dimensional integrated curve? If the authors claim that the sample is impure, I consider this characterization very poor, as how can the authors be sure that the sharp 0.47 nm peak does not come from another salt? Additionally, in Figure 3(b), why does the ELR membrane stained with ThT not appear as the continuously phased membrane previously displayed, but instead looks like fish-scale enamel after corrosion? Furthermore, the FIB image in Figure 5(e) shows that after mineralization, the structure is not very dense—is it necessary to present a larger-scale cross-sectional FIB image?

Version 1:

Reviewer comments:

Reviewer #3

(Remarks to the Author)

This is a technically robust study that presents a promising biomimetic strategy for enamel regeneration using an engineered ELR matrix. The authors have provided a comprehensive and well-reasoned rebuttal, with new data and clear clarifications that address previous concerns.

The study is particularly strong in its multiscale characterisation and functional assessment. The mechanical testing under simulated physiological conditions (abrasion, mastication, acid exposure) convincingly demonstrates the durability and potential applicability of the regenerated enamel layer.

The authors also thoughtfully distinguish between ELR's templating role and its influence on crystal orientation. Although direct evidence of ELR-regulated epitaxial growth remains limited, the combined microscopy, simulation, and structural data support a facilitative role that aligns with current understanding of biomineralisation mechanisms.

Reviewer #4

(Remarks to the Author)

Upon meticulous scrutiny of the authors' rebuttal, it is abundantly clear that their lengthy explanations amount to little more than empty rhetoric. They have utterly failed to provide a shred of convincing experimental evidence—most glaringly, there is no substantive data whatsoever to support their central claim that the ELR template can regulate HAP epitaxial growth. In light of these fundamental and inexcusable deficiencies, I must, from an objective scholarly standpoint, urge outright and unreserved rejection. The reasons are as follows:

1. The authors have merely cobbled together a rag-tag assortment of barely relevant citations in a feeble bid to shore up their experimental approach—but this circus of references does nothing to disguise the utter lack of rigor and reliability in their data. Crystal growth—whether classical or non-classical—is, by its very nature, a dynamic, time-dependent process. Yet these authors have the audacity to lean exclusively on dry-TEM “snapshots,” a technique that not only leaves rampant sample-prep artifacts unchecked but also offers zero physico-statistical validation. The result is a one-sided, unsubstantiated narrative that borders on scientific malpractice. Let us be unequivocal: any credible study of such dynamic phenomena must employ true in situ or native-state methods—cryo-TEM, solution-SAXS, in situ AFM, etc.—not this patent misrepresentation that dry-TEM alone can prove ELR-mediated control of HAP epitaxial growth. Such a flawed claim plainly has no place in a widely read, highly authoritative journal like *Nature Communications*.

2. The authors ludicrously claim, “To our knowledge, no one has yet designed a protein matrix to replicate the supramolecular organization of the natural enamel protein matrix.” This statement is not just arrogantly uninformed—it's a brazen misrepresentation of decades' worth of research. The elastin-like recombinamer (ELR) they flaunt bears no true resemblance to native enamel proteins; their primary and secondary structures are fundamentally incompatible. An overwhelming body of literature has already proven that enamel proteins drive mineralization through exquisitely tuned, dynamic assembly/disassembly behavior—yet the ELR system in this study is utterly bereft of such behavior, making any

asserted functional relevance downright nonsensical. To compound the insult, countless biomimetic enamel-protein mineralization platforms have long been published; the authors' insistence that "no one has designed a protein matrix" is either willfully oblivious or borderline disingenuous. Finally, the manuscript offers no coherent mechanistic insight into the mineralization process, and its touted enamel-regeneration results are embarrassingly inferior to established benchmarks.

3. Though the authors go through the motions of comparing their work to previous studies, they pathetically fail to surpass any established benchmarks on the metrics that truly matter—an especially glaring shortcoming given that those earlier reports buttress their claims with rigorous animal experiments as indispensable proof for clinical translation. By contrast, this manuscript offers zero *in vivo* data; the authors' incessant appeals to "legal and ethical constraints" ring hollow and serve only as a feeble smokescreen for the most glaring gap in their study. This omission isn't a minor oversight—it is a crippling flaw that utterly undermines the paper's relevance for follow-up research or real-world clinical application. In light of such meager evidence, it borders on the absurd to assert that this approach holds any genuine potential for translation—thus, we categorically reject their overblown claim.

4. The authors stubbornly cling to their technical blunders, leaning on a rag-tag assortment of flimsy excuses instead of providing solid, valid data to convince reviewers or readers. This results in a manuscript so loosely assembled that every section collapses under minimal scrutiny—and many of the reported data even contradict one another. The paper is riddled with errors; here are just a few of the most glaring:

(1) Small-angle scattering data (Figure S8) report a dry-state initial slope of 4.1, yet the authors offer no meaningful explanation. They omit any solution-state SAXS to show whether ELR actually forms fibers in dispersion. If it does, how does it transform into a film whose initial slope is 4.1—clearly implying internal crosslinking or branching? Such dramatic structural changes would undoubtedly affect subsequent mineralization—so why no discussion?

(2) The ELR–Ca²⁺ interaction characterization is full of yawning gaps. Their simulation (Figure 2d) suggests ELR–Ca²⁺ complexes are more stable than Ca²⁺–H₂O coordination—so why does drying produce massive CaCl₂ precipitation (Figure 2b)? And if Fig S6 shows Ca²⁺ modulates ELR assembly, why arbitrarily pick 1.5 mM Ca²⁺? These omissions border on the absurd.

(3) The WAXS data (Figure 2b) are outright invalid: the samples are impure, producing hopelessly confounded patterns. To characterize ELR chain stacking, one must purify the sample before measurement; to study Ca²⁺-induced nanostructural changes, one conducts Ca²⁺ titration solution SAXS—not analyze a heterogeneous slurry. The authors' cavalier approach is shockingly unprofessional.

(4) The TEM results are nothing short of a disaster. In Figure 1f, without elemental mapping there is zero evidence that the observed HAP fibers are templated by ELR—fluoride alone can yield fibrous HAP. Figure 2a offers no proof linking those fibers to ELR, as if the authors make claims at random. Figure S13 repeats this flaw: even granting their premise, only two crystals are shown—hardly convincing.

(5) Other telling lapses: Figure S15 presents only post-mineralization XRD—where is the acid-etched enamel before treatment? In Figure S17, the authors claim final film thickness depends solely on the initial ELR layer, yet their SEM shows only morphology, not composition. Moreover, SEM images reveal negligible surface change after mineralization—seriously calling into question whether any mineralization occurred at all.

In sum, this manuscript is fatally undermined by technical deficiencies and utterly fails to meet the rigor, coherence, and impact expected of a Nature Communications publication.

Reviewer #1 (Remarks to the Author):

I appreciate the authors' efforts in providing a comprehensive and detailed response, addressing concerns raised in the previous review. However, upon reviewing the latest version of the manuscript, I have identified some additional comments that merit consideration and clarification as follows.

Response: We thank the reviewer for appreciating our responses in the 1st round of revision and are grateful for bringing additional important points to our attention, which have further improved our study.

1. In the context of utilizing animal models for enamel regeneration, relevant investigations into enamel remineralization have been conducted (Wang, Dong, et al. "Controlling enamel remineralization by amyloid-like amelogenin mimics." *Advanced Materials* 32.31 (2020): 2002080; Chen, Mei, et al. "Modulated regeneration of acid-etched human tooth enamel by a functionalized dendrimer that is an analog of amelogenin." *Acta Biomaterialia* 10.10 (2014): 4437-4446.). Is it possible to extrapolate similar models to investigate potential in vivo outcomes?

Response: We thank the Reviewer for this suggestion.

We first would like to elaborate on the limitations that we have regarding the animal testing. We are aware of the studies mentioned by the Reviewer. However, these are not standard nor well-accepted models within our location and have therefore been advised against them given ethical and functional considerations. We believe that this is a major reason why there has never been, to the best of our knowledge, an enamel regeneration study in an animal model within the UK or other regions such as Europe and the US. It is important to mention that these models require hanging of a small section of human enamel inside the oral cavity of rats which can (i) irritate the animal, (ii) make feeding difficult, and (iii) lead to sample ingestion. We respectfully request the Reviewer to consider these important limitations.

However, we completely agree with the Reviewer on the importance to further validate the potential translation of our technology. This is a major focus of our study and have therefore conducted additional experiments to assess the mineralization capacity of our platform using natural saliva from different human donors. The results confirmed the restoration of prismatic structures (**Fig. 6j** and **Supplementary Fig. 45a**), mechanical (stiffness and hardness) (**Fig. 6k** and **Supplementary Fig. 45c**), and chemical properties (**Supplementary Fig. 45d, e**) of enamel. We hope that the Reviewer agrees that this experiment provides strong evidence for translation given the rich composition and variable nature of human saliva and that it serves as an alternative to the animal experiment. We have reported these results on **Page 16** of the revised manuscript as follows:

Furthermore, to test the capacity to mineralize in natural human saliva, ELR coated enamel samples were exposed to saliva from three different donors for 2 weeks. Again, the results revealed the capacity of the ELR matrix to recreate prismatic enamel over large surface areas (**Fig. 6j, Supplementary Fig. 45a**) and restore both the mechanical ($E = 74.6 \pm 15.3$ GPa, $H = 2.7 \pm 1.1$ GPa) (**Fig. 6k, Supplementary Fig. 45c**) and chemical (**Supplementary Fig. 45d, e**) properties of natural enamel. These results confirm the robust mineralizing capacity of our ELR platform by enabling a similar epitaxial growth despite the inherent compositional complexity of natural saliva. Furthermore, the structure and properties of enamel were recovered on all tested samples, independently of variations in free Ca ion concentrations in the saliva from each donor (**Supplementary Fig. 45b**).

The methodology used for remineralizing enamel using natural saliva is described in the methods section of the revised manuscript on **Page 25**:

Mineralization using natural saliva

Human saliva was collected from 3 different healthy donors after approval by the University of Nottingham Ethics Committee, reference number FMHS 313-0721. The donors were between 20 and 40 years old and had healthy periodontal conditions with no signs of gingivitis, periodontitis, gingival recession, carious decay, or dental erosion. Donors were instructed to refrain from eating or drinking (except water) for two hours before saliva collection. Unstimulated saliva samples were collected directly into polypropylene tubes. For enamel remineralization, ELR coated enamel sections were separately exposed to 1 mL of whole human saliva collected individually from the different donors for 2 weeks. Enamel sections were incubated at 37 °C and the saliva was replenished every 3 hours during a 12-hour daytime period and then left in the same spent saliva for the subsequent 12 hours overnight. Post mineralization, ELR coating was enzymatically degraded using elastase as described in the section 'Mineralization experiment'. Calcium ion concentrations in the saliva samples were measured using a colorimetric assay kit (Abcam, ab102505), according to the manufacturer's instructions.

Beyond the use of natural saliva, it is important to mention that we have conducted a large number of functional tests designed to validate the potential translation of our platform. These include:

- Testing the stability of the ELR coating against realistic insults including exposure to extreme pH, salt solutions, extensive tooth brushing, sonication, high temperature, and detachment using a standard tape peeling test (**Supplementary Fig. 46, 47, 48**).
- Testing the ability of the ELR coating to remineralize enamel using modified-artificial saliva to imitate real oral environments such as the presence of salivary enzymes and dynamic abrasive conditions (**Supplementary Fig. 44**).
- Testing the capacity of the ELR coating to remineralize in artificial saliva after exposure to extensive toothbrushing (**Fig. 6l, m, n**), sonication (**Supplementary Fig. 47a**) extreme pH (**Supplementary Fig. 46a, b**), salt solutions (**Supplementary Fig. 46c, d**), high temperature (**Supplementary Fig. 47b**), and a peeling treatment (**Supplementary Fig. 47c**).
- Testing the stability of the remineralized enamel under extensive toothbrushing (**Fig. 6a, b**) and exposure to fracture (**Fig. 6c, d**), attrition (**Fig. 6e, f, g**), and acid attack (**Fig. 6h, i, j**).

2. What is the intricate interplay between the degradation process of the ELR coating and the ELR coating-induced remineralization process? Do these processes occur synchronously, or does the ELR coating degrade after the completion of remineralization? The concern arises that if the coating degrades too rapidly, there is a risk of losing the template for ordered crystal growth. Conversely, if the degradation is overly gradual, questions arise regarding the mineralization mechanism within such a densely packed protein coating. Beyond the examinations of SEM images post-ELR coating degradation, a more comprehensive analysis of the degradation behaviour of the ELR coating is imperative for elucidating its remineralization mechanism and facilitating clinical application.

Response: We thank the Reviewer for highlighting this important point. We have conducted a number of new experiments to investigate the degradation process and, together with our previous data, provide compelling evidence of the stability of the ELR matrix to be both functional and biodegradable. We have presented these new results in **Supplementary Fig. 24** and elaborated them in **Supplementary Discussion 5** as shown here:

We have previously described how the ELR molecules assemble into a dense matrix with an ELR conformation that enables the growth of highly organized mineralized structures [1]. This study also demonstrated that this densely packed and crosslinked ELR matrix remains present during the mineralization process and can be completely degraded using elastase enzymes, demonstrating that the ELR matrix is both necessary to trigger organized mineralization and biodegradable.

To further investigate this mineralization/degradation process, we first conducted swelling experiments by exposing the ELR matrix to a mineralizing pH = 6 condition, which revealed a 22% swelling in this condition (**Supplementary Fig. 24a**). We hypothesized that this swelling would enhance ELR matrix degradation and thus we performed degradation tests by exposing the ELR matrix to a mineralization solution in the absence or presence of salivary enzymes for 3 weeks. The results confirmed that the ELR matrix degraded by 14.8 ± 0.4 wt% and 18.5 ± 0.2 wt% in the absence and presence of salivary enzymes, respectively (**Supplementary Fig. 24b**).

We then conducted a thermogravimetric analysis (TGA) to assess differences in the degradation of the organic ELR matrix before and after mineralization. The results further confirmed a loss of ~15 wt% of the ELR matrix during the period of mineralization (**Supplementary Fig. 24c**), which corresponds to the results from the degradation experiment (**Supplementary Fig. 24b**). In addition, given this swelling and degradation of the matrix as well as the critical role of the secondary structure of the ELR for mineralization, we then conducted FTIR experiments to assess changes during the mineralization process. The results revealed that the secondary structure conformation of the ELR did not change significantly during the mineralization process (**Supplementary Fig. 24d**).

Overall, these results demonstrate that the ELR matrix undergoes slow degradation but remains both stable and functional (i.e., maintaining its secondary structure conformation) during the distinctive epitaxial and organized mineralization in both artificial saliva (**Supplementary Fig. 15, 44**) and natural saliva (**Fig. 6j, k, and Supplementary Fig. 45**) conditions.

Also, as reported in the previous version of the manuscript, we have already performed several studies to demonstrate functionality of our ELR material while taking into account its degradation. These include stability of the ELR coating against diverse physical and chemical treatments that imitate oral environments such as extreme pH, salt solutions, extensive tooth brushing, sonication, high temperature, and a standard tape peeling test (**Supplementary Fig. 46, 47, 48**). The results confirmed no observable or measurable effects on surface morphology, mechanical properties (E and H), and functionality (i.e., ability to remineralize enamel) of the ELR coating after these treatments (**Supplementary Fig. 46, 47, 48**).

The methodologies used for carrying out swelling, ELR degradation, and TGA analysis are described in the methods section of the revised manuscript on **Page 25** as shown below:

Swelling analysis

The swelling behaviour of the ELR membranes fabricated using ethanol/water (85/15) was investigated using a previously reported methodology [22]. Briefly, dried ELR membranes of known initial weights (W_i) were soaked in PBS (pH 6.0) overnight at 37 °C, then carefully removed and immediately weighed (W_f) after removing excess PBS using filter paper. The experiment consisted of three samples ($n = 3$). The swelling percentage was calculated using the expression below:

$$\text{Swelling (\%)} = \frac{W_f - W_i}{W_i} \times 100$$

Degradation analysis

Degradation studies were performed by immersing the ELR membranes in 1 mL of artificial saliva without and with proteinase K (15 ng/mL) at 37 °C for different times. At each time point, 50 μ L of the artificial saliva was removed and analysed for ELR release resulting from the membrane degradation due to mineralization and enzymatic degradation. The amount of ELR degradation was analysed using the Micro BCA Protein Assay Kit (Thermo Scientific Pierce) by comparing the results to a standard curve generated with varying concentrations of ELR molecules.

Thermogravimetric analysis (TGA)

TGA analysis was carried out using TGA Q500 V20.13 (TA instruments) using a heating rate of 10 °C/min to 600 °C.

3. In Fig. 3C, an intriguing question arises concerning why the remineralized enamel (marked in blue) is exposed without the cover of the ELR coating. Is enamel growth initiated from the exterior of the ELR coating or from within?

Response: We apologize for not making this clearer. The reason that in this image the remineralized enamel (in blue) appears outside of the ELR coating is due to partial degradation of the ELR coating by elastase treatment after the mineralization process. Given the relatively slow degradation of the ELR matrix compared to its mineralization, we used an elastase solution after specific times to remove the ELR matrix and more rapidly observe the resulting mineralized structures. In this particular Figure, we used the elastase solution in a way that allowed us to expose the mineralized layer and simultaneously see the ELR coating and the remineralized enamel in the same sample. Briefly, after remineralization of the enamel for 10 days, we covered half of the ELR coated enamel with nail varnish and exposed the other half to the elastase solution, degrading only the exposed ELR coating. This has now been more clearly stated in the caption of **Fig. 3c** as shown below:

After the mineralization process, a part of the ELR coating was protected with nail varnish, while the exposed area was degraded using elastase to partially reveal the remineralized layer on the enamel surface.

To briefly elaborate on the mechanism of mineralization within the ELR coating, the ELR facilitates epitaxial mineralization from an inorganic template. When the ELR is coated on top of enamel, this inorganic template is immediately provided by the surface of the native enamel crystals, triggering an epitaxial remineralization process from the surface of the native enamel crystals and towards and within the ELR coating. This epitaxial growth maintains the

same crystallographic orientation as the native crystals. This mechanism is described in detail in **Supplementary Discussion 10** and specific evidence is provided in the manuscript in the following sections:

- TEM image showing remineralization of the apatite nanocrystals embedded within the ELR matrix (**Fig. 2a** and **Supplementary Fig. 13**) and demonstrating epitaxial crystal growth preferentially along the c-axis.
- SEM images showing new crystals growing from diazone (**Fig. 3d**) and parazone prisms (**Fig. 3e**) following the native crystallographic orientation and growing through and within the ELR coating.
- TEM images showing new crystals growing preferentially along the c-axis of the native enamel crystals (**Fig. 4a and h**).
- Simulation results confirming the preferential binding of the ELR fragments to the a-axis of the crystals instead of the c-axis of the crystals, thus, energetically favouring crystal growth along the c-axis [PNAS, 2022, 119(19), e2106965119] (**Supplementary Fig. 11**) as evident in **Fig. 2a**.

4. In Fig. 1f, comprehending the schematic illustration of the mineralization process within the fibril-rich ELR matrix from the corresponding TEM images remains challenging. For instance, discerning the HA nanocrystal from the background in Fig. 1f(iii) remains difficult. Therefore, it's suggested that the mineralization of the ELR fibrils described in the schematic illustration of Fig. 1f and Section 2.5 be substantiated with additional evidence. This substantiation would fortify the clarity and credibility of the presented mineralization process.

Response: We thank the Reviewer for pointing this out. The reason behind this difficulty in differentiating the HAp nanocrystals is because the ELR fibrils in the background are beam sensitive, which reduces the contrast and resolution of the image. To improve on this, we have now conducted additional experiments and acquired better quality TEM images, which are incorporated in **Fig. 1f (iii)** of the revised manuscript. We also conducted selected area electron diffraction (SAED) from the same region and confirmed a corresponding crystalline structure. In addition, we also performed energy dispersive X-ray (EDX) analysis that verified the formation of fluorapatite crystals (**Supplementary Fig. 10**).

5. The annotation of the ELR matrix in Fig. 2a is ambiguous. Is it encompassed within the entire background, or is its specific location within the image discernible?

Response: We apologize for not making this clearer. **Fig. 2a** shows the mineralizing nanocrystal completely embedded within the ELR matrix, which surrounds both a- and c-axis of the crystal and is present within the entire background. We have now acquired a better-quality TEM image for **Fig. 2a** based on the comment from Reviewer #4 and explained this better in the caption of **Fig. 2a** of the revised manuscript as follows:

Fig. 2a. TEM image showing the preferential growth of an apatite nanocrystal along the c-axis while embedded within the ELR matrix.

6. Several SEM images present partial blurring, particularly in Figs. 6k, S41, and 42.

Response: We agree with the reviewer and have now explained this better in the caption of Fig. 6k of the revised manuscript, as shown below:

The images presented in **Fig. 6l, S46, and 47** appear blurry due to a visual effect created by the ELR coating uniformly conforming over the enamel surfaces.

We have now conducted additional experiments and acquired better quality SEM images as presented in **Supplementary Fig. 49**, confirming that the images are in focus.

7. Regarding Fig. 2d, the second figure appears incongruous within the delineated region (black box) of the first figure. Likewise, in Fig. 5d and Fig. S14(e-f), the magnified areas seem discordant within their respective delineated regions (dashed box).

Response: We thank the Reviewer for raising these points. For **Fig. 5d** and **Supplementary Fig. 16e, f**), we have now incorporated exact images from the delineated regions in the revised version of the manuscript. However, for **Fig. 2d**, we have now removed the black box pointing to the delineated region to avoid confusion.

Reviewer #3 (Remarks to the Author):

I have carefully reviewed the revised version of the manuscript, and I am pleased to note that the authors have made substantial efforts to address the concerns and incorporate changes based on the comments provided by myself and the other reviewers.

Specifically, the authors have successfully revised sections that were previously unclear or required additional clarification. The modifications made have notably improved the overall coherence and readability of the manuscript, resulting in a more robust scientific contribution.

Response: We thank the reviewer for the constructive feedback, which significantly helped us improve both our study and manuscript.

Reviewer #4 (Remarks to the Author):

The article is organized for the aim of restoring structure and properties of human dental enamel using a tuneable and resilient supramolecular matrix based on elastin-like recombinamers (ELRs). Current strategies for enamel biomimetic remineralization primarily center on biomimetic amelogenin, typically achieved through amelogenin derivatives or its functional or structural analogs. However, it seems like the manuscript on the designed ELR fibrils lacks novelty and innovation in material design, especially considering previous reports on proteins or peptides with amyloid-like structures for enamel biomimetic remineralization (PNAS, 2022, 119, 19, e2106965119; Frontiers in Physiology, 2022, 13, 1063970).

Response: We would like to thank the Reviewer for the insights provided.

However, we respectfully disagree with the Reviewer on the lack of novelty of the study and would like to take the opportunity to elaborate on this point.

The two papers that are mentioned above provide mechanistic insights into the self-assembly of amelogenin to create the fibrillar structures that template apatite mineralization during amelogenesis. In fact, these reports have been a major inspiration for us and have guided the

design of our ELR-based material platform to imitate the supramolecular and structural characteristics of amelogenin in nature, as these papers have reported.

To the best of our knowledge, nobody so far has designed protein matrices that recreate this supramolecular organization of the natural amelogenin matrix (based on current understanding) to the level and in-depth analysis that we have (using both computational and experimental work as well as multiscale design, engineering, and validation). In our study, we not only demonstrate the capacity to recreate structural features of this protein matrix (**Fig. 1a - e**), but also demonstrate its functionality by enabling epitaxial, organized, and protein-templated mineralization (**Fig 1f; Fig. 2d, g, i; Fig. 3d, e**).

The reviewer rightly points to previous studies using amelogenin-derived peptides to remineralize enamel. However, we feel that these studies (ACS Biomater. Sci. Eng., 2018, 4, 1788-1796; ACS Omega, 2018, 3, 2546-2557; Acta Biomaterialia, 2013, 9, 7289-7297; Biomaterials, 2009, 30, 478-483) describe a different kind of material. First, they do not report on the structural supramolecular properties of their matrices nor claim to recreate those of the natural amelogenin matrix. Also, the resulting mineralization achieved by these materials is not epitaxial and fails to recreate the intricate microarchitecture and, consequently, key functional properties of enamel.

Beyond the capacity to recreate the structure/function of the amelogenin matrix and the structure and function of enamel, another key novelty of our study is the translation potential of our technology. For translation, it is critical to conduct studies that test the stability and functionality of both the ELR coating and the remineralized enamel under a range of physiological conditions and experiences of daily life. In our study, we demonstrate that the ELR coating is stable against physical insults including toothbrushing (**Fig. 6l, m, Supplementary Fig. 50**), sonication (**Supplementary Fig. 47a**), and peeling (**Supplementary Fig. 47c**); chemical insults including exposure to extreme pH (**Supplementary Fig. 46a, b**) and salt solutions (**Supplementary Fig. 46c, d**); and thermal insults (**Supplementary Fig. 47b**). In all of these tests, our ELR coating remained stable and functional, assessed by SEM observations of structure and texture (**Supplementary Fig. 46, 47**), measurements of stiffness and hardness (**Supplementary Fig. 48a, b**), and the capacity to remineralize enamel (**Fig. 6l, m, n and Supplementary Fig. 46, 47**). To the best of our knowledge, none of the other studies using amelogenin analogs (mentioned above) or other matrices (Sci. Adv. 2019, 5, eaaw9569, Adv. Sci., 2022, 9, 2103829; J. Mater. Chem. B, 2018, 6, 1984-1994; Adv. Materials, 2020, 32(31) 2002080; Scientific Reports, 2017, 7, 40701) have demonstrated this level of translational potential.

In addition, all these technologies require application procedures that are difficult to translate clinically such as long application times ranging between 15 minutes and 12 hours or the need for multiple applications (e.g., J. Mater. Chem. B, 2018, 6, 1984-1994). In contrast, our technology requires only one application of lasting 3-4 minutes. Also, some of these technologies use toxic components during their application (e.g., triethylamine in Sci. Adv. 2019, 5, eaaw9569), compared to our biocompatible ELR matrix (**Supplementary Fig. 36 - 39**). Furthermore, all these matrices offer limited stability due to poor adhesion to the enamel surface. In contrast, our ELR coating is highly stable against realistic insults including exposure to extreme pH, salt solutions, extensive tooth brushing, sonication, high temperature, and detachment using a standard tape peeling test (**Supplementary Fig. 46, 47, 48**).

Another important novelty of our study is the capacity to remineralize all the different anatomical regions of enamel including prismatic, interprismatic, and aprismatic regions (**Fig. 2d, g and Fig. 3d, e**) while restoring tissue stiffness, hardness, toughness, coefficient of friction, wear strength (**Fig. 3f, g, h**). Furthermore, we demonstrated the stability of the remineralized layer under conditions that emulate daily abrasion from toothbrushing (**Fig. 6a, b and Supplementary Fig. 32**), tooth wear from chewing and grinding (**Fig. 6e, f, g**), and stability against acid exposure from dietary habits (**Fig. 6h, i, j**). In addition, we have also demonstrated the capacity of our ELR coating to mineralize in human saliva from different donors, recreating the microarchitecture and restoring the mechanical and physio-chemical properties of enamel. These new results are presented in **Fig. 6 j, k and Supplementary Fig. 45** of the revised manuscript.

Another novelty is the capacity of our ELR coating to also trigger the epitaxial and organized enamel-like layer from exposed dentine (**Fig. 5c, d, e, Page 12-13**), demonstrating the capacity of our platform to recreate the microarchitecture and restore key functional properties of the different anatomical regions of enamel irrespective of the level of erosion down to bare dentine. To the best of our knowledge, this has not been reported before. Instead, previous studies on the deposition of mineral layers on dentine surfaces mediated by amelogenin-derived peptides do not trigger epitaxial growth (ACS Biomater. Sci. Eng., 2023, 9, 3, 1486-1495).

Altogether, we believe that this data demonstrates the translation potential of our technology and a superior remineralization performance compared to previous seminal studies including Sci. Adv., 2019, 5(8), eaaw9569 and Adv. Mater., 2020, 32(31), 2002080. Due to the underlying bioinspired mechanism, hierarchical mineralization, performance, and translation potential of our approach, we believe that our work represents a step change in functional enamel remineralization compared to previous studies.

Despite two rounds of revisions, many issues can still be found in this manuscript: The data presented in the article do not conclusively prove that enamel regeneration is due to ELR-mediated epitaxial growth. Additionally, TEM and SEM data are inconsistent and contradictory. Based on the reviewers' comments, it seems that the author has not presented very solid data to persuade Reviewers 1 and 2. Thus, it is suggested that the manuscript is not suitable for publication in Nature Materials in its current form.

Response: We respectfully disagree with these comments based on the following reasons.

We have only had one round of revisions and we believe that we addressed most of the comments from all three reviewers. To briefly summarize, Reviewer #1 appreciated our comprehensive and detailed responses in the first round of revision and have now brought some additional comments to our attention which we have addressed. Reviewer #3 confirmed substantial improvement and scientific robustness of our work and seemed to approve of our revised manuscript. While Reviewer #2 did not provide an additional review, we believe that we have addressed the main comments including providing compelling data to support the mechanism of epitaxial mineralization and more accurate and realistic conclusions.

Regarding the mechanism on epitaxial mineralization, we believe that we have presented compelling evidence that confirmed the ELR mediated epitaxial growth. (i) We demonstrated epitaxial mineralization on single apatite crystals (**Fig. 2a**) and summarized the key mechanistic features in the **Supplementary Discussion 11**. (ii) We also demonstrated crystallographic integration of the mineralized layer to the underlying enamel crystals and co-orientation using SEM (**Fig. 3d, e**) and TEM (**Fig. 4a, b, h**). (iii) It is the combination of the epitaxial mineralization and the capacity of our ELR matrix to promote aligned nanocrystal growth (Nat. Commun., 2018, 9:2145) that allows our technology to recreate the different anatomical structures of native enamel (**Fig. 2d and g, Fig. 3d and e**) and, consequently, restore its mechanical (**Fig. 3f**) and tribological properties (**Fig. 3g, h**).

We also carefully revised the conclusions in the previous rebuttal and clarified that our technology offers a unique opportunity to prevent and treat enamel erosion and to treat non-cavitated lesions by regenerating thin (up to $\sim 10 \mu\text{m}$) layers or sections of enamel. Since all cavities begin as non-cavitated lesions, stopping their progression would prevent their exacerbation to cavitated lesions.

In the previous rebuttal, we stated that our technology can realistically offer an alternative with the capacity to grow thin, crystallographically integrated, and functional mineralized layers on enamel or dentine. These layers could (i) serve as protective layers against tooth attrition or erosion on outer aprismatic enamel, (ii) repair micron-scale surface lesions on aprismatic or prismatic enamel, and (iii) serve as protective and functional layers on dentine surfaces for example to prevent or treat hypersensitivity. This capability far surpasses current commercial dental remineralization alternatives and overcomes key translational obstacles that have up to date prevented the capacity to regrow enamel tissue in patients, even as thin functional layers.

Furthermore, we have now conducted additional experiments to test the remineralization capacity of our ELR matrix under natural saliva taken from different donors. The results further demonstrate the capacity to remineralize enamel, while maintaining its intricate structure despite the rich and variable composition of natural saliva. We have included these new results in **Fig. 6j, k**, and **Supplementary Fig. 45** and elaborated on these points on **Page 16** as follows:

Furthermore, to test the capacity to mineralize in natural human saliva, ELR coated enamel samples were exposed to saliva from three different donors for 2 weeks. Again, the results revealed the capacity of the ELR matrix to recreate prismatic enamel over large surface areas (**Fig. 6j, Supplementary Fig. 45a**) and restore both the mechanical ($E = 74.6 \pm 15.3 \text{ GPa}$, $H = 2.7 \pm 1.1 \text{ GPa}$) (**Fig. 6k, Supplementary Fig. 45c**) and chemical (**Supplementary Fig. 45d, e**) properties of natural enamel. These results confirm the robust mineralizing capacity of our ELR platform by enabling a similar epitaxial growth despite the inherent compositional complexity of natural saliva. Furthermore, the structure and properties of enamel were recovered on all tested samples, independently of variations in free Ca ion concentrations in the saliva from each donor (**Supplementary Fig. 45b**).

Also, we have revised thoroughly our TEM and SEM data and respectfully disagree that it is inconsistent and contradictory. Please let us elaborate. First, it is important to mention that TEM and SEM data is normally used to confirm epitaxial mineralization, as has been reported in two recent seminal studies in Sci. Adv., 2019, 5(8), eaaw9569 and Adv. Mater., 2020, 32(31),

2002080. Also, **Supplementary Fig. 7a** shows oriented nanocrystals (as we previously reported in Nat. Commun., 2018, 9:2145) but organised as spheres because there is no inorganic template. In contrast, when an inorganic template is incorporated within the ELR matrix as in **Fig. 2a** or when enamel nanocrystals are presented, then epitaxial growth takes place as demonstrated experimentally (**Fig. 4a, b, e, h**) and with simulations (**Supplementary Discussion 8, Supplementary Fig. 9**). In addition, **Fig. 1f** shows a TEM image of oriented nanocrystals growing within the ELR matrix and we now have TEM images of the nanocrystals in **Supplementary Fig. 7b**, showing a similar morphology and growth.

Overall comments:

1. (a) The mechanism in ELR film-induced enamel regeneration is not clear. Although the author proposes a strategy for enamel regeneration through epitaxial growth of HAp nanocrystals mediated by ELR in figures 1 and 2, the presented data are indeed very contradictory and unconvincing. It seems that the author has not recognized the knowledge gap between template-mediated mineralization, crystal epitaxial growth, and enamel regeneration. The data presented do not bridge these gaps. Firstly, in the Supplementary Information, SEM in Figure S6 shows that the HAp grown within the ELR membrane is mostly non-oriented, spherical crystals. Generally, such spherical nucleation and growth are characteristic of self-nucleation and growth without using any template. However, the schematic in figure 1f presents a typical mineralization process mediated by an ELR fibre template, where amorphous precursors adsorb and wet at the protein template interface then further mineralizes. Clearly, this is contradictory.

Response: We appreciate the various points raised by the Reviewer and have attempted to address them point-by-point in the following text. We have repeated and also performed a number of new experiments to support our responses.

We respectfully disagree with the Reviewer's comment that the data presented in **Fig. 1** and **2** are contradictory and unconvincing. Please let us elaborate. We first demonstrated the capacity of the ELR fibrils/matrix to act as a protein template that adsorbs amorphous precursors and enables their transformation into apatite crystals (**Fig. 1f**). This is a typical mineralization process (Nat. commun., 2018, 9(1), 4170) and has been reported for amelogenin peptide-templated (PNAS, 2011, 108(34), 14097-14102) and protein-templated (Nat. Commun., 2018, 9:2145) mineralization.

However, when an inorganic template is incorporated, epitaxial growth takes place. We have shown this in two different ways. First, by embedding HAP nanocrystals within the ELR matrix as in **Fig. 2a** and then by presenting enamel nanocrystals to the ELR matrix. In these cases, the nanocrystals serve as an inorganic template where new nanocrystals template from and grow epitaxially. We have demonstrated this experimentally (**Fig. 2a, c, d, Fig. 3d, e, and Fig. 4a, b, e, h**) and supported the data with simulations (**Supplementary Fig. 11, Supplementary Discussion 11**). In contrast, when no inorganic template is presented, the ELR matrix promotes the growth of oriented nanocrystals (**Supplementary Fig. 7a**) but organised as spheres. We agree with the Reviewer that this needs to be made clearer and have therefore made this point clearer in the description of **Supplementary Fig. 7** in the revised supplementary information document. We have also conducted new TEM experiments (**Supplementary Fig. 7b**) that now better demonstrate the similarity of these aligned, yet spherically organized crystals, to those in **Fig. 1f**.

(b) Secondly, the evidence for crystal epitaxial growth is solely based on two dry TEM images (figure 2a and figure S10) without in-situ or statistically meaningful data. Clearly, this is not convincing for a journal of the caliber of Nature Materials and is far from sufficient. In solution, the nucleation and growth of crystals, as well as the adsorption of amorphous particles, occur as random events. Without using in-situ TEM or XRD, these two TEM images might simply capture different random events, which do not explain anything definitive.

Response: We thank the reviewer for this very valuable suggestion. We have repeated the experiment to capture more TEM images and performed FFT analysis (**Fig. 2a**) to further confirm epitaxial growth along c-axis and provide statistical evidence. However, we would like to mention that dry TEM imaging is the standard method reported for investigating and confirming epitaxial mineralization. This can be seen for example in the two seminal papers *Sci. Adv.*, 2019, 5(8) eaaw9569 and *Adv. Mater.*, 2020, 32(31), 2002080.

(c) It is important to note that the TEM images shown in Figure 2a are of poor quality, with obvious astigmatism, which would not be acceptable even for publication in general journals.

Response: We understand the Reviewer's comment that the image appears to be of low quality and apologize for not explaining this better. The image appears to be of poor quality because the crystals are embedded within the ELR matrix and hence interfere with the imaging. Also, the ELR matrix is beam-sensitive so it can experience minor damages due to ionization and electrostatic charging which is a common phenomenon reported widely for poorly conducting organic samples (*Micron*, 2019, 119, 72-87). We have now included this explanation in the caption of **Fig. 2a** in the revised manuscript, as shown below:

The lattice lines in the apatite nanocrystal grown within the ELR coating appear blurry due to imaging interference and potential beam-induced damages caused by ionization and electrostatic charging of the beam-sensitive ELR matrix, a common issue for poorly conducting organic samples

Furthermore, to explain this better and provide the highest quality data possible, we have included a large TEM image in **Supplementary Fig. 13** and repeated the experiment to acquire TEM images with better quality, which are now presented in **Fig. 2a** of the revised manuscript.

2. (a) The assembly of ELR fibers is not well explained in the article. It is not clear whether ELR initially aggregates into proto-fibers or remains in monomeric form in the initial DMF/DMSO mixed solution, and then forms a film through solvent evaporation-induced assembly. Additionally, the subsequent SEM and TEM images of the film formation do not show any obvious fiber subunit.

Response: We thank the Reviewer for pointing this important point out. We have now performed new experiments that, together with previous data, we believe substantiate the critical role of Ca ions and crosslinking in the formation of the supramolecular ELR structures in the solvent mixture. We have now included these results in **Supplementary Fig. 6** and elaborated in the **Supplementary Discussion 3** as shown here:

The results demonstrate that ELR molecules in the presence of Ca ions in a DMF/DMSO solvent mixture exhibit an increase in β -conformations (**Supplementary Fig. 3a** and **Supplementary discussion 2**), suggesting assembly into fibre-like supramolecular structures

as previously reported [2]. To confirm this, we performed dynamic light scattering (DLS) and FTIR analyses in a biocompatible ethanol/water (9/1) mixture, chosen over DMF/DMSO to align with the translational focus of our study. The ELR solution (1 mg/mL) without Ca ions exhibited structures with hydrodynamic radius (R_h) of 2.3 nm (**Supplementary Fig. 6b**), which is similar to the size of monomers reported for similar sized proteins (~33 kDa) [3]. However, addition of 1.5 mM Ca ions to the above ELR solution resulted in the formation of ELR structures with increased size ($R_h = 8.7$ nm) and amounts of β -sheet conformation (36.7% compared to 28.9% without Ca) (**Supplementary Fig. 6d, e**). Size distribution analysis of these structures revealed the formation of spherical shaped ELR aggregates or oligomers as reported for other proteins ([4] [5]). Furthermore, addition of a small amount of glutaraldehyde crosslinker (0.1%) to the ELR solution containing Ca ions resulted in the formation of additional, larger structures (ranging from 300 nm to 2000 nm) with a further increased β -sheet conformation (48.2%) (**Supplementary Fig. 6f, g**). These results corroborate the critical role of Ca ions and crosslinking in the formation of supramolecular ELR structures in solution.

The methodology used for carrying out DLS analysis on ELR solutions is described in the methods section of the revised manuscript on **Page 23** as shown below:

Dynamic light scattering (DLS) analysis:

Dynamic light scattering (DLS) experiments were carried out using Zetasizer nano-ZS90 (Malvern, Inc.) instrument at room temperature (25 °C). ELR solutions (1 mg/mL) without and with 1.5 mM Ca ions and 0.1% glutaraldehyde were prepared in ethanol/water (9/1) solvent mixture and analysed using a laser wavelength of 633 nm and a scattering angle of 90°.

(b) More confusingly, the small-angle data (Figure S7) show that the initial slope is nearly 2, suggesting that the film contains more of lamellar structures rather than fibril.

Response: We respectfully disagree with this comment. We have analysed again our SAXS spectrum in SASview software and confirmed that the initial powerlaw slope is 4.1 (**Supplementary Fig. 8a, b**) rather than 2. As we stated in our previous version of the manuscript, this slope of 4.1 points towards a large surface fractal nanostructure with smooth surfaces and correlates with our SEM observations (**Fig. 1a**) depicting a network of smooth fibrils.

(c) The WAXS data show several sharp peaks, but the authors have only identified the peak at 0.48 nm. Why have the other peaks not been identified?

Response: We thank the Reviewer for pointing this out. In the WAXS data, we identified only the key peaks at $\sim 10\text{\AA}$ and 4.7\AA that suggest the presence of fibrillar structures (PNAS, 2021, 118(48), e2112783118). We have now performed WAXS analysis on calcium chloride only and confirmed that the additional peaks in **Fig. 1b** are from calcium chloride crystals. We have now incorporated a WAXS spectrum of calcium chloride crystals in **Supplementary Fig. 8c** and clarified this in the manuscript on **Page 3** as shown below:

Additional diffraction peaks in the WAXS spectrum (**Fig. 1b**) can be attributed to the crystalline structures of calcium chloride (**Supplementary Fig. 8c**).

(d) Furthermore, if the solvent composition and evaporation conditions (including concentration, time, environmental humidity, and substrate etc.) affect the assembly of ELR into films, why is there no structural characterization under corresponding conditions?

Response: Thank you for this important suggestion. We have now performed new experiments focussed on the use of the ethanol/water solvent mixture and investigated the effect of different ratios of ethanol in the mixture, drying speed, crosslinker concentration, and different substrates to deposit the ELR material on the secondary structure conformation of the ELR coatings. We believe that this new data, together with previous results on structural and functional characterization of ELR coating, demonstrate the robustness and potential translation of our technology. We have now included these results in **Supplementary Fig. 43** and elaborated in the **Supplementary Discussion 9** as shown here:

The SEM results demonstrated similar ELR fibrillar structures when prepared using both DMF/DMSO (9/1 ration) (**Fig. 1a**) and ethanol/water (9/1 ration) (**Supplementary Fig. 42a**) solvent mixtures. Furthermore, ELR coatings prepared from both solutions exhibited similar secondary structure conformations (**Supplementary Fig. 3** and **Supplementary Fig. 42b**), leading to comparable mineralization outcomes including recreation of structure and restoration of properties (**Supplementary Fig. 42c – f** and **Fig. 3d – f**). Thus, motivated by these results, we further investigated the effect of different ratios of ethanol/water solvent mixture, drying speed, crosslinker concentration, and different substrates to deposit the ELR material on the secondary structure conformation of the ELR coatings. First, we explored the effect of varying the concentration of ethanol (0, 25, 50, and 85%) in the ethanol/water solvent mixture on the secondary structure conformation of ELR coating. The results demonstrated that increasing ethanol in the ethanol/water mixture increases the drying speed of the ELR coatings (**Supplementary Fig. 43a**) but without affecting their secondary structure conformation (**Supplementary Fig. 43b**). Furthermore, no differences were observed in the secondary structure conformation of the ELR coating when prepared using varying concentrations of glutaraldehyde crosslinker (0.1, 0.5, 1.0, and 1.5%) (**Supplementary Fig. 43c**). Also, we did not observe any differences in the secondary structure composition of the ELR coatings when prepared on top of different substrates including enamel and PDMS (**Supplementary Fig. 4**).

Overall, these results demonstrate the robustness of our technology in preserving the structural properties of ELR coatings independently of the type of solvent used, ethanol concentration, drying rate, crosslinker concentration, or substrate used. Furthermore, structural properties are essential for defining the resulting functional performance, and the ability to maintain these properties across various environmental conditions represents an important translational advantage of our technology.

3. (a) The introduction section of the paper does not seem to be well crafted and lacks the posing of key scientific questions that could attract a wide readership of Nature Materials, not only clinicians but also researchers in the fields of fundamental materials science. From a fundamental research perspective, the paper does not clearly categorize or accurately describe previous studies on enamel regeneration, which makes the discussion somewhat superficial.

Response: We thank the reviewer for this suggestion. We have now revised the introduction section to better highlight key scientific questions of interest to the materials science community and previous studies on enamel regeneration. This is on **Page 2** as shown below:

The recreation of dental enamel has been a major goal in materials science [15]. Methods based on acidic [16], hydrothermal [17], and high-power laser [18] treatments have enabled the ambient deposition of mineral but without the capacity to recreate the architecture and function of healthy enamel. Recently, several enamel analogs comprising organic-inorganic nanocomposites such as polyacrylic acid-zinc oxide nanowires [19] and polyvinyl alcohol-HAp nanowires [20] have been reported to generate functional hierarchical architectures. However, these technologies rely on non-physiological fabrication conditions, which restrict their applicability in clinical settings [21].

To address the clinical need, bioinspired approaches offer promising alternatives to regrow different anatomical regions of enamel [22]. For instance, Moradian-Oldak and colleagues have developed amelogenin-based scaffolds capable of growing layers of aprismatic enamel-like structures *ex vivo* [9]. Other approaches based on phase-transited lysozyme films with amelogenin-like peptides [23], amelogenin-derived shortened ADP5 (shADP5) oligopeptide [24], modified leucine-rich amelogenin peptide (mLRAP) [25], phosphorylated and phosphonated peptides [26], or triethylamine stabilized calcium phosphate ions [27] have reported remineralization of prismatic enamel. However, these strategies suffer from drawbacks that restrict their clinical translation such as having toxic and noxious components (e.g., triethylamine in Ref. [27]), time-consuming application processes (e.g., 30 min in Ref. [23], 15 min in Ref. [27], 12 hours in Ref. [26], 10 min/day in Ref. [28]), partial recovery of architecture and functional properties (e.g., aprismatic enamel in Ref. [9], prismatic enamel in Ref. [23], [25], [26], [27]), and limited control of the mineralization process. Thus, remineralizing technologies that recreate the diverse architecture and spectrum of functional properties of dental enamel in a patient and clinically-friendly manner remain an unmet challenge.

(b) I also highly agree with the reviewers' comments—the conclusions are somewhat exaggerated. While the article demonstrates the ability to regenerate a few micrometers of enamel, mature cavities typically range in diameter from hundreds of micrometers to a few millimeters, several orders of magnitude larger than the remineralized layer. It cannot be used to treat early lesions as they are subcutaneous. Some findings in the manuscript might lead to useful techniques, but at present, it is not a long-term mineralization solution.

Response: We respectfully disagree with this point based on the following reasons.

First, Reviewer #2 provided this comment on our previous version of the manuscript, which we agreed with and have carefully addressed to ensure that the conclusions are accurate and realistic.

It is important to point out that we have not claimed that our technology would treat mature cavities or sub-surface lesions. Instead, we have clarified that our technology offers a unique opportunity to prevent and treat enamel erosion and to treat non-cavitated lesions by regenerating thin (up to ~10 μm) layers on sections of enamel. This application will be clinically significant since all cavities begin as non-cavitated lesions and stopping their progression would prevent their exacerbation into cavitated lesions. Our technology could be used in combination with early caries detector stains that recently have been brought to market (e.g., Blue Check by Incisive Technologies, <https://www.incisive-technologies.com/product>). These stains detect early stages of enamel demineralization which may be reversed through treatments with our ELR coatings.

In the previous rebuttal, we stated that our technology can realistically offer an alternative with the capacity to grow thin, crystallographically integrated, and functional mineralized layers on enamel or dentine. These layers could (i) serve as protective layers against tooth attrition or erosion on outer aprismatic enamel, (ii) repair micron-scale surface lesions on aprismatic or prismatic enamel, and (iii) serve as protective and functional layers on dentine surfaces for example to prevent or treat hypersensitivity. This capability far surpasses current commercial dental remineralization alternatives and overcomes key translational obstacles that have up to date prevented the capacity to regrow enamel tissue in patients, even as thin functional layers. Its ease of use may also allow for preventive applications in patients with high caries risk and could be applied at regular intervals to reduce or reverse enamel demineralization processes.

We apologise that we incorrectly mentioned treatment of caries in the last line of the abstract, this has been corrected now in the revised manuscript, as shown below:

The study demonstrates the translational potential of our innovative mineralizing technology for treating loss of enamel in clinical settings such as the treatment of enamel erosion and dental hypersensitivity.

(c) From responding letter, authors seem unwilling to conduct animal experiments for validation.

Response: Please let us elaborate on this. In our first comment to Reviewer #1, we have expanded on the rationale for not conducting the animal experiments. Briefly, we have encountered critical limitations regarding the testing of our technology in animals. We are aware of some *in vivo* studies that have been conducted as mentioned by Reviewer #1 (Adv. Mater., 2020, 32(31), 2002080 and Acta Biomater., 2014, 10(10), 4437-4446). However, these are not standard nor well-accepted models, particularly in our region and have therefore been advised against them based primarily on ethical and functional considerations. However, we completely agree with the Reviewer on the importance to validate the potential translation of our technology and have therefore conducted additional experiments to assess the mineralization capacity of our platform using natural saliva from different human donors. We have elaborated on these new results in the response to Comment 1 of Reviewer #1, as shown below:

Furthermore, to test the capacity to mineralize in natural human saliva, ELR coated enamel samples were exposed to saliva from three different donors for 2 weeks. Again, the results revealed the capacity of the ELR matrix to recreate prismatic enamel over large surface areas (**Fig. 6j, Supplementary Fig. 45a**) and restore both the mechanical ($E = 74.6 \pm 15.3$ GPa, $H = 2.7 \pm 1.1$ GPa) (**Fig. 6k, Supplementary Fig. 45c**) and chemical (**Supplementary Fig. 45d, e**) properties of natural enamel. These results confirm the robust mineralizing capacity of our ELR platform by enabling a similar epitaxial growth despite the inherent compositional complexity of natural saliva. Furthermore, the structure and properties of enamel were recovered on all tested samples, independently of variations in free Ca ion concentrations in the saliva from each donor (**Supplementary Fig. 45b**).

(d) While the author claims that ethanol/water can replace DMF/DMSO in a more biocompatible manner, the structural and mineralization capabilities of ELR films obtained in

ethanol/water systems need comprehensive and detailed evaluation, rather than just comparing modulus and hardness.

Response: We agree with the Reviewer and have therefore performed new experiments to further investigate performance of our technology using the ethanol/water solvent. We confirmed that the ELR coatings exhibit similar structural properties including secondary structure conformations when prepared under different conditions such as varying ratios of ethanol in the solvent mixture, drying speed, and crosslinker concentrations. These results are now included in **Supplementary Fig. 43** and elaborated in the **Supplementary Discussion 9** as shown here:

The SEM results demonstrated similar ELR fibrillar structures when prepared using both DMF/DMSO (9/1 ration) (**Fig. 1a**) and ethanol/water (9/1 ration) (**Supplementary Fig. 42a**) solvent mixtures. Furthermore, ELR coatings prepared from both solutions exhibited similar secondary structure conformations (**Supplementary Fig. 3** and **Supplementary Fig. 42b**), leading to comparable mineralization outcomes including recreation of structure and restoration of properties (**Supplementary Fig. 42c – f** and **Fig. 3d – f**). Thus, motivated by these results, we further investigated the effect of different ratios of ethanol/water solvent mixture, drying speed, crosslinker concentration, and different substrates to deposit the ELR material on the secondary structure conformation of the ELR coatings. First, we explored the effect of varying the concentration of ethanol (0, 25, 50, and 85%) in the ethanol/water solvent mixture on the secondary structure conformation of ELR coating. The results demonstrated that increasing ethanol in the ethanol/water mixture increases the drying speed of the ELR coatings (**Supplementary Fig. 43a**) but without affecting their secondary structure conformation (**Supplementary Fig. 43b**). Furthermore, no differences were observed in the secondary structure conformation of the ELR coating when prepared using varying concentrations of glutaraldehyde crosslinker (0.1, 0.5, 1.0, and 1.5%) (**Supplementary Fig. 43c**). Also, we did not observe any differences in the secondary structure composition of the ELR coatings when prepared on top of different substrates including enamel and PDMS (**Supplementary Fig. 4**).

Overall, these results demonstrate the robustness of our technology in preserving the structural properties of ELR coatings independently of the type of solvent used, ethanol concentration, drying rate, crosslinker concentration, or substrate used. Furthermore, structural properties are essential for defining the resulting functional performance, and the ability to maintain these properties across various environmental conditions represents an important translational advantage of our technology.

It is also important to mention that in the previous version of our manuscript, a significant amount of data on functional characterization of the ELR coatings prepared using ethanol/water solvent had been already included as summarised here:

- We confirmed biocompatibility of the ELR coatings (**Supplementary Fig. 36 - 39**) by assessing the material's effect on the viability and growth of three different cell types including mouse fibroblast cell line 3T3, human immortalised mesenchymal stem cells (MSCs), and human umbilical vein endothelial cells (HUVECs).
- We also assessed the ELR coating based on its functional performance including:
 - Capacity to deposit a uniform ELR coating over large areas and convoluted tooth structures in under 4 minutes (**Supplementary Fig. 41**).

- Stability of the ELR coating against realistic insults including exposure to extreme pH, salt solutions, extensive tooth brushing, sonication, high temperature, and detachment using a standard tape peeling test (**Supplementary Fig. 46, 47, 48**).
- Capacity of the ELR coating to remineralize in artificial saliva after exposure to extensive toothbrushing (**Fig. 6l, m, n**), sonication (**Supplementary Fig. 47a**) extreme pH (**Supplementary Fig. 46a, b**), salt solutions (**Supplementary Fig. 46c, d**), high temperature (**Supplementary Fig. 47b**), and peeling treatment (**Supplementary Fig. 47c**).
- Capacity of the ELR coating to remineralise enamel under conditions that imitate real oral environments such as presence of salivary enzymes and abrasive conditions (**Supplementary Fig. 44**).

Overall, we believe that our new results on structural characterization, combined with the previous data on the functional performance of ELR coatings prepared using ethanol/water solvent, demonstrate the robustness of our technology, which is a key translational advantage.

Specific comments:

1. The manuscript lacks discussion on the meticulous removal of the ELR matrix via enzymatic digestion, a step not commonly practiced in clinical dental procedures. It would be beneficial to elaborate on how this organic matrix removal affects biomineralization and provide a detailed analysis of how ELR fibrils mediate epitaxial remineralization.

Response: We thank the Reviewer for pointing this out and apologize for the confusion. To explain here briefly, the ELR coating is biodegradable under natural physiological conditions and would not require an enzymatic digestion treatment in the dental practice. Enzymatic digestion was only employed to expedite the complete removal of the ELR coating and expose the underlying remineralized layer. We have now made this point clearer on Page 8 of the revised manuscript, as shown below:

Page 8: While the ELR matrix slowly degrades during the mineralization process, it remains stable and functional during the enamel remineralization process (**Supplementary Fig. 24** and **Supplementary Discussion 5**). To expedite this ELR degradation process and expose the underlying remineralized layer, we treated the samples with elastase after the remineralization process was completed (**Fig. 3d, 3e, Supplementary Fig. 21, 22, 23**).

Also, we have now conducted a number of new experiments to investigate the interplay between ELR degradation and mineralization process (**Supplementary Fig. 24**). These new results combined with previous data demonstrate that the ELR coating is functional and stable enough to enable complete mineralization within a 2 week period in both artificial (**Supplementary Fig. 15, 44**) and natural (**Fig. 6j, k** and **Supplementary Fig. 45**) saliva while slowly undergoing degradation. To describe this process in detail, we have now presented new results in **Supplementary Fig. 24** and elaborated them in **Supplementary Discussion 5** as shown below:

We have previously described how the ELR molecules assemble into a dense matrix with an ELR conformation that enables the growth of highly organised mineralized structures [1]. This study also demonstrated that this densely packed and crosslinked ELR matrix remains present during the mineralization process and can be completely degraded using elastase

enzymes, demonstrating that the ELR matrix is both necessary to trigger organized mineralization and biodegradable.

To further investigate this mineralization/degradation process, we first conducted swelling experiments by exposing the ELR matrix to a mineralizing pH = 6 condition, which revealed a 22% swelling in this condition (**Supplementary Fig. 24a**). We hypothesized that this swelling would enhance ELR matrix degradation and thus we performed degradation tests by exposing the ELR matrix to a mineralization solution in the absence or presence of salivary enzymes for 3 weeks. The results confirmed that the ELR matrix degraded by 14.8 ± 0.4 wt% and 18.5 ± 0.2 wt% in the absence and presence of salivary enzymes, respectively (**Supplementary Fig. 24b**).

We then conducted a thermogravimetric analysis (TGA) to assess differences in the degradation of the organic ELR matrix before and after mineralization. The results further confirmed a loss of ~15 wt% of the ELR matrix during the period of mineralization (**Supplementary Fig. 24c**), which corresponds to the results from the degradation experiment (**Supplementary Fig. 24b**). In addition, given this swelling and degradation of the matrix as well as the critical role of the secondary structure of the ELR for mineralization, we then conducted FTIR experiments to assess changes during the mineralization process. The results revealed that the secondary structure conformation of the ELR did not change significantly during the mineralization process (**Supplementary Fig. 24d**).

Overall, these results demonstrate that the ELR matrix undergoes slow degradation but remains both stable and functional (i.e., maintaining its secondary structure conformation) during the distinctive epitaxial and organized mineralization in both artificial saliva (**Supplementary Fig. 15, 44**) and natural saliva (**Fig. 6j, k**, and **Supplementary Fig. 45**) conditions.

We believe that this new data, together with our previous data including stability of the ELR coating against diverse physical and chemical treatments (**Supplementary Fig. 46, 47, 48**), provide compelling evidence of the stability of the ELR matrix to be both functional and biodegradable.

With regards to the mechanism behind the epitaxial mineralization, we have provided experimental (**Fig. 2a**) and computational evidence (**Supplementary Fig. 11**) demonstrating epitaxial mineralization on both a single apatite crystal and the prismatic enamel surface (**Fig. 4**). Additionally, we have summarized the key mechanistic features on epitaxial mineralization in the **Supplementary Discussion 11**. In conclusion, we have provided more evidence on epitaxial mineralization (experimental and computational) compared to those presented in the seminal studies (experimental only) including *Sci. Adv.*, 2019, 5(8), eaaw9569 and *Adv. Mater.*, 2020, 32(31), 2002080.

Also, we have repeated the experiment to acquire TEM images with better quality and performed FFT analysis, confirming epitaxial growth along c-axis. These results are now presented in **Fig. 2a** and **Supplementary Fig. 13** of the revised manuscript.

2. The choice of a mineralization solution containing fluoride for HAP growth should be

justified. Previous literature suggests that fluoride at certain concentrations can also induce oriented growth of HAP (Cryst. Growth Des. 2018, 18, 4, 2279–2288).

Response: Thank you for raising this important point. We have now conducted mineralization studies using a variety of mineralization solutions to help demonstrate the robustness of our technology. Please let us elaborate.

First, we have selected a fluoride concentration of 84 ppm in the mineralization solution which is used as a standard for conducting mineralization processes described in previous studies (Cryst. Growth Des., 2006, 6, 1504–1508 and Nat. Commun., 2018, 9:2145). In addition, to approximate the amounts of F in natural saliva, we also conducted experiments using a mineralization solution containing 1 ppm F, which has been reported to be the average amount in natural saliva (BMC Oral Health, 2012, 12:3) and in previous reports on enamel remineralization using artificial saliva (ACS Omega, 2018, 3, 2546-2557; Biomaterials, 2009, 30, 478-483) (**Supplementary Fig. 15a, b**). To further assess the performance of our technology in a realistic scenario, we have now included mineralization results using natural saliva from different donors (**Fig. 6j, k**, and **Supplementary Fig. 45**).

However, it is important to mention that we also tested our technology using a mineralization solution without any fluoride ions (**Supplementary Fig. 30**), and again the results revealed the recreation of prismatic region of enamel. We have now clarified the rationale behind the amounts of F used on **Page 7** of the revised manuscript as shown here:

Page 7: In all three cases, similar results were obtained using artificial saliva (1.2 mM Ca²⁺, 0.72 mM PO₄³⁻) containing 1 ppm F ions to imitate physiological conditions (**Supplementary Fig. 15**). We used 1 ppm of F concentration in artificial saliva, which has been reported to be the average amount present in natural saliva ranging between 0.02 ppm to 1.93 ppm [49].

3. The TEM results (Fig. 2a and Sfig.10) show that lattice lines grown with ELR coating appear less clear compared to those without ELR coating. Exploring the reasons behind this phenomenon is necessary.

Response: Thank you pointing this out. The lattice lines grown with ELR coating (**Fig. 2a**) appear less clear because the crystals are embedded within the ELR matrix and interfere with the imaging. Also, the ELR matrix is beam-sensitive so it can experience minor damages due to ionization and electrostatic charging, which is a common phenomenon reported widely for poorly conducting organic samples (Micron, 2019, 119, 72-87). These are the reasons why we included a large TEM image in **Supplementary Fig. 13** that shows the lattice lines in the crystalline region. We have now included this explanation in the caption of **Fig. 2a** in the revised manuscript, as shown below:

The lattice lines in the apatite nanocrystal grown within the ELR coating appear blurry due to imaging interference and potential beam-induced damages caused by ionization and electrostatic charging of the beam-sensitive ELR matrix, a common issue for poorly conducting organic samples.

Also, we thank the Reviewer for pointing this out and thus have repeated the experiment to capture a better quality TEM image showing more distinct lattice lines. This is now presented in **Fig. 2a** of the revised manuscript. We have also included its large TEM image in **Supplementary Fig. 13**.

4. It is known that phosphorylated amino acid residues are crucial for enamel protein mineralization (PNAS, 2022, 119, 19, e2106965119), but this aspect is not covered in the current paper.

Response: We thank the Reviewer for this suggestion and agree that phosphorylated serine in natural mineralizing proteins plays an important role in binding mineral ions during mineralization. However, our ELR molecule does not have any serine, tyrosine, nor threonine residues that can undergo phosphorylation during post translational modification. Instead, the highly acidic statherin-derived analog DDDEEKFLRRIGRFG present in our ELR molecule is the key mineralizing motif. However, the main mechanistic novelty of our study is the possibility to trigger such epitaxial and organized remineralization by tuning the secondary structure of the protein. Therefore, we feel that investigating the role of phosphorylation in our ELR molecule falls outside of the scope of the current work.

5. The absence of animal experiments is a notable limitation. While in vitro experiments demonstrate effectiveness, the efficacy of the method in the complex oral environment of humans remains unknown. Discussion on potential hindrances to mineral formation, such as a saliva-derived layer containing proteins like casein, would be valuable.

Response: We have now elaborated on this in detail in response to Comment 1 of the Reviewer #1 and in response to the current Reviewer on Page 15.

Briefly, we completely agree with the Reviewer on the importance to further validate the potential translation of our technology. This is a major focus of our study and have therefore conducted additional experiments to assess the mineralization capacity of our platform using natural saliva from different human donors. The results confirmed the restoration of prismatic structures (**Fig. 6j** and **Supplementary Fig. 45a**), mechanical (stiffness and hardness) (**Fig. 6k** and **Supplementary Fig. 45c**), and chemical properties (**Supplementary Fig. 45d, e**) of enamel. We hope that the Reviewer agrees that this experiment provides strong evidence for translation given the rich composition and variable nature of human saliva (**Supplementary Fig. 45b**) and that it serves as an alternative to the animal experiment. We have reported these results on **Page 16** of the revised manuscript as follows:

Furthermore, to test the capacity to mineralize in natural human saliva, ELR coated enamel samples were exposed to saliva from three different donors for 2 weeks. Again, the results revealed the capacity of the ELR matrix to recreate prismatic enamel over large surface areas (**Fig. 6j, Supplementary Fig. 45a**) and restore both the mechanical ($E = 74.6 \pm 15.3$ GPa, $H = 2.7 \pm 1.1$ GPa) (**Fig. 6k, Supplementary Fig. 45c**) and chemical (**Supplementary Fig. 45d, e**) properties of natural enamel. These results confirm the robust mineralizing capacity of our ELR platform by enabling a similar epitaxial growth despite the inherent compositional complexity of natural saliva. Furthermore, the structure and properties of enamel were recovered on all tested samples, independently of variations in free Ca ion concentrations in the saliva from each donor (**Supplementary Fig. 45b**).

The methodology used for remineralizing enamel using natural saliva is described in the methods section of the revised manuscript on **Page 25**:

Mineralization using natural saliva

Human saliva was collected from 3 different healthy donors after approval by the University of Nottingham Ethics Committee, reference number FMHS 313-0721. The donors were between 20 and 40 years old and had healthy periodontal conditions with no signs of gingivitis, periodontitis, gingival recession, carious decay, or dental erosion. Donors were instructed to refrain from eating or drinking (except water) for two hours before saliva collection. Unstimulated saliva samples were collected directly into polypropylene tubes. For enamel remineralization, ELR coated enamel sections were separately exposed to 1 mL of whole human saliva collected individually from the different donors for 2 weeks. Enamel sections were incubated at 37 °C and the saliva was replenished every 3 hours during a 12-hour daytime period and then left in the same spent saliva for the subsequent 12 hours overnight. Post mineralization, ELR coating was enzymatically degraded using elastase as described in the section 'Mineralization experiment'. Calcium ion concentrations in the saliva samples were measured using a colorimetric assay kit (Abcam, ab102505), according to the manufacturer's instructions.

6. It is essential to further investigate the stability and bonding strength characteristics of the ELR fibril coating on the tooth surface.

Response: We agree with the Reviewer on the importance of this parameter and have therefore conducted new experiments to confirm the bonding and stability of the ELR coating on enamel. Briefly, our profilometry analysis showed no reduction in the thickness of the ELR coating after 5 minutes of toothbrushing, confirming the stability of the ELR coating against physical wear. These results are now included and elaborated in **Supplementary Fig. 50** as shown here:

An ELR coating on an enamel surface was prepared using the ethanol/water solvent mixture and assessed for stability and bonding strength. Profilometry analysis revealed that there was no significant reduction in the ELR coating thickness after toothbrushing for 5 minutes (imitating 1 month of brushing), confirming its stability against physical wear.

The methodology used for carrying out profilometer analysis on ELR coated enamel sample is described in the methods section of the revised manuscript on **Page 25** as shown below:

Profilometer analysis

The ELR coating on the enamel sample was prepared using the ethanol/water solvent mixture as described in the section 'Mineralization experiment'. Briefly, 5% (w/v) ELR solution was prepared in ethanol/water (75/25) solvent mixture followed by the addition of 1.5 mM Ca ions and 0.5% (v/v) glutaraldehyde. The ELR solution was deposited on an acid etched enamel sample and allowed to dry for 4 minutes. For profilometer analysis, a long scratch mark was made on the ELR coating at the centre of the sample. A height profile was created along the scratch mark using profilometer Alicona G5 XL optical profilometer (Bruker, Germany) before and after brushing for 5 minutes.

Also, it is important to mention that in the previous version of our manuscript, a large number of functional tests on the stability of the ELR coatings had been already included as summarised here:

- We tested the stability of the ELR coating against realistic insults including exposure to extensive toothbrushing (**Fig. 6 l, m, n**), sonication (**Supplementary Fig. 47a**) extreme pH (**Supplementary Fig. 46a, b**), salt solutions (**Supplementary Fig. 46c, d**), high

temperature (**Supplementary Fig. 47b**), and a peeling treatment (**Supplementary Fig. 47c**). Some of these techniques are standard and have been reported previously by other groups as a way to assess stability and bonding (Adv. Mater., 2020, 32(31), 2002080).

- We demonstrated that these insults did not affect the physical properties (i.e., stiffness, hardness) (**Supplementary Fig. 48a, b**), coating texture (**Supplementary Fig. 48c**), and functional properties (i.e., ability to remineralize enamel) (**Fig. 6 l, m, n** and **Supplementary Fig. 46, 47**) of the ELR coatings.

7. After epitaxial remineralization, the fate of the ELR fibril coating is unclear. It would be insightful to discuss whether they remain on the newly formed enamel rod surface or undergo other changes.

Response: We thank the Reviewer for pointing this out. Towards this goal, we performed new experiments to understand the interplay between ELR degradation and mineralization. We have elaborated on this in response to Comment 2 from Reviewer #1. Briefly, we have conducted a number of new experiments to investigate the degradation process and provide compelling evidence of the stability of the ELR matrix to be both functional and biodegradable. We have presented these new results in **Supplementary Fig. 24** and elaborated them in **Supplementary Discussion 5** as shown here:

We have previously described how the ELR molecules assemble into a dense matrix with an ELR conformation that enables the growth of highly organized mineralized structures [1]. This study also demonstrated that this densely packed and crosslinked ELR matrix remains present during the mineralization process and can be completely degraded using elastase enzymes, demonstrating that the ELR matrix is both necessary to trigger organized mineralization and biodegradable.

To further investigate this mineralization/degradation process, we first conducted swelling experiments by exposing the ELR matrix to a mineralizing pH = 6 condition, which revealed a 22% swelling in this condition (**Supplementary Fig. 24a**). We hypothesized that this swelling would enhance ELR matrix degradation and thus we performed degradation tests by exposing the ELR matrix to a mineralization solution in the absence or presence of salivary enzymes for 3 weeks. The results confirmed that the ELR matrix degraded by 14.8 ± 0.4 wt% and 18.5 ± 0.2 wt% in the absence and presence of salivary enzymes, respectively (**Supplementary Fig. 24b**).

We then conducted a thermogravimetric analysis (TGA) to assess differences in the degradation of the organic ELR matrix before and after mineralization. The results further confirmed a loss of ~15 wt% of the ELR matrix during the period of mineralization (**Supplementary Fig. 24c**), which corresponds to the results from the degradation experiment (**Supplementary Fig. 24b**). In addition, given this swelling and degradation of the matrix as well as the critical role of the secondary structure of the ELR for mineralization, we then conducted FTIR experiments to assess changes during the mineralization process. The results revealed that the secondary structure conformation of the ELR did not change significantly during the mineralization process (**Supplementary Fig. 24d**).

Overall, these results demonstrate that the ELR matrix undergoes slow degradation but remains both stable and functional (i.e., maintaining its secondary structure conformation)

during the distinctive epitaxial and organized mineralization in both artificial saliva (**Supplementary Fig. 15, 44**) and natural saliva (**Fig. 6j, k, and Supplementary Fig. 45**) conditions.

To determine if the ELR coating underwent any changes after enamel mineralization, we conducted FTIR analysis and confirmed that the secondary structure conformation of the ELR coating remained largely unchanged before and after mineralization (**Supplementary Fig. 24 d**).

Point-by-point response to Reviewers:

Reviewer #1

I appreciate the authors' comprehensive and detailed response. While they have provided a rationale for not conducting *in vivo* experiments to validate the clinical potential of their ELR coating, the *in vitro* experiments using human saliva, though valuable, do not fully capture the complexity of the dynamic oral environment. Specifically, factors such as mechanical stresses (including chewing, friction, and fluid flow) play a crucial role in the real clinical performance of coatings and remineralization processes. Since the authors emphasize the clinical applicability of their method as a key advantage, I recommend explicitly acknowledging the limitations of the current experimental design within the manuscript. Doing so would enhance the transparency of the study and allow readers to better evaluate the translational potential of the findings.

Response: We thank the reviewer for the constructive feedback. We agree with the reviewer that while we have attempted to conduct *in vitro* studies that incorporate accessible features such as enzymes, mechanical forces, pH variations, natural saliva, etc, these do not fully recreate the complexity of the *in vivo* oral environment. Therefore, we have now included the following text on Page 17 acknowledging the limitations of not having an *in vivo* experiment in our study:

Moreover, our findings demonstrate a capacity to grow enamel-like structures under conditions that closely imitate various mechanical and chemical challenges found in the mouth. However, these tests do not fully recreate the complexity of the *in vivo* oral environment and, consequently, to fully confirm the capacity to regenerate natural enamel would require *in vivo* validation, which we envision to pursue in future work.

Reviewer #3

The authors have provided a comprehensive and well-structured rebuttal, addressing all concerns raised in the previous review rounds. The additional experiments, particularly those related to the ELR matrix degradation, mineralization in natural saliva, and structural validation, have significantly strengthened the scientific rigor of the study. The new data effectively demonstrate the stability, functionality, and translational potential of the ELR coating.

Based on the substantial revisions and the robust new experimental data, I am satisfied that the authors have adequately addressed all concerns.

Response: We thank the reviewer for appreciating our responses and for the constructive feedback, which significantly helped us improve both our study and manuscript.

Reviewer #4

(A) After carefully reading the authors' response letter and the revised manuscript, I believe that, compared with the previous version, this work does not provide substantial evidence to support the authors' claim that ELR can effectively regulate the epitaxial growth of enamel.

Response: We thank the reviewer for the constructive comments throughout the review process, which have helped us improve the scientific rigor and quality of our work. We believe that we have now demonstrated epitaxial mineralization within our ELR matrix and respectfully request the opportunity to summarize this evidence:

While our goal for all of the different aspects of our study has been to push the boundaries of the state-of-the art, it is first important to mention that we have assessed and demonstrated epitaxial mineralization using the same well-accepted methods reported by leading papers in the field. These are:

- High-Resolution TEM (HRTEM) to visualize the atomic lattice continuity (integration), confirming epitaxial mineralization. Used by Ref. [23 and 27].

- Selected Area Electron Diffraction (SAED) to visualize similar crystallographic orientation between new and native nanocrystals. Used by Ref. [27].
- Scanning Electron Microscopy (SEM) to provide low-resolution visualization of crystal morphology and integration with native enamel. Used by Ref. [23], *Materials & Design*, 2023, 226, 111654, and *Advanced Materials*, 2024, 36(16), 2311659.

Inspired by these studies, we have conducted similar experiments, which together confirmed epitaxial growth. We have described the mechanism behind this ELR mediated epitaxial remineralization in **Supplementary Discussion 11** and provided specific evidence in the manuscript in the following sections:

- HRTEM (**Fig. 2a** and **Supplementary Fig. 13**) and SAED (**Fig. 4a (inset)** and **4j (inset)**) images showing remineralization of the apatite nanocrystals embedded within the ELR matrix and demonstrating epitaxial crystal growth preferentially along the c-axis.
- FFT patterns from single crystal growth demonstrating co-orientation between new mineral and native crystal and confirming epitaxial mineralization (**Fig. 2a (inset)**, **Fig 4c (inset)** and **4i (inset)**).
- SEM images showing new crystals growing from diazone (**Fig. 3d**) and parazone prisms (**Fig. 3e**) following the native crystallographic orientation and growing through and within the ELR coating.
- TEM images showing new crystals growing preferentially along the c-axis of the native enamel crystals (**Fig. 4a** and **h**).
- Simulation results confirming the preferential binding of the ELR fragments to the a-axis of the crystals instead of the c-axis of the crystals, thus, energetically favouring crystal growth along the c-axis (Ref. [51] and PNAS, 2022, 119(19), e2106965119) (**Supplementary Fig. 11**) as evident in **Fig. 2a**.

Beyond this direct evidence for epitaxial mineralization, we have also generated relevant indirect evidence. Specifically, crystallographic integration resulting from epitaxial growth from the natural enamel tissue and within the ELR matrix can be inferred by assessing stability under a variety of mechanical and functional tests.

- First, the remineralized enamel exhibited high wear strength (**Fig. 3h**), suggesting that the regenerated mineralized layer resists friction-induced delamination as a result of its integration to the underlying tissue.
- Second, the remineralized enamel retained its structural integrity and mechanical strength after simulating 1 year of toothbrushing, which again suggests integration with the underlying enamel (**Fig. 6a, b, Supplementary Fig. 33**).
- Third, when simulating 3.5 years of mastication, the remineralized enamel maintained its structural integrity (**Fig. 6e, f, Supplementary Fig. 35**), again indicating high integration between the remineralized enamel and the underlying tissue.
- Finally, the remineralized enamel displayed similar apparent fracture toughness (resistance to crack initiation and propagation) as healthy enamel, which once more hints at its crystallographic integration with the underlying natural enamel (**Fig. 6c, d, Supplementary Fig. 34**).

These tests offer indirect evidence that the mineralized layer is highly integrated to the natural tissue. If the mineralized layer were not integrated, these tests would lead to its delamination.

(B) Moreover, in terms of the repair effect, compared with other enamel repair studies, this work does not offer a strong improvement to justify its publication in a high-impact journal such as

Nature Communications. Therefore, I suggest that this work is more suitable for publication in a more specialized journal.

Response: We agree with the reviewer on the importance to demonstrate the novelty and clear advances of our study compared to other leading studies. Please allow us to elaborate and make comparisons with key leading studies:

To the best of our knowledge, nobody so far has designed protein matrices that recreate the supramolecular organization of the natural amelogenin matrix (based on current understanding) to the level and in-depth analysis that we have (using both computational and experimental work as well as multiscale design, engineering, translational potential, and validation). In our study, we not only demonstrate the capacity to recreate structural features of this protein matrix (**Fig. 1a - e**) but also demonstrate its functionality by enabling epitaxial and organized mineral growth (**Fig 1f; Fig. 2d, g, i; Fig. 3d, e**).

Another important novelty of our study is the capacity to remineralize all the different anatomical regions of enamel including prismatic, interprismatic, and aprismatic regions (**Fig. 2d, g and Fig. 3d, e**) while restoring tissue stiffness, hardness, toughness, coefficient of friction, and wear strength (**Fig. 3f, g, h**). Furthermore, we demonstrated the stability of the remineralized layer under conditions that emulate daily abrasion from toothbrushing (**Fig. 6a, b and Supplementary Fig. 33**), tooth wear from chewing and grinding (**Fig. 6e, f, g**), and stability against acid exposure from dietary habits (**Fig. 6h, i, j**). In addition, we have also demonstrated the capacity of our ELR coating to mineralize in human saliva from different donors, recreating the microarchitecture and restoring the mechanical and physio-chemical properties of enamel. These results are presented in **Fig. 6 j, k and Supplementary Fig. 46** of the revised manuscript.

Beyond the capacity to recreate the structure/function of the amelogenin matrix and the structure and function of enamel, another key novelty of our study is the potential for clinical translation and usability of our technology. For translation, it is critical to conduct studies that test the stability and functionality of both the ELR coating and the remineralized enamel under a range of extreme physiological conditions and experiences of daily life. In our study, we demonstrate that the ELR coating is stable against physical insults including toothbrushing (**Fig. 6l, m, Supplementary Fig. 51**), sonication (**Supplementary Fig. 48a**), and peeling (**Supplementary Fig. 48c**); chemical insults including exposure to extreme pH (**Supplementary Fig. 47a, b**) and salt solutions (**Supplementary Fig. 47c, d**); and thermal insults (**Supplementary Fig. 48b**). In all of these tests, our ELR coating remained stable and functional, assessed by SEM observations of structure and texture (**Supplementary Fig. 47, 48**), measurements of stiffness and hardness (**Supplementary Fig. 49a, b**), and the capacity to remineralize enamel (**Fig. 6l, m, n and Supplementary Fig. 47, 48**).

Another novelty is the capacity of our ELR coating to also trigger the epitaxial and organized aprismatic enamel-like layer from exposed dentine (**Fig. 5a-e**), recreating the dentine enamel junction (DEJ) (**Fig. 5 d, e**), and exhibiting enamel-like properties (**Fig. 5f, g, h**). Overall, our results demonstrated the capacity of our platform to recreate the microarchitecture and restore key functional properties of the different anatomical regions of enamel irrespective of the level of erosion down to bare dentine. To the best of our knowledge, this level and quality of remineralization and enamel regeneration has not been reported before.

In an attempt to further facilitate assessment of the clear advances of our study compared to the state-of-the-art, briefly comparing with key recent papers:

- **Compared to Nat. Commun., 2025, 16, 58;** our technology enables enamel regeneration by regaining the architecture of aprismatic, prismatic, and interprismatic enamel and

restoring mechanical strength, microtribological performance, and long-term stability under brushing, mastication, and acid exposure. In contrast, the approach reported in Nat. Commun. 2025, 16, 58 does not achieve epitaxial mineralization and lacks recreation and comprehensive characterization of enamel microstructure, mechanical recovery, or functional durability. Additionally, our technology is able to epitaxially grow an aprismatic enamel-like layer on dentine and offers control over mineral growth. This has not been reported by this other study.

- **Compared to Sci. Adv. 2019, 5, eaaw9569**; we have demonstrated high biocompatibility of our matrix (comprising of FDA approved components) compared to toxic chemical used in their work. Moreover, our technology enables coating deposition in 4 minutes under clinically-friendly conditions compared to 15 minute duration for the deposition of initial coating on tooth surface. Furthermore, through extensive testing under physical and chemical insults, we demonstrated the stability and functionality of the ELR organic coating as well as the resulting mineralized layer, which was not performed in this other study.
- **Compared to Adv. Mater. 2020, 32, 2002080**; we have demonstrated recreation of all the anatomical features of enamel including prismatic, interprismatic, and aprismatic regions compared to only prismatic enamel in their work. Furthermore, we carried out multifunctional assessment of the mechanical properties of the regenerated mineral layer (toothbrushing abrasion equivalent to 1 year, chewing and grinding abrasion equivalent to 3.5 years, fracture toughness, wear strength, stiffness, and hardness) compared to only stiffness and hardness in their work. Finally, their technology requires a slow fabrication process (i.e., 50 minutes), has significant challenges in terms of scalability, and offers limited control over the mineral growth. In comparison, our technology can be easily and rapidly applied (within 4 minutes), is user-friendly, cost effective, can be scaled up, and offers control over the mineral growth.

The comments are as follows:

1. In introduction part, the motivation behind work to develop an ELR-based strategy to regulate the epitaxial growth of enamel, is still unclear. In most cases, the epitaxial growth of crystals does not require regulation by an organic matrix template; its driving force is the intrinsic dipole field within the crystal. The key challenge is in controlling the kinetics of the process. Most studies focus on stabilizing or destabilizing small-sized precursors (such as ACP) to regulate epitaxial growth. However, this work does not clearly elucidate the role of ELR, lacking both logical depth and a comprehensive analysis.

Response: We thank the reviewer for highlighting this important point. We have now more clearly stated the rationale behind the design of our ELR matrix in both the Introduction on Page 2 and in a new Rationale paragraph at the beginning of the results section on Page 3 as follows:

Page 2 (new text in the Introduction):

In this study, we report on the design and performance of a clinically-friendly supramolecular protein matrix using disordered elastin-like recombinamer (ELR) molecules to emulate key structural and functional features of the β -rich fibrillar amelogenin matrix driving epitaxial and hierarchical mineralization in dental enamel.

Page 3 (new rationale paragraph):

2.1. Rationale for the design of the mineralizing ELR matrix to emulate enamel development
Recent studies have shown that intrinsically disordered amelogenin molecules assemble into functional β -rich fibrillar structures via intermolecular interactions in the presence of Ca^{2+} ions [29, 30]. These structures nucleate and grow apatite nanocrystals *in vitro* and during enamel

development [4, 5]. We have demonstrated the capacity to grow organized mineralized structures by modulating the levels of order and disorder of an intrinsically disordered elastin-like recombinamer (ELR) consisting of hydrophobic (VPGIG), hydrophilic (VPGKG), and acidic statherin (DDDEEKFLRRIGRFG) motifs [31]. We hypothesized that the disordered nature of the ELR molecules can be harnessed to generate supramolecular ELR ensembles that imitate structural and functional characteristics of the amelogenin matrix [2-4]. By generating these amelogenin-like β -rich supramolecular assemblies over enamel or dentine, it may be possible to facilitate epitaxial, integrated, and hierarchical enamel-like mineralization from dental tissues. Thus, we combined Ca^{2+} ions with ELR molecules, which are reported to promote β -sheet conformation in proteins [32] and peptides [33] via ion bridge formation. We also used drying as a mechanism to induce molecular order through molecular crowding, as previously reported [34, 35]. We reasoned that the incorporation of Ca^{2+} ions and drying of the ELR solution can together be used to promote β -rich fibrillar ELR structures, which we refer to as 'ELR fibrils'. Furthermore, the incorporated Ca^{2+} ions would also serve as nucleating points to facilitate mineralization. In addition, given the capacity of the ELR matrix to create a confined environment for mineral nucleation and growth [31], we envisage that the thickness of the ELR layer applied to dental tissue can spatially regulate the resulting mineral layer thickness *in situ*. To further support clinical translation, the choice of components in the formulation has been guided by considerations of rapid application, ease of use, and overall practicality in a clinical setting.

In the next comment, we have elaborated our answer around the mechanism behind our ELR matrix.

2. The presentation and description of the data are also puzzling. The paper emphasizes the templating function of ELR, but the data presented, including all the TEM and SEM images, are intended solely to demonstrate epitaxial growth. These data do not prove that epitaxial growth is achieved through the templating effect. In other words, based on the current data, the claim that the ELR template promotes the epitaxial growth of enamel is unconvincing.

Response: We thank the reviewer for pointing this out and apologize if there has been a misunderstanding. We have now modified the text on Page 6 to clarify which data in our manuscript demonstrate templating and which support epitaxial mineralization. Briefly, summarizing here:

Templating is demonstrated in **Fig. 1f**, where ELR fibrils exposed to a mineralizing solution initiated the nucleation and deposition of an amorphous mineral phase within 2 hours, which subsequently transformed into crystalline apatite aligned along the fibril axis. This process reflects a typical pathway of biomineralization [50]. Similar mechanisms have been reported in amelogenin-rich enamel matrices [4, 5], as well as in systems involving amelogenin peptide-templated [48] and other protein-templated mineralization [31]. To make this clear, we have now implemented changes in the text on Page 6 as follows:

Upon exposure to a mineralization solution supersaturated with respect to fluorapatite [31], **ELR fibrils templated the formation** of ~20-40 nm thick needle-like amorphous mineral platelets within 2 h (**Fig. 1f, (ii)**), which evolved into highly crystalline fluorapatite nanocrystals after 24 h (**Fig. 1f, (iii), Supplementary Fig. 10**). These nanocrystals were ~50 nm in diameter, ~1 μm in length, and aligned along the direction of the ELR fibrils. **These geometrical characteristics are similar to those growing within the amelogenin-rich matrix during enamel development [4, 5], as well as in systems involving amelogenin peptide [48] and other protein-templated mineralization [31].**

Epitaxial mineralization is demonstrated in different sections of the manuscript. First, **Fig. 2a** shows how the amorphous mineral precursor is aligned along the c-axis of existing nanocrystals

while embedded within the ELR matrix. This precursor subsequently transforms into crystalline apatite with the same orientation, consistent with classical amorphous-to-crystalline transitions observed during apatite formation [27, 50]. These results indicate that the supramolecular ELR matrix facilitates the oriented, integrated, and anisotropic growth of the underlying enamel crystals (**Fig. 3d, e**). Aligned with the goal of our study, this process is reminiscent of the mineralization mechanism observed during amelogenesis (Ref. [4, 5]). This is also the reason why we have used terminology that is used when describing nucleation and growth of nanocrystals during enamel development [4, 5, and PNAS, 2022, 119 (19), e2106965119] and repair [27]. To make this point clearer, we have modified the text on Page 6.

Page 6:

During early enamel development, the amelogenin matrix stabilizes and fuses prenucleation clusters via a non-classical crystallization process, guiding the growth of apatite nanocrystals preferentially along the c-axis [4, 48]. Thus, we focussed on assessing the capacity of the ELR matrix to preferentially control the direction of crystal growth. To do this, we embedded commercially available HAp nanocrystals within the ELR matrix deposited on the TEM grid. After 24 h in the mineralization solution, precursor amorphous mineral was observed, which may arise by the fusion of prenucleation clusters along the c-axis of the existing nanocrystals (**Fig. 2a**, blue square), as previously reported [48, 49]. This amorphous mineral undergoes transformation into crystalline structures with identical orientation (**Fig. 2a**, green square), as seen in classical amorphous-to-crystalline transitions during apatite formation [27, 50]. **These results demonstrate the capacity of our supramolecular ELR matrix to facilitate the oriented, integrated, and anisotropic growth of the existing crystals, reminiscent of the mineralization mechanism observed during amelogenesis [4, 5].**

Second, **Supplementary Fig. 11** provides *in silico* evidence of epitaxial mineralization. Here, umbrella sampling simulations revealed that ELR fragments exhibit stronger binding to the a-axis than the c-axis of apatite nanocrystals. This finding is consistent with previous reports using a 12-mer amelogenin sequence [48], suggesting that detachment from the a-axis requires more energy, thereby energetically favouring crystal growth along the c-axis as evident in **Fig. 2a**. We have also modified the text on Page 6.

Page 6:

In addition, these results were complemented with umbrella sampling simulations, which revealed that more energy is required to remove ELR fragments bound on the a-axis than the c-axis of the apatite nanocrystal (**Supplementary Fig. 11**), as reported previously for a small 12-mer amelogenin sequence [51], thus energetically favouring crystal growth along the c-axis. In contrast, nanocrystals mineralized without an ELR matrix exhibited irregular growth due to the fusion of amorphous mineral along all the crystal axes (**Supplementary Fig. 12**). **Overall, these results confirmed the role of the ELR matrix in directing epitaxial crystal growth preferentially along the c-axis.**

Third, **Fig. 3d, e** and **Fig. 4** provide direct evidence of epitaxial mineralization using HRTEM, SAED, and SEM, which together are commonly used to demonstrate epitaxial mineralization. These tools reveal the lattice-level orientation, integration, and alignment of apatite crystals, allowing us to identify epitaxial growth interfaces.

Fourth, in addition to direct evidence of epitaxial mineralization, indirect evidence presented in **Fig. 3f–h** and **Fig. 6a–f** supports the stability of the ELR-induced remineralized enamel-like layer when assessed under mechanical tests that infer its crystallographic integration to the underlying tissue. Further details on this point are provided in response to Question #A.

3. There is a lack of animal models or preclinical trial data, and the experiments using natural saliva from different human donors as a substitute for animal experiments are questionable, as factors such as saliva washing and microbial interference have not been considered. In particular, the authors stated in their response letter that almost all animal models are unacceptable, yet they did not specify the reasons, which is puzzling. I speculate that the fundamental reason the authors mentioned for not conducting animal experiments is that the preparation of the ELR membrane requires the use of organic reagents that pose health risks; to date, the authors have not overcome this limitation. Therefore, at present, the use of ELR is quite restricted.

Response: We appreciate the reviewer's emphasis on the importance of validating the translational potential of our technology in animal models. We agree with the reviewer that *in vitro* data alone does not fully recreate the complexity of the *in vivo* environment and have therefore added a comment in the conclusion section on Page 17 to acknowledge this limitation of our study:

Moreover, our findings demonstrate a capacity to grow enamel-like structures under conditions that closely imitate various mechanical and chemical challenges found in the mouth. However, these tests do not fully recreate the complexity of the *in vivo* oral environment and, consequently, to fully confirm the capacity to regenerate natural enamel would require *in vivo* validation, which we envision to pursue in future work.

In addition, while testing our material in animals is not a viable option for us within our geographical location (as we explained in our previous responses), we intend to conduct clinical testing in a future study. Nonetheless, we hope that the reviewer appreciates our effort to conduct *in vitro* studies that can test key challenges that are found in the *in vivo* scenario. First, we conducted experiments to assess the mineralization using natural saliva from different human donors. The results confirmed the restoration of prismatic structures (**Fig. 6j** and **Supplementary Fig. 46a**), mechanical (stiffness and hardness) (**Fig. 6k** and **Supplementary Fig. 46c**), and chemical properties (**Supplementary Fig. 46d, e**) of enamel. Also, in addition to using human saliva, we have tested mineralization under dynamic and abrasive conditions, as well as in the presence of salivary enzymes to simulate washing, physical forces, and enzymatically degrading environments. These results are provided in **Supplementary Fig. 45**. We have also tested our platform under extreme conditions of pH, low Ca concentration, salt solutions, high temperature, physical forces such as sonication and adhesive tape peeling. These results are included in **Supplementary Figs. 47, 48, 49**.

We would like to take the opportunity to mention that the formulation of our material has been designed to facilitate translation. All of the components are currently being used in dental products, which facilitates their use from a regulatory stand-point. In addition, we are using both ethanol and glutaraldehyde at lower concentrations than those being used in other Food and Drug Administration (FDA) approved dental products such as ICON® and GLUMA®. We mentioned this in our paper on Page 16. In addition, we evaluated and confirmed the biocompatibility of the ELR matrix using several cell lines, which is presented in **Supplementary Figs. 37, 38, 39, 40**. In addition, it is opportune to mention that ELRs have been reported to exhibit excellent biocompatibility, both *in vitro* and *in vivo* (Biomaterials, 2014, 35(29), 8339-8347; Acta Biomaterialia, 2014, 10 (1), 134-141; Front. Bioeng. Biotechnol., 2022, 10, 836386; Acta Biomaterialia, 2010, 6(6), 2108-2115).

4. The repair efficiency is average: approximately 10 microns grown in 10 days, which is not very fast and lacks clinical immediacy, potentially limiting its practical applications (such as rapid

outpatient treatment). In contrast, a recent publication in Nature Communications (2025, 16:58) showed a growth of 100–300 nm in 30 minutes, indicating a much faster speed, with the structure of the repair layer closely resembling natural enamel.

Response: We appreciate the reviewer’s comment. As our study serves as proof-of-concept, our primary focus was to demonstrate the feasibility, mechanism, and translation of our biomimetic approach. The capacity to accelerate mineralization is an exciting possibility and we will look to work on this in a future study. Nevertheless, it is important to highlight that while numerous remineralization strategies exist, the primary innovation of our study lies beyond merely remineralization, but in the ability to regenerate enamel in a biomimetic and translational manner. We have now created a rationale paragraph on Page 3 to make this point clearer.

5. Although the paper conducted a relatively systematic study on the mechanical properties of the repaired enamel layer, the mineralization period in the study is uniformly 10 days, which is quite long. Moreover, the research on mechanical properties and stability focuses only on the conditions before and after remineralization, without mentioning the stability and mechanical properties during the 10-day mineralization process, when mineralization is not yet complete.

Response: We appreciate the reviewer’s comment and suggestion. While we agree with the reviewer that investigating the mechanical properties at different stages of mineralization would provide valuable insights, this is currently outside the scope of our study. We plan to include this analysis in future studies to build a more comprehensive understanding of the temporal evolution of mechanical properties during enamel regeneration.

6. There are many technical issues.

(i) For example, in Figure 1(b), although the authors indicate that the scattering peaks originate from CaCl₂ in the sample, they still cannot explain these strange signals.

Response: We thank the reviewer for highlighting this important point. We apologise that we did not assign the other peaks in the 1D WAXS previously. We identified the peak at 1.04 Å⁻¹ that corresponds to interactions between α-helices. Peaks at q = 1.07, 1.49, 1.72, and 2.0 Å⁻¹ align with the (001), (111), (112), and (211) Miller indices of CaCl₂ suggesting peaks arising from CaCl₂ crystallization. We have now incorporated these details in the revised manuscript on Page 4 as:

Peak at q = 1.04 Å⁻¹ can be attributed to interactions between α-helices [41] while peaks at q = 1.07, 1.49, 1.72, and 2.0 Å⁻¹ align with the (001), (111), (112), and (211) Miller indices of CaCl₂ suggesting peaks arising from CaCl₂ crystallization.

We have also characterized the peaks from the 1D WAXS of CaCl₂·2H₂O only sample and included the following description in the **Supplementary Fig. 8** as follows:

Using q values from 1D WAXS, we calculated the d-spacings and 2θ angles for CaCl₂·2H₂O using the following expressions:

$$d = 2\pi/q \quad (\text{Eq. 1})$$

$$q = 4\pi \sin\theta/\lambda \quad (\text{Eq. 2})$$

Cu Kα radiation exhibiting a wavelength (λ) of 1.5418 Å was used to compute above 2θ values. Powder Diffraction File (PDF) 01-075-0305 from the International Centre for Diffraction Data (ICDD) database was used to calculate the Miller indices as shown below.

Table showing q values, d-spacing, 2θ (degrees), and Miller indices for CaCl₂·2H₂O crystals.

q (Å ⁻¹)	d-spacing (Å)	2θ (degrees)	Miller indices
1.07	5.87	15.1	(001) plane
1.25	5.03	17.6	(002) plane

1.38	4.52	19.5	(110) plane
1.53	4.11	21.6	(111) plane
1.77	3.55	25.1	(112) plane
2.00	3.14	28.4	(211) plane

(ii) In the two-dimensional image, the scattering ring corresponding to 0.47 nm is clearly diffused, yet why is it so sharp in the one-dimensional integrated curve? If the authors claim that the sample is impure, I consider this characterization very poor, as how can the authors be sure that the sharp 0.47 nm peak does not come from another salt?

Response: We appreciate the reviewer's point and believe that this can be subjective. We respectfully believe that the scattering ring corresponding to 0.47 nm is not diffused as it is of similar quality to others reporting on nanofibrillar structures such as Prion, 2008, 2(3), 112–117 and Structure, 2003, 11(8), 915-926. Moreover, it corresponds to the sharp peak in the 1D integrated curve, which is a characteristic feature of nanofibrils (Ref. [38, 47]).

We also have confirmed that the sharp 0.47 nm peak originates from the ELR fibrillar structure and is not due to the presence of salt crystals. To verify this, we conducted a control experiment using a CaCl₂-only sample and confirmed that the 0.47 nm peak does not result from CaCl₂ crystallization.

(iii) Additionally, in Figure 3(b), why does the ELR membrane stained with ThT not appear as the continuously phased membrane previously displayed, but instead looks like fish-scale enamel after corrosion?

Response: We thank the reviewer for highlighting this important point. The ELR coating was deposited on an acid-etched enamel surface, which exhibits a fish-scale-like morphology. The discontinuous appearance of the ThT-stained ELR-coated enamel image is a visual effect caused by the ELR coating uniformly conforming to the enamel surface as shown in **Supplementary Fig. 49c, 50**. To make this point clear in the Manuscript, we have now repeated the experiment and provided a high magnification image in **Supplementary Fig. 19c, d** as shown below:

Supplementary Fig. 19: (a) SEM image showing 2 μm thick ELR coating on enamel surface and (b) confocal laser scanning microscopy (CLSM) image of bare enamel section (i.e., without ELR

coating) after ThT staining. **(c)** CLSM image showing ThT staining of an ELR coated enamel surface. **(d)** Illustrations showing how different regions of the sample may appear more or less green, depending on the section of the image. The discontinuous or dark spot features observed in the image in panel (c) result from the sectioning of the confocal image. The ELR coating conforms closely to the underlying enamel topography and so even though the ELR coating is uniform, dark regions may appear depending on the depth at which the confocal image is taken. In other words, dark regions will appear when the focal plane of the confocal image goes through densely packed enamel crystals where the ELR matrix is unable to penetrate as shown in **d-i** and **d-ii**. In contrast, regions where the ELR is present appear bright green.

(iv) Furthermore, the FIB image in Figure 5(e) shows that after mineralization, the structure is not very dense—is it necessary to present a larger-scale cross-sectional FIB image?

Response: We thank the Reviewer for bringing this to our attention. Figure 5(d) already includes TEM images demonstrating denser mineralization on the dentine surface. However, we have now provided an additional large-scale cross-sectional FIB-SEM and TEM images to more clearly illustrate the high nanocrystal density and extent of mineralization growing from the dentine surface. These results are presented in **Supplementary Fig. 31** as shown below:

Supplementary Fig. 31. Integration between the mineralized layer and mineralized collagen fibrils (MCFs) at interface.

(a, b) Low magnification FIB-SEM images showing cross-section of the mineralized layer grown over dentine surface. White arrows point to the porous-looking dentine formed by 3D collagen network. **(c)** TEM image showing the cross-section of the integrated mineralized layer and dentine at interface. **(d)** TEM image showing integration between the mineralized layer and MCFs of dentine.